# Modeling dispersal of the Pearl River-derived sediment over the Shelf

Guang Zhang[1,2,3], Suan Hu[1,2,3], Xiaolong Yu[1,2,3], Heng Zhang[1,2,3], Wenping Gong[1,2,3]*

[1] School of Marine Sciences, Sun Yat-sen University, Zhuhai, 519082, China

[2] Guangdong Provincial Key Laboratory of Marine Resources and Coastal Engineering, Zhuhai, 519082, China

[3] Pearl River Estuary Marine Ecosystem Research Station, Ministry of Education, Zhuhai 519082, China

Corresponding author: Wenping Gong (gongwp@mail.sysu.edu.cn)

**Abstract**
This study employs the Coupled-Ocean-Atmosphere-Wave-Sediment-Transport
(COAWST) modeling system to quantitatively assess the seasonal suspension,
transport, and annual fate of Pearl River-derived sediment (riverine slow-settling
single fine grains and high-settling flocs) on the northern continental shelf of the
South China Sea (SCS). Following careful model validation, a series of sensitivity
experiments were conducted to investigate the effects of tides, waves, background
circulation, sediment settling velocity, critical shear stress, and sediment spin-up
durations. The results reveal strong seasonal variations in sediment dynamics driven
by the East Asian monsoon. During the wet summer, weaker hydrodynamic
conditions promote the initial deposition of riverine sediment via the surface buoyant
plume. In contrast, stronger winds and waves during winter enhance sediment
resuspension and southwestward transport, particularly toward the Beibu Gulf.
Spatially, approximately two-thirds of the annual Pearl River-derived sediment load is
retained near the estuary. About 9% reaches the continental shelf east of the PRE,
while similar proportions accumulate in the Beibu Gulf and south of Hainan Island.
Sensitivity experiments highlight the distinct roles of different physical forcings: tidal
dynamics strongly influence sediment behavior in the estuary, where the absence of
tidal forcing reduces bottom shear stress, leading to increased local deposition and
reduced offshore transport. Wave forcing plays a dominant role in sediment
resuspension near the river mouth and along the coast, especially during winter.
Excluding waves leads to greater sediment retention near the estuary and diminished
transport toward distant regions. Ambient circulation, particularly in summer, is
essential for eastward sediment transport; when it is omitted, northeastward dispersal
is greatly diminished. Model outcomes are also sensitive to sediment parameterization.
Using non-seasonal critical shear stress for erosion increases wintertime sediment
mobility east of the Leizhou Peninsula. Higher settling velocities decrease suspended
sediment concentrations and promote near-source retention, limiting long-distance
transport. Spin-up duration experiments indicate that Pearl River–derived sediment,
which enters and accumulates in various regions of the model domain during the first
year, continues to migrate southwestward in the second year under the influence of
the mean annual flow field. In contrast, the spin-up duration of seabed sediment has
little impact on the retentions of Pearl River–derived sediment on the shelf. Overall,
this study reveals the transport pathway and fate of the Pearl River-derived sediment
and provides a model-based assessment of its seasonal behavior and dispersal
mechanisms on the northern SCS shelf. It identifies key physical drivers regulating
sediment transport and deposition patterns, offering new insight into sediment fate in
a monsoon-dominated shelf system.
**Keywords**
Riverine sediment transport; Sediment retention; Numerical modeling; Pearl River
Estuary
**1. Introduction**
The transport process of suspended sediment from river source to ocean sink is an
important link in the global material cycle (Geyer et al., 2004; McKee et al., 2004;
Kuehl et al., 2016; Liu et al., 2016; Cao et al., 2019). Much of the riverine sediment is
trapped on the shallow shoals in estuaries, while the rest is transported by buoyant
plume out of the estuary (Meade, 1969; Burchard et al., 2018; Zhang et al., 2019). The
riverine sediment carried by the buoyant plume has a significant impact on the water
quality, ecology, and geomorphology of the estuaries and continental shelves (Wright
and Coleman, 1973; Turner and Millward, 2002).

The transport and deposition of riverine sediments from river source to estuarine,

coastal, and shelf environments are controlled by diverse physical processes,
including tidal forces, wave action, and shelf circulation dynamics (Dalyander et al.,
2013; Gao and Collins, 2014; Xu et al., 2016; Warner et al., 2017; Zang et al., 2019;
Wang et al., 2020). Tides play a critical role in sediment transport dynamics in
estuarine and shelf regions, as spring tides typically produce higher bed shear stress,
enhanced sediment resuspension, and greater offshore sediment transport flux
compared to neap tides (Bever and MacWilliams, 2013; Zhang et al., 2019; Wang et
al., 2020). In nearshore regions, wave-induced bed shear stress often exceeds
current-induced stress by an order of magnitude (Xue et al., 2012; Dalyander et al.,
2013). Furthermore, wave-driven sediment resuspension frequently surpasses, and is
often several times greater than, the peak levels achieved by current-induced
resuspension (Sanford, 1994; Harris et al., 2008; Brand et al., 2010; Xu et al., 2016).
In shelf regions, circulation patterns significantly modulate sediment transport, with
the magnitude of along-shelf transport substantially exceeding the cross-shelf
component in most areas (Nittrouer and Wright, 1994; Geyer et al., 2004; Gao and
Collins, 2014; Wang et al., 2020).
Furthermore, sediment properties, including settling velocity (Xia et al., 2004;
Chen et al., 2010; Cheng et al., 2013), critical shear stress for erosion (Dong et al.,
2020), and bed grain size distribution (Xue et al., 2012; Bever and MacWilliams,
2013), significantly influence sediment transport dynamics and
deposition/resuspension processes. Settling velocity can influence the location of
sediment depocenters, with higher settling velocities leading to more proximal
entrapment and vice versa (Ralston and Geyer, 2017). Similarly, critical shear stress
for erosion can affect the resuspension of deposited sediment, with higher critical
shear stress resulting in less resuspension and more deposition especially during neap
tides and weak wind wave periods (Dong et al., 2020; Choi et al., 2023).
A comprehensive understanding of sediment transport and deposition from river
source to ocean sink requires the integrated consideration of both physical forcing
factors and inherent sediment characteristics. Here, we present the transport and
deposition of the Pearl River-derived sediments on the continental shelf as a case
study. The Pearl River, ranking as China's second-largest river in terms of freshwater
discharge (Hu et al., 2011), forms the Pearl River Estuary (PRE) in its lower reaches
(Figures 1 and S1). Its freshwater and sediment discharge are primarily delivered
through eight major outlets (Figure S1b; Wu et al., 2016; Zhang et al., 2019; Zhang et
al., 2025), forming distinct buoyant plumes that extend across the northern South
China Sea (SCS) shelf (Zhang et al., 2025). The present average annual (2001-2022)
freshwater and riverine sediment loads are $2.74 \times 10^{11}$ m$^3$ and $2.84 \times 10^7$ tons,

respectively, as reported by the Ministry of Water Resources of the People's Republic

of China (http://www.mwr.gov.cn/sj/#tjgb). The distribution of these inputs shows

significant seasonal variability: approximately 80% of the freshwater and 95% of the

sediment load are transported during the wet summer season (April to September),

while the remaining portion is discharged during the dry winter season (Xia et al.,

2004).

The northern SCS, shaped by the East Asian Monsoon, displays marked seasonal

contrasts, featuring winter monsoon winds averaging 7-10 m s$^{-1}$ and summer winds

typically below 6 m s$^{-1}$ (Su, 2004; Ou et al., 2009). This seasonal shift drives coastal

currents: northeastward in summer and southwestward in winter (Gan et al., 2009;

Gan et al., 2013). Beyond the coastal zone, the consistent SCS Warm Current

(SCSWC) flows northeastward along the shelf break and inner continental slope

toward the Taiwan Strait, originating near Hainan Island and persisting year-round,

even during the winter northeast monsoon, across a remarkable distance of 600-700

km to the southern tip of the Taiwan Strait (Su, 2004; Yang et al., 2008).

The PRE is situated in the central part of the northern South China Sea boundary,

positioned between the Taiwan Banks and Hainan Island. The PRE has a micro-tidal

and mixed semi-diurnal regime, with daily inequality in the range and in the time

between the high and low tides (Mao et al., 2004). The neap and spring tides

alternately influence the water elevation downstream of the estuary, with tidal ranges

varying from approximately 0.7 m during neap tides to over 2 m during spring tides

(Chen et al., 2016; Gong et al., 2018b). The PRE and the nearby shelf exhibit strong

seasonal variation in water column stability and are highly stratified during the wet summer season, while the PRE becomes partially mixed or vertically well-mixed during the dry winter season (Dong et al., 2004). Offshore of the PRE region, wave conditions display distinct seasonal patterns: the waves are mild during summer, and become stronger during winter, marked by larger southeasterly waves (Gong et al., 2018a; Gong et al., 2018b; Zhang et al., 2021).

Previous studies have focused on sediment transport within the PRE (Zhang et al., 2019; Zhang et al., 2021; Ma et al., 2024). Most Pearl River-derived sediments are deposited within the estuary, and neglecting tidal effects can lead to higher deposition rates and lower offshore sediment flux when compared to those with tides (Hu et al., 2011). The depositional dynamics of sediments from different PRE outlets are regulated by outlet location, topography, and tidal conditions, with neap tides favoring sediment accumulation on shoals and spring tides driving erosion and enhancing offshore sediment transport (Zhang et al., 2019). Waves further intensify both lateral trapping within the PRE and offshore sediment transport (Liu and Cai, 2019; Zhang et al., 2021).

However, numerical studies on the transport of the Pearl River-derived sediments across the continental shelf remain scarce, even amidst the widespread adoption of computer modeling approaches. Previous research on the distribution of these sediments has primarily relied on analyses of seismic profiles, gravity cores, and laboratory-based radiometric dating of sediment samples (Ge et al., 2014; Liu et al., 2014; Cao et al., 2019; Lin et al., 2020; Chen et al., 2023). Outside the PRE, gravity

core and seismic survey data were used to examine the Holocene sedimentary
processes, revealing two distinct mud depo-centers: an eastward proximal depo-center
extending southeastward from the PRE's mouth and a southwestward distal mud belt
(Ge et al., 2014; Liu et al., 2014; Chen et al., 2023). However, seismic and drilling
data cannot confirm that the Pearl River sediment can be transported to the Beibu
Gulf (Ge et al., 2014). Due to the lack of sufficient gravity core samples and seismic
data, it is difficult to quantitatively attribute the sediment in the Beibu Gulf to the
Pearl River-derived sediment (Cao et al., 2019). Afterward, Lin et al. (2020) used the
$^{226}Ra$—$^{238}U$ and $^{232}Th$—$^{238}U$ endmembers model based on measurements of
radionuclides in the surface sediment samples. They found that approximately 15% of
the surface sediment in the nearshore area of the Beibu Gulf originates from the PRE
region. However, their studies only address the proportion of the PRE sediment in the
surface sediment of the Beibu Gulf, without directly indicating the seasonal transport
pathways, flux, and annual deposition of the sediment from the Pearl River.
A gap persists in understanding how physical processes (such as tides, waves, and
ambient circulations) and sediment characteristics (such as critical shear stress for
erosion, settling velocity) and sediment initial conditions influence the seasonal
suspension, transport, and annual deposition of the Pearl River-derived sediment on
the shelf. Specifically, this study focuses on sediment classes 4 and 5 in Table 1,
which represent the Pearl River-derived components: slow-settling single fine grains
(Class 4) and fast-settling flocs (Class 5), in contrast to the background seabed
sediments represented by classes 1 to 3. To address this, we utilize numerical
modeling, complemented by extensive collection of field observations and seabed
grain size distribution data for model calibration and validation—a highly effective
approach for exploring mechanisms and testing hypotheses derived from limited
observational datasets. This study aims to systematically investigate the dispersal
dynamics of the Pearl River-derived sediment over the northern South China Sea shelf,
with particular emphasis on the following objectives:
(1) Quantify the seasonal dispersal and annual deposition of the Pearl
River-derived sediment (classes 4 and 5 in Table 1) over the continental shelf.
(2) Examine the relative roles of physical forcings (tides, waves, and ambient
circulations), sediment characteristics (critical shear stress for erosion, settling
velocity) and (Pearl River-derived versus Seabed) sediment spin-up durations on the
dispersal of the Pearl River-derived sediment.

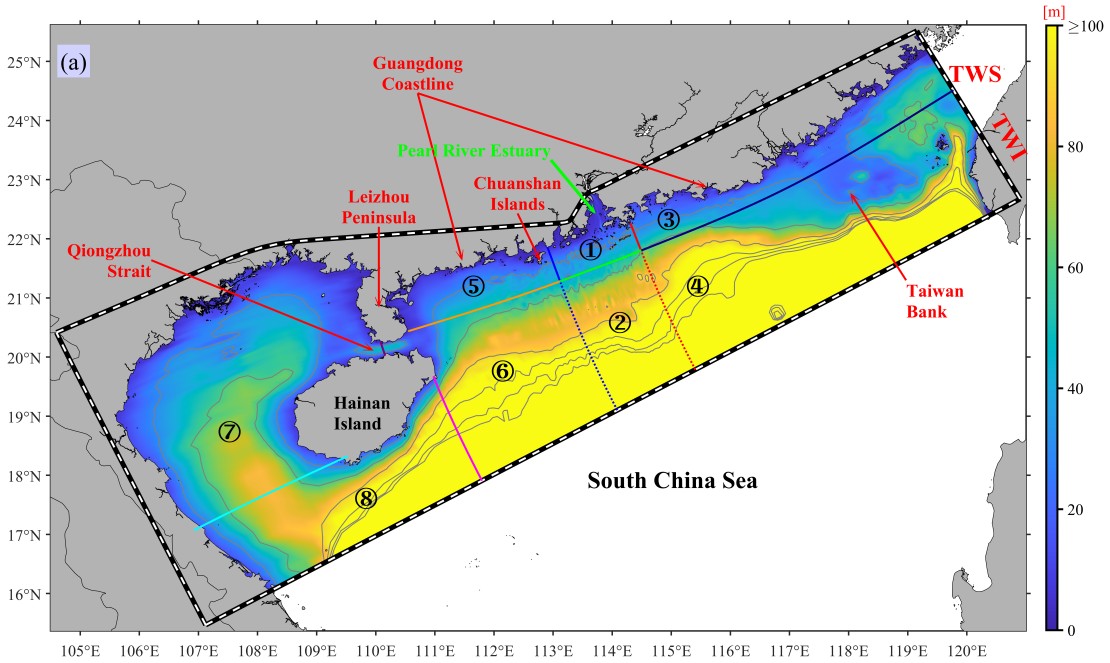


**Figure 1.** Bathymetry (shading) and isobath contours of the study area, with the ROMS/SWAN model grid domain outlined by black-to-white dashed lines. Circled numbers ① - ⑧ denote the eight regions: "Proximal", "Southern", "Eastern", "Southeastern", "Western", "Southwestern", "Gulf", and "Distal" regions, as defined by transects and detailed in Section 2.1. The abbreviations TWI and TWS mean Taiwan Island and Taiwan Strait, respectively. The gray contours represent 30-180 m isobaths at 30 m intervals, a consistent feature maintained in all subsequent figures that include these isobath contours.

## 2. Methods

## 2.1 Model coupling

This study employed the Coupled Ocean Atmosphere Wave Sediment Transport (COAWST, version 3.4) modeling system (Warner et al., 2005; Warner et al., 2008; Warner et al., 2010), which includes a Model Coupling Toolkit (MCT) to facilitate data exchange among different modules (Jacob et al., 2005; Larson et al., 2005). The COAWST system consists of several modeling components, mainly comprises a hydrodynamic module (Regional Ocean Modeling System; ROMS) (Shchepetkin and McWilliams, 2005; Haidvogel et al., 2008), an atmospheric module (Advanced Research Weather Research and Forecasting; WRF) (Skamarock et al., 2005), a wave module (Simulating Waves Nearshore; SWAN) (Booij et al., 1999), and a sediment transport module (Community Sediment Transport Modeling System; CSTM) (Warner et al., 2008).

In this study, we established a coupling between ROMS, SWAN, and CSTM. The

model grid covered the northern continental shelf of the South China Sea, including
the PRE (Figure 1). The regional model was configured with $170 \times 482$ horizontal
grid cells, with horizontal resolution varying from approximately 0.1 km near the PRE
to about 10 km at outer open boundaries (Hu et al., 2024; Zhang et al., 2025). The
model grid bathymetry data was obtained from nautical charts compiled by the China
Maritime Safety Administration and the General Bathymetric Chart of the Oceans
(GEBCO) (Weatherall et al., 2015). The vertical grid used a terrain-following
S-coordinate system (Song and Haidvogel, 1994) with 20 layers and a stretching
transformation for higher resolution near the surface and bottom. For model
validations, please refer to the Supplementary Material (Supplement Figures S1-S10).
To improve the understanding of the spatial-temporal variabilities in the riverine
sediment dispersal, and the estimation of the fate of the Pearl River sediment during
the wet summer season, dry winter season, and throughout the year, we partitioned the
model domain into eight distinct regions delineated by various transects as illustrated
in Figure 1. The division criteria are mainly based on the distance from the estuary
and the natural separation by the Leizhou Peninsula and Hainan Island (Figure 1).
These regions include:

① Proximal region: Proximity to the estuary,

② Southern region: Located deeper in the southern part of the estuary,

③ Eastern region: Eastern side of the estuary, closer to the shoreline,

④ Southeastern region: Further offshore on the eastern side of the estuary,

⑤ Western region: Western side of the estuary, closer to the shoreline,
⑥ Southwestern region: Offshore on the western side of the estuary,
⑦ Gulf region: Mainly the Beibu Gulf,
⑧ Distal region: South of the Hainan Island.
By dividing the model domain into these delineated regions, we calculated the
riverine sediment flux for each transect, thereby determining the total riverine
sediment volume retained in each region.

## 2.2 ROMS model setup

For the ROMS model, we utilized the Generic Length Scale turbulence closure
scheme (Warner et al., 2005) for vertical turbulence parameterization. The method of
Smagorinsky (1963) was employed to calculate the horizontal eddy viscosity and
diffusivity. The Flather and Chapman boundary conditions were applied to barotropic
current and water elevation at open boundaries, respectively (Flather, 1976; Chapman,
1985). Meanwhile, the open-boundary conditions for temperature, salinity, and
sediment concentration were imposed by radiation methods (Orlanski, 1976;
Raymond and Kuo, 1984). Surface forcing (including wind, net shortwave radiation,
air temperature, atmospheric pressure, specific/relative humidity, and rain, etc.) data
were sourced from the Climate Forecast System Reanalysis of the National Centers
for Environmental Prediction (NCEP) (Saha et al., 2014), with a temporal resolution
of 1 h and a spatial resolution of $0.3° \times 0.3°$. Water level and current open-boundary
conditions comprised two components: tidal and subtidal. The tidal component was
obtained from the Oregon State University Tidal Prediction Software database (Egbert
and Erofeeva, 2002), while the subtidal component was interpolated from the HYbrid
Coordinate Ocean Model (HYCOM) outputs (Chassignet et al., 2007).

## 2.3 Wave model setup

The SWAN model was executed and coupled to the same grid as the ROMS
model (Warner et al., 2010). It was driven by surface atmospheric forces, real-time
water level, and current fields from the ROMS and boundary reanalysis data. Wave
boundary conditions were specified using nonstationary wave parameters from
outputs of the NOAA WAVEWATCH III global ocean wave model solutions (Tolman
et al., 2016). Information was exchanged at 15-minute intervals to introduce
wave-current interaction (WCI) between the ROMS and SWAN models (McWilliams
et al., 2004; Kumar et al., 2012). This exchange included significant wave height
(Hsig), surface peak wave period, mean wave direction and length, wave energy
dissipation, and the percentage of breaking waves from SWAN to ROMS, as well as
water level and current from ROMS to SWAN.
Additionally, the wave-current bottom boundary module based on Madsen (1994),
was activated to simulate the wave-current bottom boundary layer. The vortex force
module of wave forces was also activated to compute the wave-induced momentum
flux, utilizing the method proposed by McWilliams et al. (2004) and implemented in
COAWST by Kumar et al. (2012). The bottom friction was computed based on a
logarithmic velocity profile (Warner et al., 2008).

## 2.4 Specifications of riverine input and sediment model

The freshwater discharge for the Pearl River was specified at the northern
boundary using daily measured data from the Pearl River Water Resources
Commission, while downstream precipitation within the Pearl River Basin was
neglected. The full simulation model was initialized on the first day of January 2016
using temperature, salinity, and current fields interpolated from the HYCOM model,
and it concluded on March 31, 2018. This study primarily analyzes the last 12 months,
specifically from April 1, 2017, to March 31, 2018. This year was selected because
the freshwater discharge and sediment load of the Pearl River closely approximated
the average values of the past two decades, with a runoff of $3.35 \times 10^{11}$ m$^3$ and a
sediment load of $3.45 \times 10^7$ tons, closely resembling the averages from 2001 to 2022.
Since the daily riverine sediment loads were unavailable, we modified the
previous research results on sediment rating curves (Zhang et al., 2012) to suit for our
study, as expressed by

$$y = 0.00002263x^{1.792} \tag{1}$$

where $y$ is the Pearl River-derived suspended sediment concentration (mg L$^{-1}$), $x$ is the
Pearl River freshwater discharge rate (m$^3$ s$^{-1}$). Based on this relationship, the total
amount of Pearl River sediment input over our 12-months study period (Figure 3b)
was 34.52 million tons, aligning closely with the annual load reported in 2017 by the
Pearl River Water Resources Commission. The riverine sediment input, derived from
the river discharge, was allocated across the eight outlets along the north boundary
(Figure S1b) based on the distribution approach of Hu et al. (2011). The subsequent
step involved establishing the proportion of seabed sediment particle size components.
Sediments are typically categorized into three grain-size classes: clay (0-4 μm), silt
(4-63 μm), and sand (63-2000 μm), as outlined by Shepard (1954). Data on sediment
particle size composition for the northern continental shelf of the South China Sea and
the PRE area were acquired through multiple voyage observations (Zhang et al., 2013;
Zhang et al., 2019). Furthermore, publicly available data from published literature
were compiled (Gao et al., 2007; Kirby et al., 2008; Gao et al., 2010; Huang et al.,
2013; Liu et al., 2014; Wang et al., 2014; Wang et al., 2015; Wang et al., 2016; Ge et
al., 2017; Lu et al., 2017; Zhong et al., 2017; Yang et al., 2018; Ge et al., 2019).
Finally, component distribution data for different particle size classes of seabed
sediment were obtained from a total of 1981 measured stations (Figure 2a-c).

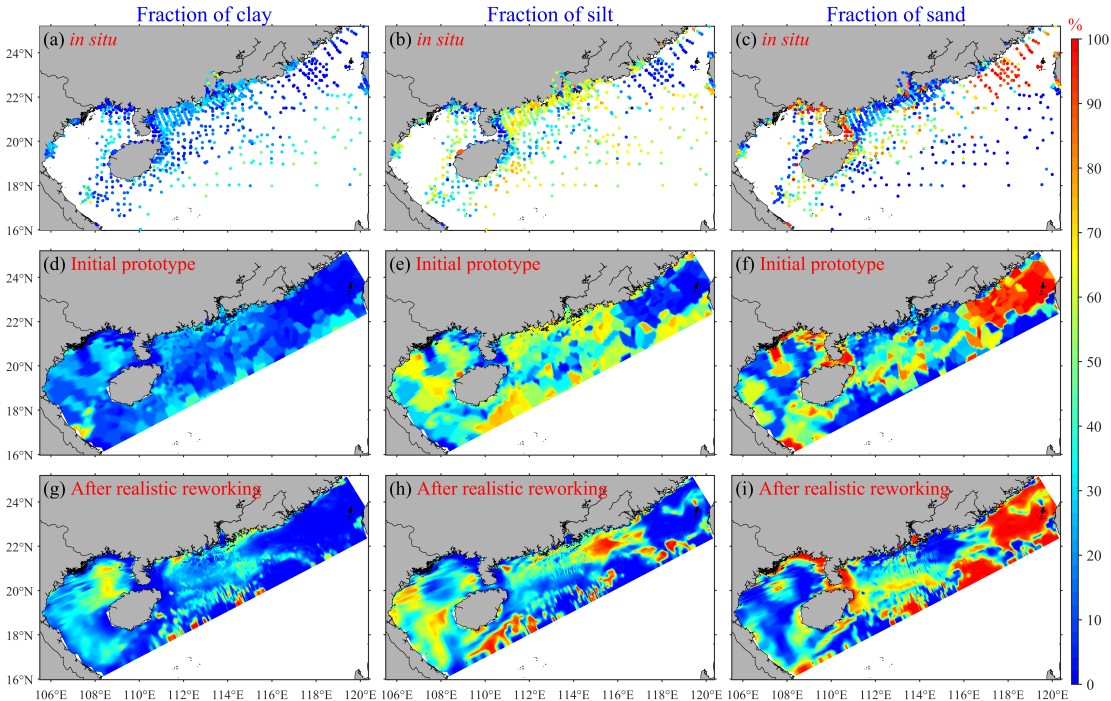

**Figure 2.** Row 1 presents the spatial distribution patterns of seabed sediment fractions
derived from 1981 sampling sites, while Row 2 demonstrates the initial spatial
distribution prototype of seabed sediment fractions developed based on the
observational data presented in Row 1. Row 3 shows the spatial distribution patterns
of seabed sediment fractions following the completion of spin-up phase in the Control
run case on April 1st, 2017, with Columns 1, 2, and 3 representing the fractions of
clay, silt, and sand, respectively.

As illustrated in Figure 2a-c, the measured stations exhibit a widespread
distribution, offering comprehensive coverage of the entire northern continental shelf
of the South China Sea, including the PRE. Particularly dense distribution is observed
in the PRE and the coastal areas of western Guangdong. These regions represent the
primary scope of transport and deposition associated with the Pearl River-derived
sediment. Hence, the stations utilized in this study well represent the distribution of
bed sediment particle size components in these study areas. It is evident that among
the stations in the offshore area of the northern continental shelf of the South China
Sea, silt dominates, followed by clay, while sand with the largest particle size is the
least abundant. This suggests a significant presence of terrestrial sediment or Pearl
River sediment in the offshore area of the northern continental shelf of the South
China Sea. It should be noted that the lack of in situ grain size distribution data in
specific regions of the model domain, especially in the Beibu Gulf area, may lead to
uncertainties in sediment transport predictions. We will address these limitations and
quantify their potential errors in the discussion part of this study.
To derive the component proportions of the initial prototype field on the model
grid, this study employed the Kriging method (Krige, 1951), widely recognized for
spatially interpolating various types of observational data. The sediment distribution
pattern obtained through interpolation (Figure 2d-f) closely resembles the original
1981 measured sediment particle size distribution patterns (Figure 2a-c), suggesting
the suitability of this interpolation method for the study area.
The initial prototype field underwent a 15-month spin-up period (from January 1,
2016, to March 31, 2017), during which the bottom sediment composition evolved
under realistic hydrodynamic forcings from the ROMS, SWAN, and CSTM models.
This method has been utilized in numerous previous studies, including those by Bever
et al. (2009), van der Wegen et al. (2010), and Zhang et al. (2021). This process
allows the initially idealized sediment distribution to evolve under realistic dynamic
forcings, including tides, waves, and currents, thereby minimizing unreasonable
spatial patterns introduced by the Kriging interpolation method. Such unreasonable
spatial patterns may arise due to limitations in the number, representativeness, and
timing of field sediment samples relative to the model start date. As a result, the
sediment field after the spin-up period (Figures 2g–i) exhibits spatial patterns that are
more physically plausible and better aligned with the hydrodynamic conditions of the
study region. During both the 15-month spin-up period and the subsequent 12-month
formal model experiments (see Section 2.6 and Table 2), the CSTM utilized five
sediment classes (Table 1), representing a range of sediment sizes and characteristics.
These included three types of seabed sediments (clay, silt, and sand, corresponding to
sediment Classes 1 to 3 in Table 1) and two types of Pearl River-derived sediments
(Class 4 and Class 5 in Table 1). The riverine sediments consisted of slow-settling
single fine grains (Class 4) and high-settling flocs (Class 5), which were delivered
into the model domain during both the 15-month spin-up period and the subsequent
12-month formal model experiments. The riverine flocs correspond to the flocculated
fractions of clay and silt, whereas the single fine grains represent the non-flocculated
components within the Pearl River-derived sediments, following the setting of Bever
and MacWilliams (2013). To clarify, at the start of the 12-month formal model
experiments, the retained Pearl River-derived sediments (Classes 4-5 in Table 1) that
entered the model during the 15-month spin-up period were added as Class 1 and
Class 2, respectively, to avoid contaminating the data analysis of the formal
experiments. This approach allows for a better distinction between Pearl River
sediment and seabed sediment, enabling separate analysis of the suspension, transport,
and deposition of Pearl River-derived sediment (Harris et al., 2008; Zhang et al.,
2019). Specifically, the fractions of the two types of Pearl River-derived sediments
were set at 40% and 60%, respectively, following Zhang et al. (2019) and Zhang et al.
(2021). The parameters for all five sediment classes are summarized in Table 1.
Sediment density, porosity, and erosion rate for all sediment classes were set to 2650
kg m$^{-3}$, 0.672 (Zhang et al., 2019; Zhang et al., 2021), and $1 \times 10^{-4}$ kg m$^{-2}$ s$^{-1}$ (Ralston
et al., 2012), respectively. The settling velocity ($w_s$), critical shear stresses for erosion
($\tau_{ce}$), and other parameters were set following previous studies or were based on
model calibration (Ralston et al., 2012; Warner et al., 2017; Zhang et al., 2019; Dong
et al., 2020; Zhang et al., 2021; Cao et al., 2025).
Our model configuration incorporates seasonal variations in $\tau_{ce}$, supported by
multiple lines of evidence from field observations, laboratory experiments, and
numerical analyses (Dong et al., 2020; Cao et al., 2025). Previous studies have
established a distinct seasonal pattern in the PRE, with winter $\tau_{ce}$ values significantly
exceeding those in summer. Dong et al. (2020)'s laboratory experiments using the
UMCES-Gust Erosion Microcosm System (U-GEMS) on 2017-winter sediment
samples yielded a $\tau_{ce}$ of 0.26 Pa, which effectively reproduced observed suspended
sediment concentration (SSC) in winter simulations. However, this value proved
excessive for summer conditions, when a $\tau_{ce}$ of 0.15 Pa provided better agreement
with field observations in summer simulations, indicating a winter-to-summer $\tau_{ce}$ ratio
of 1.73. Recent 2020-summer in situ measurements by Cao et al. (2025) using a
benthic quadrapod-mounted 3D Profiling Sonar revealed a two-layer erosion
threshold system: a surface "fluffy layer" with $\tau_{ce} = 0.06$ Pa overlying a consolidated
seabed with $\tau_{ce} = 0.13$ Pa. The latter value aligns with Dong et al. (2020)'s summer
calibration, suggesting that Dong et al. (2020)'s laboratory measurements, potentially
affected by sediment consolidation during sample transport, might have missed the
lower $\tau_{ce}$ of the surface fluffy layer. Based on these consistent findings, we
implemented a seasonal $\tau_{ce}$ adjustment factor of 1.73 (winter/summer) in our model
configuration (Table 1).

**Table 1.** CSTM model Sediment Properties

| Source | Seabed | | | Pearl River | |
|---|---|---|---|---|---|
| Class | 1 | 2 | 3 | 4 | 5 |
| Sediment Type | Clay | Silt | Sand | Single grains | Flocs |
| $w_s$ (mm s$^{-1}$) | 0.02[c] | 1.2 | 57[d] | 0.005[c] | 0.6 |
| Summer $\tau_{ce}$ (Pa) | 0.14[e] | 0.03 | 0.27[d] | 0.15[abef] | 0.05[abe] |
| Winter $\tau_{ce}$ (Pa) | 0.24[f] | 0.05[f] | 0.47[df] | 0.26[abf] | 0.09[abf] |
| Fraction | Spatially variable, see Figure 2g-i | | | 40%[ab] | 60%[ab] |

[a]Zhang et al. (2019), [b]Zhang et al. (2021), [c]Calibrated, [d]Warner et al. (2017), [e]Cao et al. (2025), and
[f](Dong et al., 2020).

## 2.5 Wet and dry season regimes

The study area exhibits pronounced seasonal variability, which can be distinctly categorized into two primary seasons (Dong et al., 2004; Su, 2004; Liu et al., 2014; Zhang et al., 2021). This seasonal classification is supported by multiple environmental parameters, including river freshwater discharge, riverine sediment load (Figure 3a), wind patterns (Figure 3b), air temperature (Figure 3c), and modeled wave conditions (Figure 3d-f) at a representative site (21.5°N, 114°E; corresponding to station W1 in Figure S1a, located immediately south of the PRE). The meteorological data for wind and air temperature were obtained from the NCEP reanalysis dataset, while wave parameters were derived from numerical model simulations. These comprehensive indicators collectively characterize the distinct seasonal patterns observed in the study area (Figure 3). The entire year (from April 1,

2017, to March 31, 2018) is typically divided into two main seasons: wet summer
(from April 1, 2017, to September 30, 2017) and dry winter (from October 1, 2017, to
March 31, 2018).
During the wet summer season, freshwater discharge tends to be notably high,
often exceeding 10,000 $m^3 s^{-1}$ and reaching a maximum of 53,000 $m^3 s^{-1}$, with an
average value of 15,266 $m^3 s^{-1}$. This discharge constitutes a significant portion of the
entire year, accounting for 72.06% of the annual total. During this period, the river
carries a substantial sediment load of 32.85 megatons, constituting 95.17% of the total
annual sediment transport. Prevailing winds predominantly blow from the south. For
example, Figure 3b depicts the average monthly wind vector direction during the
summer months as northward, with weak southeasterly winds in April, May, and
September, and moderate southeasterly winds in July. June and August experience
moderate southwesterly winds. The 2 m height air temperatures typically range
between 20℃ and 30℃. The daily average Hsig remains relatively low, with the
monthly average Hsig less than 1 m. The wave propagation direction is generally
consistent with the wind direction, being easterly in April and May, and southerly
from June to September.
In stark contrast, the dry winter season demonstrates markedly lower runoff,
typically falling below 10,000 $m^3 s^{-1}$, with an average value of 5,953 $m^3 s^{-1}$. The
sediment load during this period is significantly reduced to merely 1.67 megatons,
marking a substantial decrease compared to the wet summer season. Prevailing winds
during the dry winter are predominantly northeasterly, with relatively high wind

speeds. Except for moderate wind intensity in March, the monthly average wind speed

in other months exceeds 5 m s$^{-1}$. The 2 m height air temperatures typically range

between 10℃ and 25℃ during this season. The wave propagation direction aligns

with the prevailing northeasterly winds of the season, predominantly northeasterly.

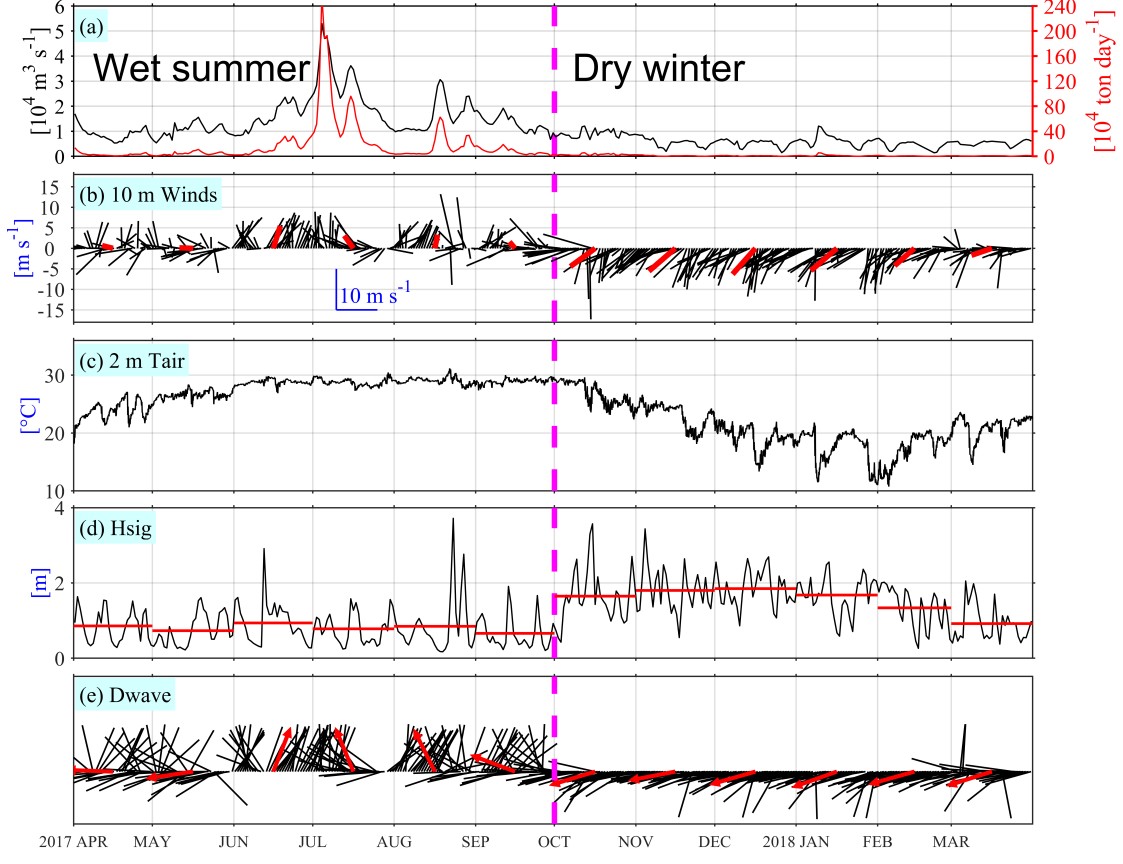

**Figure 3.** Time series of (a) the daily Pearl River freshwater discharge and sediment

load, (b) daily (black vectors) and monthly (red vectors) averaged 10-meter height

winds, (c) hourly 2-meter height air temperature, daily (black) and monthly (red)

averaged (d) significant wave height (Hsig, lines) and (f) wave propagation direction

(vectors) weighted by Hsig$^{2}$. Two distinct seasons are delineated by the dashed

magenta line.

## 2.6 Model experiments

To assess the relative importance of tides, waves, ambient shelf currents and
residual water levels, seasonal variation in critical shear stress for erosion, the settling
velocity, and the spin-up duration of Pearl River-derived sediment (Classes 4-5 in
Table 1) in the transport and dispersal of Pearl River-derived sediments, we conducted
seven simulation experiments (Table 2). In all experiments, we implemented the
Charnock approach within COAWST's bulk air-sea flux parameterization scheme to
calculate surface wind stress using the NCEP 10-m wind product (Charnock, 1955;
Fairall et al., 1996), ensuring consistency in wind stress forcing across all simulations.

Exp 1 (the **Control** run) incorporated all the aforementioned forcing agents
(including winds) and accounted for the seasonal variation in critical shear stress for
erosion, with the winter critical shear stress for erosion set to be 1.73 times of that in
summer. Exp 2 (**NTS** hereafter) was identical to Exp 1 but excluded tides, while Exp
3 (**NWS** hereafter) excluded waves. In Exp 4 (**NAS** hereafter), waves, tides, and the
seasonal variation in critical shear stress for erosion were included, but the shelf
current and residual water levels were omitted (i.e., no subtidal circulation forcing at
open boundaries) to examine the influence of the South China Sea circulation. Exp 5
(NVS hereafter) replicated the setup of Experiment 1, but with one modification: it
used a constant critical shear stress for erosion ($\tau_{ce}$) across both seasons, specifically
adopting the summer $\tau_{ce}$ value from Table 1 throughout the simulation (i.e., no
seasonal adjustment between winter and summer). Exp 6 (**DSV** hereafter) was

identical to Exp 1, except that it set a double sediment settling velocity of the Exp 1.
Finally, to assess the model's sensitivity to the spin-up duration of Pearl River-derived
sediment, particularly regarding the retention of riverine sediments in both the water
column and the seabed, we adopted the sediment distributions (Classes 1 to 5) from
the Control run on March 31, 2018, as the alternative initial conditions for the Cycle
experiment (designated as Exp 7, **Cycle** hereafter). This setup carries over the full
year's evolution of riverine sediment transport and deposition from the Control run
(Exp 1), including changes in all sediment classes, into the start of Exp 7. As a result,
Exp 7 mainly evaluates how the presence of previously deposited riverine sediments
influences subsequent sediment transport estimates.

**Table 2.** Experiment Settings

| Experiments | Tides | Waves | Ambients | $\tau_{ce}$ | $w_s$ | Re-run |
|---|---|---|---|---|---|---|
| Exp 1 (Control) | ✓ | ✓ | ✓ | Variable | Original | ✗ |
| Exp 2 (NTS) | ✗ | ✓ | ✓ | Variable | Original | ✗ |
| Exp 3 (NWS) | ✓ | ✗ | ✓ | Variable | Original | ✗ |
| Exp 4 (NAS) | ✓ | ✓ | ✗ | Variable | Original | ✗ |
| Exp 5 (NVS) | ✓ | ✓ | ✓ | Constant | Original | ✗ |
| Exp 6 (DSV) | ✓ | ✓ | ✓ | Variable | Double | ✗ |
| Exp 7 (Cycle) | ✓ | ✓ | ✓ | Variable | Original | ✓ |

The term 'Ambients' denotes ambient shelf currents and residual water levels. Variable
indicates simulations employing seasonally varying $\tau_{ce}$ values (from Table 1), while
'Constant' refers to runs using exclusively the summer $\tau_{ce}$ value throughout the entire
experiment. 'Original' designates cases utilizing the settling velocities specified in
Table 1, whereas 'Double' indicates simulations with these values doubled."


**3 Results**
**3.1 Seasonal hydrodynamics and transport patterns of the Pearl**
**River-derived sediment**
We quantified the spatial distributions of seasonal mean wind stress, Hsig, wave
bottom orbital velocity (WBOV), and bottom shear stress for both the wet summer
and dry winter periods (as defined in Section 2.5). These distributions serve as
representative hydrodynamic conditions for typical summer and winter scenarios,
respectively (Figure 4).
During summer, the prevailing winds predominantly originate from the south,
with the average wind stress generally below 0.03 Pa, except in the eastern coastal
waters of Hainan Island, where localized values reach up to 0.05 Pa (Figure 4a). In
contrast, during the dry winter season, the prevailing winds shift to a northeasterly
direction, resulting in generally higher average wind stress compared to summer
(Figure 4b), with values typically exceeding 0.1 Pa in areas deeper than 40 m and
surpassing 0.2 Pa in the offshore eastern Guangdong Coast near the Taiwan Bank (see
Figure 1).
Corresponding to the seasonal wind stress (Figures 4a-b), the seasonally averaged
wave characteristics in the PRE and the adjacent northern continental shelf of the
South China Sea exhibit significant seasonal variations (Figures 4c-d).

During the wet summer season, the Hsig in the studied area is relatively low, with

waves predominantly coming from the southeast (Figure 4c). The seasonal average
Hsig across the entire shelf remains below 1 m, with areas deeper than 60 m showing
Hsig values above 0.8 m, while in shallower nearshore regions (water depth < 20 m),
Hsig is less than 0.6 m (Figure 4c). Corresponding to the lower Hsig in the wet
summer, the seasonally-averaged WBOV is also relatively small, generally less than 1
cm s$^{-1}$ in areas deeper than 40 m, except in some nearshore shallow water regions
where it reaches up to 10 cm s$^{-1}$ (Figure 4e). The seasonally-averaged bottom shear
stress during the wet summer is relatively high in the PRE, nearshore regions, and the
Taiwan Bank, where tidal dissipation is strong (Figure 4g).

In the dry winter season, the Hsig increases significantly compared to the wet

summer, with waves primarily coming from the northeast, although refraction occurs
in some nearshore regions, changing the wave direction to southeasterly (Figure 4d).
The area with water depths exceeding 60 m has a Hsig greater than 1.5 m, while in the
20-meter depth region, the Hsig reaches approximately 1 m (Figure 4d). Compared to
the wet summer, the WBOV increases significantly in the PRE mouth and many
nearshore regions, reaching up to 10-20 cm s$^{-1}$ (Figure 4d). The average bottom shear
stress on the continental shelf outside the estuary also increases significantly during
the dry winter compared to the wet summer (Figure 4f).

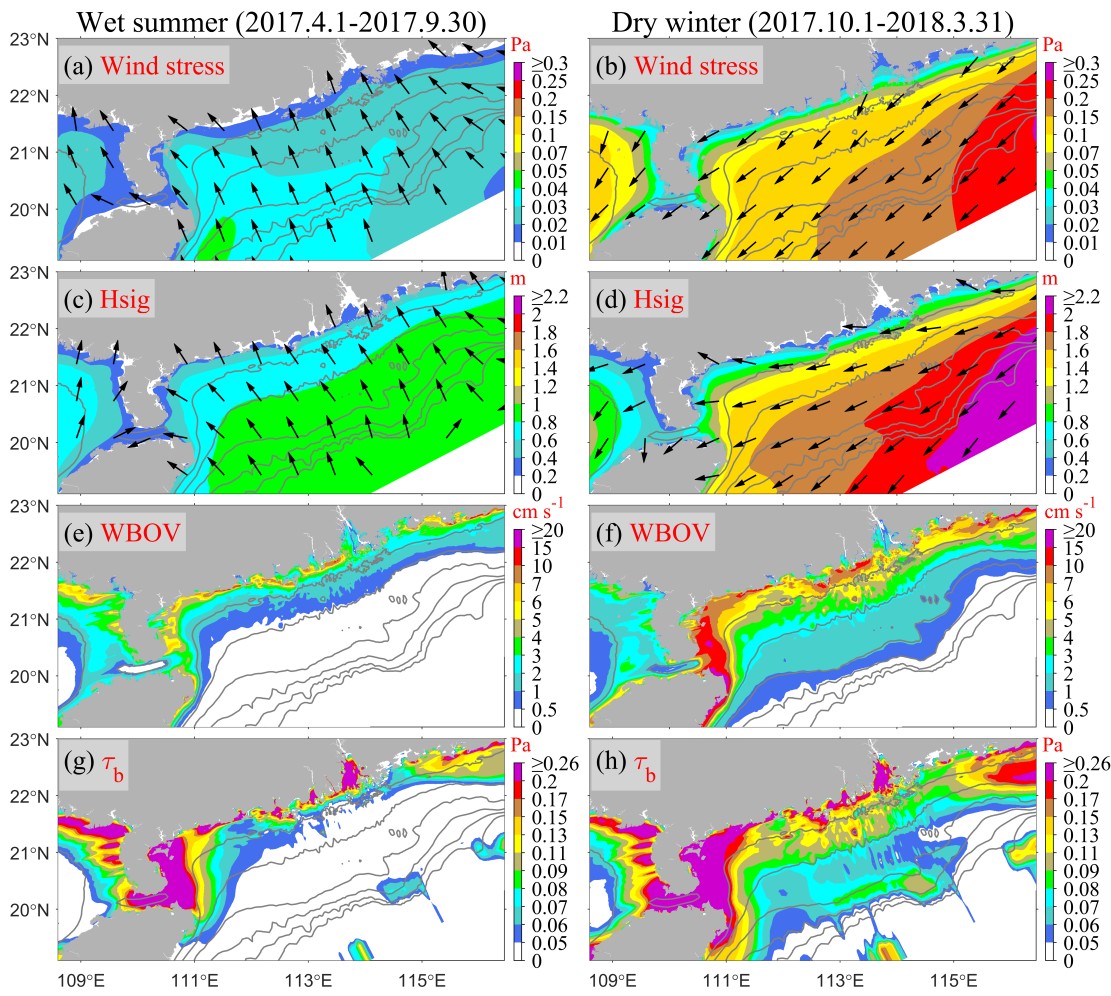

**Figure 4.** The patterns of variables averaged for the wet summer season (April 1st to

September 30th, 2017; Column 1) and the dry winter season (October 1st to March

31st, 2018; Column 2) in the Control case. Row 1 (a-b) depicts wind stress (color) and

direction (vectors), Row 2 (c-d) shows Hsig (color) and propagation direction (vectors)

weighted by Hsig², Row 3 (e-f) presents wave bottom orbital velocity (WBOV), and

Row 4 (g-h) displays bottom shear stress magnitude.

The patterns of residual sediment dispersal, flux, and deposition over the

simulation period provide clear information on the mechanisms for sediment

redistribution on both annual and seasonal timescales. The following section presents

a detailed analysis of the seasonally averaged fields of salinity, flow, riverine SSC,
depth-integrated riverine sediment flux, and riverine sediment deposition patterns
during the wet summer season (Figure 5) and dry winter season (Figure 6) on the
continental shelf.
During the wet summer season, when freshwater discharge is high and water
column stratification is strong, riverine SSC ("riverine" means only Pearl
River-derived sediment, classes 4-5 in Table 1, as follows) is primarily influenced by
advection from the buoyant river plume (salinity less than 33.5 in Figure 3a, as
follows) into the shelf sea, primarily in the surface layer (Figures 5a-b), high SSC
regions closely align with the buoyant plume, as sediment is efficiently transported by
the low-salinity, high-momentum freshwater outflow (Figures 5a-d). The buoyant
plume extends both northeastward and southwestward along the coastline (Figure 5a).
Due to the influence of southerly winds (Figure 4a) and ambient shelf currents, the
extent of the buoyant plume extending northeastward is significantly higher than that
extending southwestward. In terms of riverine sediment suspension, its estuarine
turbidity maxima (ETM) zone (~100 mg $L^{-1}$) is situated in the shallow water area
within the estuary (water depth < 10 m) (Figure 5c-d). Beyond the estuary, suspended
riverine sediment disperses across the shelf through the buoyant plume. Further away
from the estuary, its distribution aligns with that of the buoyant plume, with
concentrations diminishing as dispersal distance increases. The depth-integrated
advective horizontal flux (without including vertical processes such as settling,
resuspension, or diffusion, which are handled separately within the model) of riverine
sediment offers a clear indication of the primary net transport pathway of the riverine
sediment (Figure 5e). The riverine sediment exhibits both southwestward and
northeastward fluxes (Figure 5e). Southwestward coastal transport can extend as far
as the Leizhou Peninsula and Hainan Island. On the eastern side, the northeastward
transport extends toward the Taiwan Bank. However, the primary transport pathway
there is diverted southward (Figure 5e) due to the obstruction caused by summer
upwelling near the Guangdong east coast (Chen et al., 2017a; Chen et al., 2017b), as
evidenced by the cross-shore current in the bottom layer (Figure 5b). The
southwestward transport pathway follows the region where the water depth is
shallower than 30 m, with a riverine sediment flux of 10–20 g m$^{-1}$ s$^{-1}$. In contrast, the
northeastward transport pathway occurs in the 30–60 m depth range, but the riverine
sediment flux is below 10 g m$^{-1}$ s$^{-1}$. Throughout the wet summer season, substantial
amounts of riverine sediment are deposited near the estuary (Figure 5d), particularly
leading to notably high deposition of riverine sediment near the river mouth (> 100
mm). Outside the estuary, the thickness of riverine sediment is comparatively lower,
but it can reach approximately ~0.5 mm during the wet summer season in certain
areas off the western Guangdong coast.

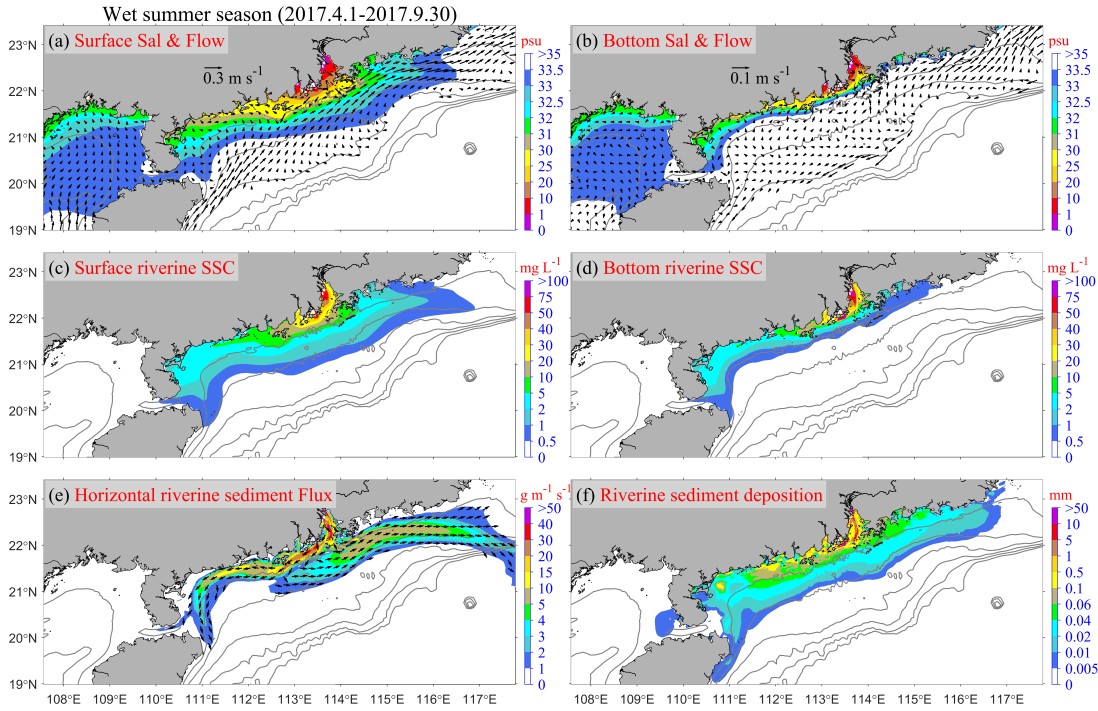

**Figure 5.** Patterns averaged over the entire wet summer season in the Control case: (a) surface and (b) bottom salinity (color, psu) and flow (arrows, m s$^{-1}$); (c) surface and (d) bottom riverine (classes 4 and 5 in Table 1, as follows) SSC (mg L$^{-1}$); (e) depth-integrated horizontal riverine sediment transport rate (color, g m$^{-1}$ s$^{-1}$) and direction (arrows); and (f) riverine sediment deposition thickness (mm) on the seabed during the wet summer season. Flow vectors in regions with water depths exceeding 100 m are masked for clarity.

In contrast, during winter, when river discharge is low and vertical mixing is more intense, the correlation between the buoyant plume and riverine SSC is much weaker, and the riverine SSC is largely governed by resuspension processes driven by strong northeasterly winds and waves, rather than by freshwater transport. The

expansion of the Pearl River buoyant plume is constrained to the southwestward
direction by strong northeasterly winds (Figure 6a), resulting in a narrow cross-shore
width of the buoyant plume and the formation of a strong horizontal salinity gradient
(i.e., a salinity front, particularly within the 30–33.5 psu range shown in Figure 6a)
outside the estuary (Figure 6a). Flow velocity increases near this salinity front,
facilitating the westward extension of the buoyant plume through the Qiongzhou
Strait into the "Gulf" region. The riverine SSC is significantly lower than in the wet
summer: in the ETM zone inside the PRE, riverine SSC falls from roughly 100 mg L$^{-1}$
in summer to about 10 mg L$^{-1}$, while on the offshore shelf, it decreases from
approximately 5 mg L$^{-1}$ to around 2 mg L$^{-1}$ (Figures 6c-d vs. 5c-d). During the dry
winter, following the coastal current, the riverine suspended sediment primarily
moves southwestward along the coast, deflecting southward along the topography
near the Leizhou Peninsula (Figure 6c). It then bifurcates near the east entrance of the
Qiongzhou Strait, with one branch continuing into the "Gulf" region, and the other
one proceeding southward along the east coast of Hainan Island. Stronger winds and
waves in the dry winter lead to the resuspension of a considerable amount of riverine
sediments, originally deposited in "Proximal", "Western", and "Eastern" regions
during summer. The resuspended sediments are then transported to coastal bays as
well as to the sides and rears of the islands (Figure 6d). Additionally, a portion of the
riverine sediment transported to the "Gulf" region gets deposited on the seabed during
the dry winter season.

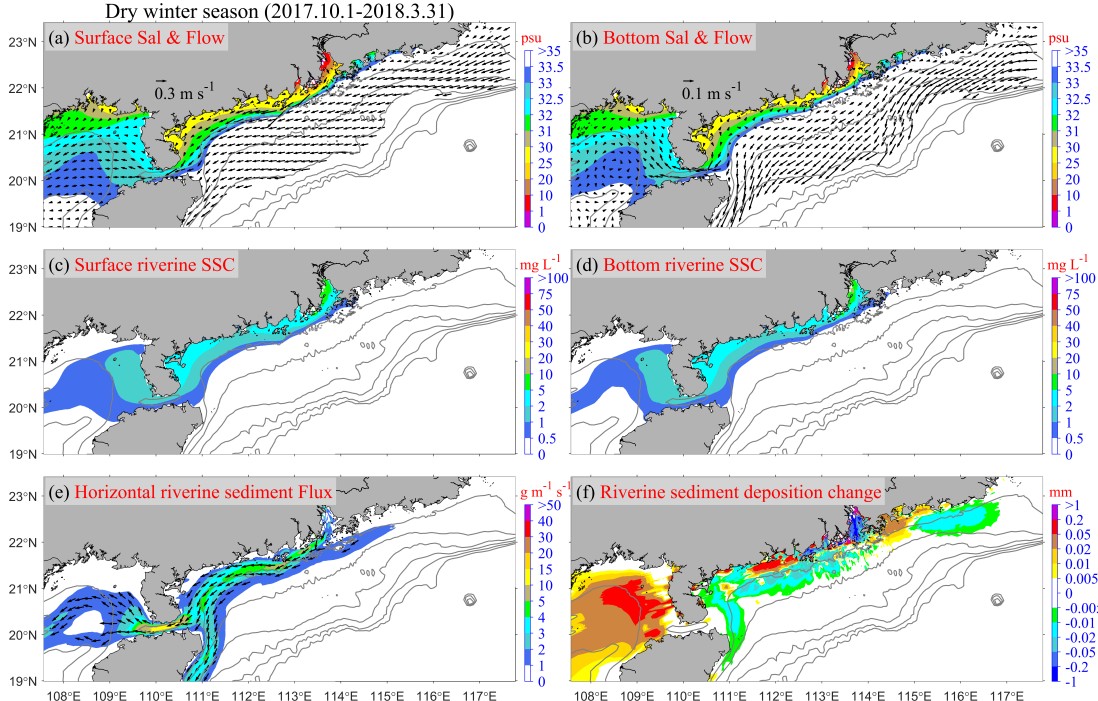

**Figure 6.** Same as Figure 5, but for the dry winter season in the Control case. Notably, (f) illustrates the changes in riverine sediment (classes 4 and 5 in Table 1) deposition on the seabed at the end of the dry winter season compared to the end of the wet summer season.

## 3.2 Riverine sediment budgets and annual deposition over the shelf

We present the sediment fluxes and retention amounts in different regions. Figure 7a-c illustrates the proportion of riverine sediment retention budget within each region, expressed as a percentage of the total annual river sediment input, for the wet summer season, the dry winter season, and the entire year under the Control run case, respectively. Meanwhile, Figure 7d illustrates the annual deposition over the shelf.

The retention of Pearl River sediment on the continental shelf exhibits significant

seasonal variations (Figures 7a-c). During summer (characterized by high discharge
and low wind/waves), the PRE and continental shelf receive 95.17% of the annual
sediment load from the Pearl River (Figures 3a and 7a). Approximately two-thirds of
this sediment is retained in the "Proximal" region (Figure 7a). Additionally, influenced
by the prevailing southerly winds and northeastward shelf circulation, 13.01% of the
annual sediment load is retained in the "Eastern" and "Southeastern" regions (Figure
7a). Meanwhile, the shelf west of the PRE (⑤−⑧ regions) retains 15.87% of the
annual load, with the "Western" region being the primary receiver, accounting for 8.48%
(Figure 7a). Only 0.92% and 2.3% of the annual load enter the "Gulf" and "Distal"
regions, respectively, during summer (Figure 7a). The "Southern" region retains a
mere 1.22% of the sediment (Figure 7a). In winter (characterized by low discharge
and energetic winds/waves), the PRE and the continental shelf receive only 4.83% of
the annual sediment load (Figures 3a and 7b). The sediment distribution during this
season is primarily a result of the dynamic reworking of the sediments of summer
deposition (Figure 7b). While the "Proximal" region continues to receive sediment,
with a 1.38% increase in retention, the ②−⑥ regions experience a decrease in
sediment retention. This sediment is predominantly transported and retained in the
more distant "Gulf" and "Distal" regions (Figure 7b). The annual sediment budget
reveals that 66.45% of the Pearl River sediment is retained in the "Proximal" region
(Figure 7c). Additionally, 9.2% is retained in the "Eastern" and "Southeastern" regions
(Figure 7c), primarily during summer (Figures 7a vs. 7c), while 24.12% is retained on
the shelf west of the PRE (⑤−⑧ regions), with the majority in the "Gulf" and
"Distal" regions, mainly during winter (Figures 7b vs. 7c).
The annual deposition thickness of the Pearl River sediments, as illustrated in
Figure 7d, reveals significant deposition within the "Proximal" region, with many
areas exceeding 10 mm despite winter resuspension and transport. Additionally,
deposition on the "Eastern" region reached a magnitude of 0.1 mm, while the western
shelf of the PRE ("Western" and "Gulf" regions) exhibited significantly greater
accumulation. For instance, the deposition west of the Chuanshan Islands reached a
magnitude of 0.5 mm. In the "Gulf" region, deposition was primarily concentrated in
the northeastern part, extending southwestward along the 30-60 m isobaths.
Sediments transported southwestward along the east coast of Hainan Island and
retained in the "Distal" regions did not predominantly settle on the seabed due to the
greater water depth but remained largely suspended in the water column.

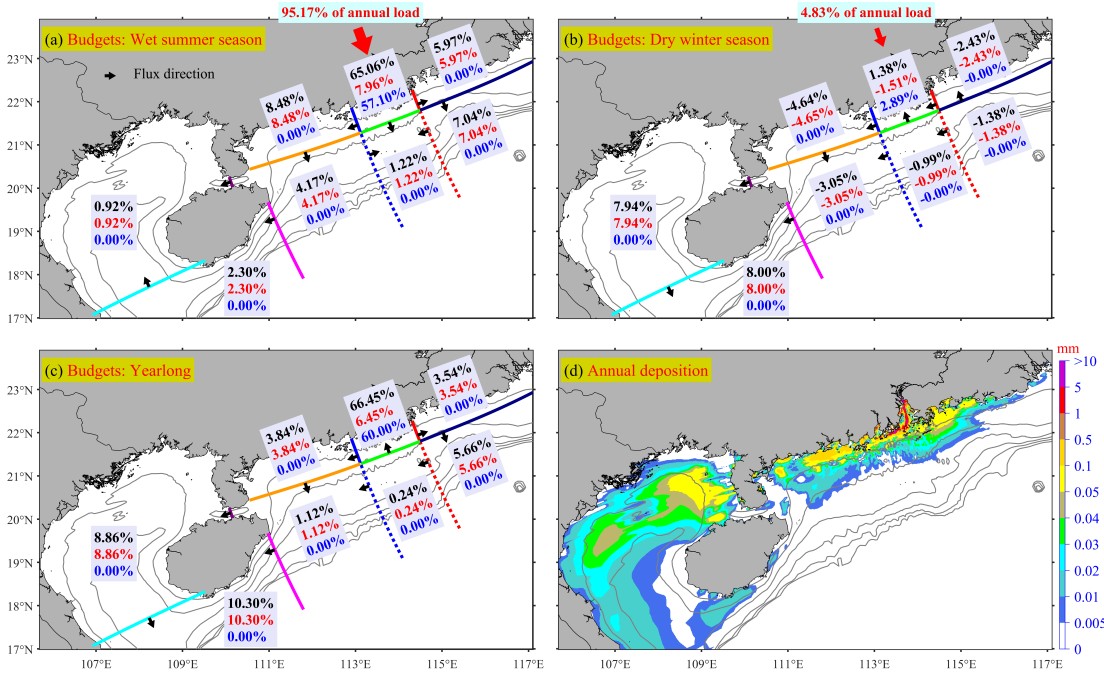


**Figure 7.** Riverine sediment (classes 4 and 5 in Table 1) retention budget percentages
at eight regions (see Figure 1) during (a) the wet summer season, (b) the dry winter
season, and (c) the entire year in the Control run case. (d) the annual deposition
patterns spanning from April 1st, 2017, to March 31st, 2018 in the Control Run. All
percentages displayed in the figure are relative to the annual riverine sediment load
(see Figure 3a). The black percentage values represent the combined total of riverine
sediment Class 4 and Class 5, while the red and blue values denote sediment Class 4
and Class 5, respectively. Arrows indicate the direction of net riverine sediment flux
at each transect during the specified period.

**3.3 Model sensitivity experiments: relative roles of physical processes,**
**sediment properties, and spin-up durations**

Six sensitivity simulations, namely Exp 2-7 (NTS, NWS, NAS, NVS, DSV, and

Cycle), were conducted (Table 2). As the latter three experiments do not impact
hydrodynamics, we focus on presenting the seasonal mean differences in bottom shear
stress between the Control run and the first three cases (NTS-Control, NWS-Control,
NAS-Control) for both summer and winter (Figure 8).

In the NTS case, bottom shear stress is reduced relative to the Control run by a

similar amount in both summer and winter due to the minimal seasonal variation in
tidal intensity. This reduction primarily occurs in the PRE, around the Taiwan Bank,
and near the Leizhou Peninsula (Figures 8a-b). In contrast, in the NWS case, the
reduction in bottom shear stress is greater in winter than in summer, reflecting the
intense seasonal variability of wind and wave activities (Figures 3b, 3d-e, and 4a-f).
Unlike the NTS case, the NWS-induced decrease occurs mainly in the nearshore areas
outside the PRE, although similar declines are also found around the Taiwan Bank and
along the eastern side of the Leizhou Peninsula (Figures 8c-d). For the NAS case, the
impact on bottom stress is minimal compared to the NTS and NWS cases. The effect
is almost negligible on the inner shelf at depths less than 100 m, with widespread
impacts generally below 0.02 Pa. Some pronounced deviations are noted in localized
deeper areas near the southern boundary of the domain (Figures 8e-f). These
deviations, likely arising from boundary condition effects, are situated far from the
Pearl River-derived sediment distribution areas (Figures 5-6). Consequently, they do
not influence the dynamics of the Pearl River-derived sediment transport over the
continental shelf (Figures 8e-f).

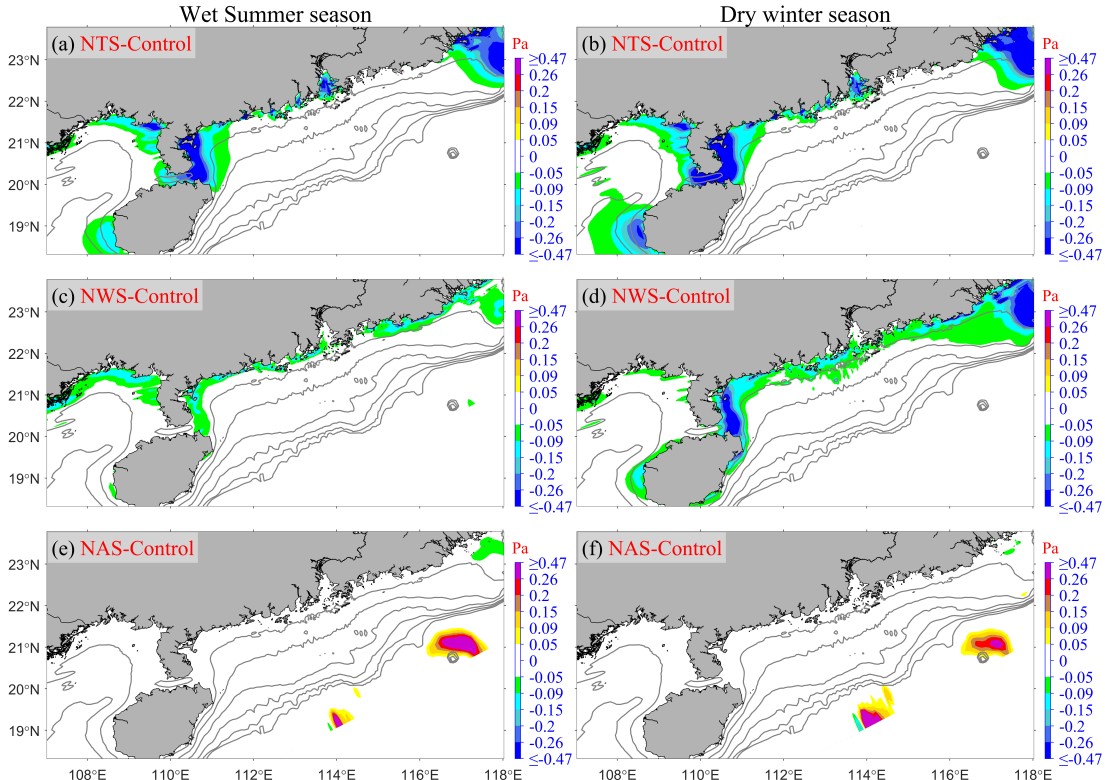


**Figure 8.** The seasonal mean differences in bottom shear stress between the Control
run and the following cases: (a-b) NTS (NTS minus Control), (c-d) NWS (NWS
minus Control), and (e-f) NAS (NAS minus Control). The first column represents the
wet summer season, while the second column corresponds to the dry winter season.

Then, we analyzed seasonal riverine sediment transport and deposition patterns
("riverine" means only the Pearl River-derived sediment, classes 4-5 in Table 1, as
follows) by comparing the control run with six sensitivity experiments (NTS-Control,
NWS-Control, NAS-Control, NVS-Control, DSV-Control, and Cycle-Control)
(Figures 9-11). The study focuses on the Pearl River-derived sediment dynamics,
indicated by surface circulation and riverine SSC distribution patterns (Figures 5 and
6). Specifically, Figures 9 and 10 present seasonal surface currents and SSC
differences between control and sensitivity runs, complemented by deposition pattern
differences in Figure 11.
Compared to the Control case, the NTS case demonstrates that while tides
significantly enhance bottom stress (Figures 8a-b) but have minimal impact on the
mean circulation (Figures 5a, 6a, and 9a-b), and their exclusion reduces bottom shear
stress by over 0.2 Pa in the PRE and near the Leizhou Peninsula. Consequently,
increased deposition of the Pearl River-derived sediments occurs inside the PRE, its
adjacent areas, and on both sides of the Leizhou Peninsula (Figure 11a). During
summer, riverine SSC notably decreases in the ①−⑥ regions (Figure 9a). This
reduction pattern persists in winter, particularly in the PRE and on both sides of the
Leizhou Peninsula (Figure 9b).
Like the NTS, NWS has a relatively minor impact on circulation (Figures 5a, 6a,
and 9c-d). However, NWS leads to more Pearl River-derived sediment being
deposited in the nearshores of "Western" and "Eastern" regions (Figure 11b).
Consequently, the riverine SSC in summer is much lower in the downstream of the
PRE and in ②−⑥ regions (Figure 9c). This similar reduction pattern persists in the
winter, but is slightly in more western regions (Figure 9d).
For the NAS case, the impact on bottom stress is minimal compared to the NTS
and NWS cases. However, NAS has a relatively large impact on the mean circulation
(Figures 5a, 6a, and 9e-f). It mainly influences the summer circulation. Specifically,
ignoring these factors would cause the relatively strong northeastward flow along the
Guangdong coast to become very weak (Figure 9e). When it comes to winter, the
influence of NAS on circulation is relatively small. That is, in the absence of the
background residual water level and residual circulation, due to the strong
northeasterly winds in winter, the overall circulation is still southwestward (Figure 9f).
The decreased northeastward flow in summer leads to the Riverine SSC being
scarcely transported to the vicinity of the "Eastern" and "Southeastern" regions.
Consequently, the Riverine SSC there is decreased (Figure 9e) and sediment
deposition is significantly reduced (Figure 11c). Most of the suspended Riverine
sediment is transported southwestward, resulting in an increase in the Riverine SSC
along the "Western" region. In winter, since most of the suspended Riverine sediment
has been transported southwestward in summer, the Riverine SSC decreases
compared to the Control run (Figure 9f). Ultimately, NAS mainly causes a significant
reduction in sediment deposition in the "Eastern" region, while sediment deposition
increases in the "Gulf" and the "Distal" regions (Figure 11c).

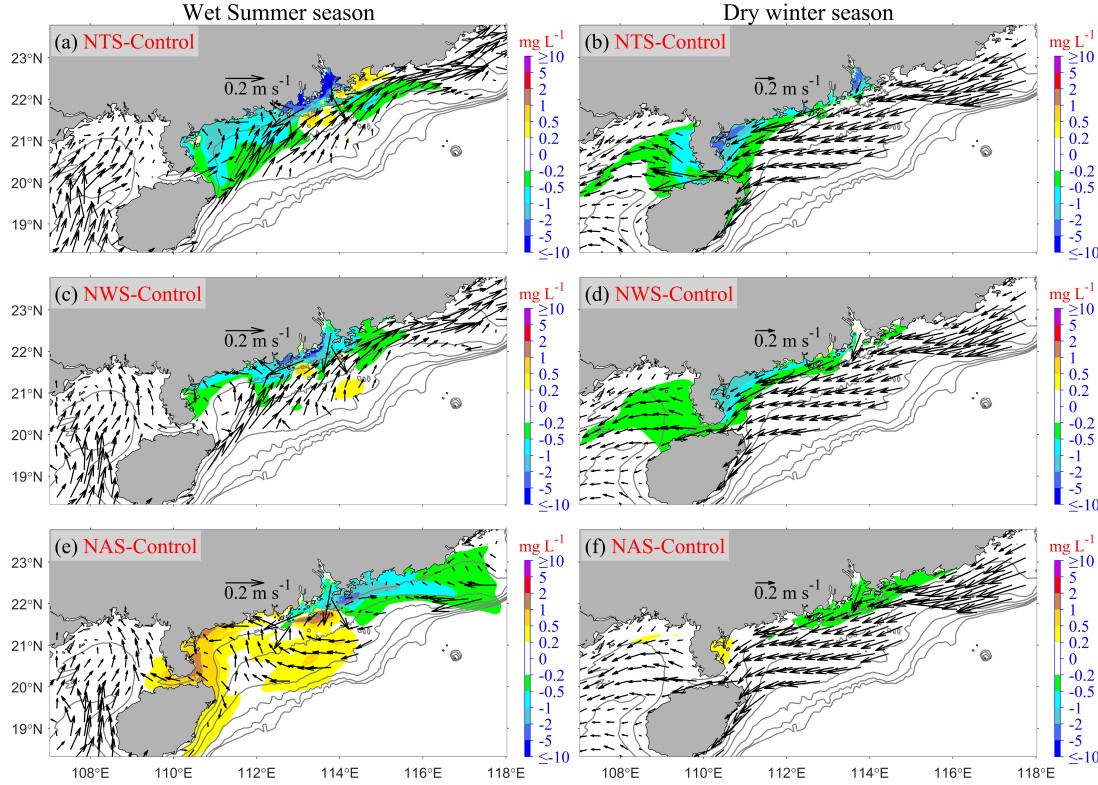



**Figure 9.** Same as Figure 8, but for seasonal mean differences in surface riverine SSC

between the Control run and the following cases: (a-b) NTS-Control, (c-d)
NWS-Control, and (e-f) NAS-Control. The first column represents the wet summer
season, while the second column corresponds to the dry winter season. Vectors show
the seasonal mean surface current fields in each experiment. Note that the riverine
SSC values shown in the figure correspond to classes 4 and 5 as defined in Table 1.

For the NVS case, the summer conditions of NVS are precisely the same as those
of the Control run (Figure 10a). Since the critical shear stress for erosion in winter is
lower than that in the Control run, this leads to an increase in re-suspension in the
"Proximal", "Western" and the "Gulf" regions, increasing Riverine SSC (Figure 10b).
Eventually, this causes a reduction in the deposition thickness of the Pearl
River-derived sediments in these regions(Figure 11d).

In the DSV case, significant reductions occur in the primarily high SSC areas in

the Control run in both summer and winter (Figures 5c, 6c, 10c-d). The enhanced
settling velocity results in an increased deposition of Pearl River-derived sediments
along the Guangdong coastline ("Western" and "Eastern" regions) and the eastern
"Gulf" region, accompanied by a reduced deposition thickness in the western "Gulf"
region (Figure 11e).

In the Cycle case, the new riverine sediment input and its transport processes

during the Cycle experiment are nearly identical to those in the Control run. Therefore,
compared to the Control run, the Cycle experiment specifically focuses on examining
the impact of the presence of pre-existing Pearl River-derived sediments on estimating
the riverine SSC and the annual seabed riverine sediment budget in the second year.
Consequently, during the summer period, the Cycle case experiences elevated riverine
SSC in the primary depocenters identified in the Control run (Figures 7d and 10e),
while this effect is diminished by winter (Figure 10f). Figure 11f thus captures the
transport trends of pre-existing riverine sediments in the second year, demonstrating
that riverine sediments deposited during the first year can be resuspended and
transported further southwestward during the second year. This migration is driven by
the annually averaged net alongshore coastal current, which remains predominantly
directed toward the southwest. The current becomes stronger during the winter
monsoon under the influence of prevailing northeasterly winds, whereas the opposing
summer southerly winds are comparatively weaker, indicating a persistent
southwestward sediment transport trend over multi-year timescales.

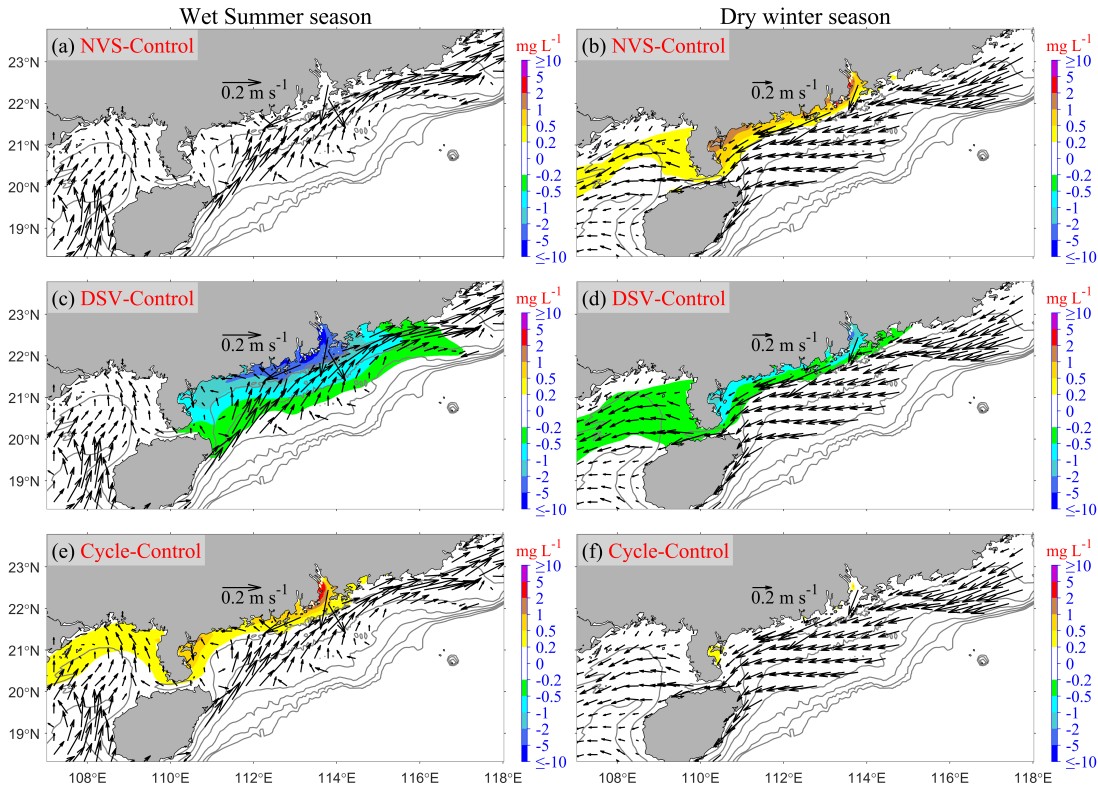


**Figure 10.** Same as Figure 9, but for the latter three experiments (NVS, DSV, Cycle).
(a-b) NVS-Control, (c-d) DSV-Control, and (e-f) Cycle-Control.

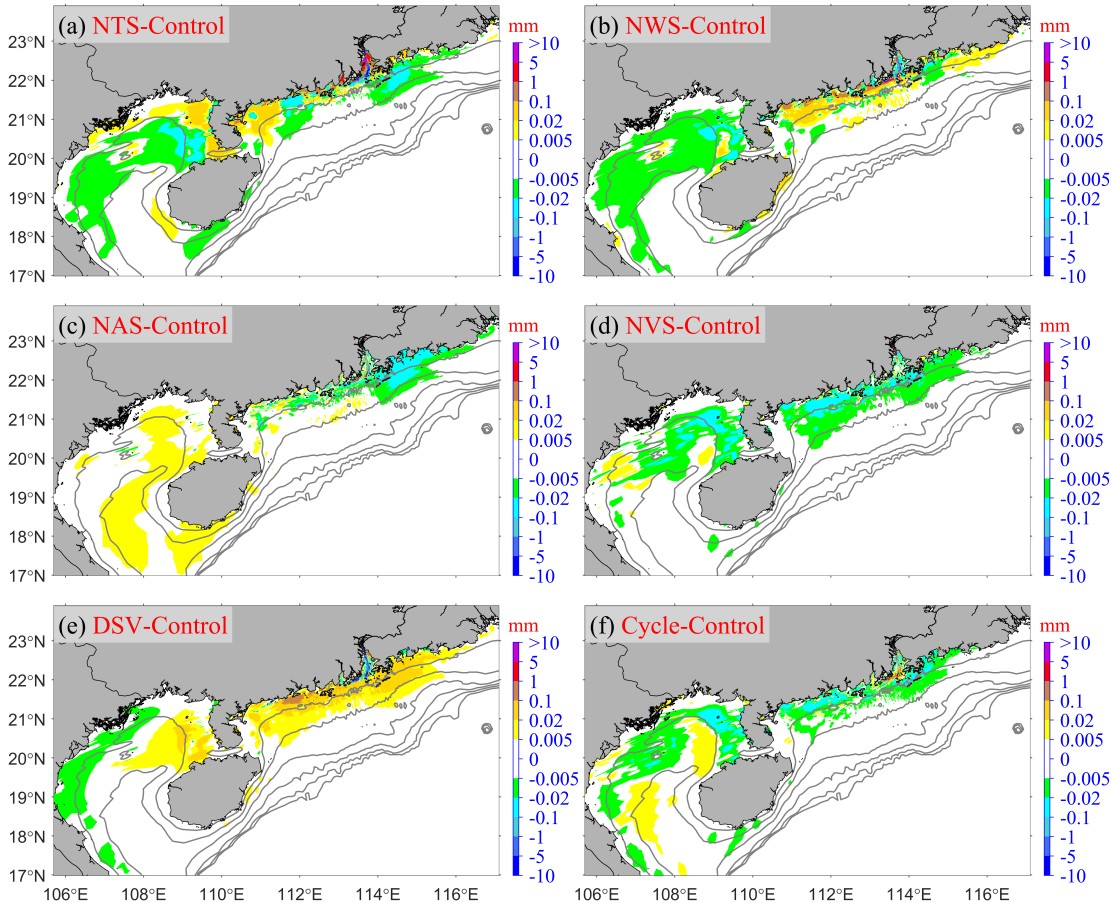

**Figure 11.** The differences in annual deposition patterns of Pearl River-derived sediment (classes 4-5, Table 1) on the seabed between the Control run and the following cases: (a) NTS-Control, (b) NWS-Control, (c) NAS-Control, (d) NVS-Control, (e) DSV-Control, and (f) Cycle-Control.

## 3.4 Modeled regional retention budgets in sensitivity experiments

Finally, we analyze the impact of various factors on the annual riverine sediment retention budget across different regions. Specifically, Figure 12 illustrates the annual riverine sediment retention budget in various regions under six sensitivity simulations, namely Exp 2-7 (including NTS, NWS, NAS, NVS, DSV, and Cycle). It should be

noted that the retention percentages budget and their variations discussed hereinafter
are all relative to the annual riverine sediment load (Figure 3a).

As shown in Figure 12, tides and sediment settling velocity have the most

significant impact on the retention in the "Proximal" region. In the NTS case and the
DSV case, the retention in the "Proximal" region is 70.92% and 71.57%, respectively
(Figures 12a and 12e), which is higher than 66.45% in the Control run (Figure 7c).
This indicates that ignoring tides will cause the "Proximal" region to capture more
riverine sediments, and a larger settling velocity will result in more riverine sediments
being retained within the "Proximal" region. In these two cases, compared with the
Control run, the retention in the "Gulf" and "Distal" regions decreases. Meanwhile,
the DSV case causes the greatest increase in retention in the "Western" region, with an
increase of +1.91%.

Furthermore, the NWS also leads to a 2.2% increase in retention in the

"Proximal" region (Figure 12b), which is lower than that in the NTS case. This shows
that tides dominate resuspension versus deposition in the "Proximal" region more than
waves do. However, for the "Western" region, compared with the NTS case, the NWS
causes a greater increase in retention, indicating that waves dominate the resuspension
of Pearl River-derived sediments in these nearshore areas more than tides do.

For the "Eastern" and "Southeastern" regions, NAS brings about the most

dramatic changes, the retention of Pearl River-derived sediments in these regions
drops from 9.1% to 0.84% compared to the Control run (Figure 12c). Meanwhile,
ignoring these background circulations results in a substantial increase in the retention
in the "Distal" region, with an increase of 6.49%.

The NVS case leads to a decrease in the retention of the Pearl River-derived

sediments in ①−⑥ regions compared to the Control run. The reduction ranges from
-0.05% to -0.85% (Figure 12d), which in turn causes the retention in the "Gulf" and
"Distal" regions to increase by 0.7% and 1.47%, respectively. Overall, compared with
scenarios that ignore physical processes and alter sediment settling velocity (NTS,
NWS, NAS, and DSV), the NVS scenario has a relatively smaller impact on the
retention of the Pearl River-derived sediments.

Finally, in the Cycle case, to isolate the pre-existing Pearl River-derived

sediments, the initial retentions (the end conditions of the Control run on March 31,
2018) were subtracted before calculating the retention in the Cycle case (Figure 12f).
The retention of Pearl River-derived sediments in ①−⑥ regions shows little
variation, with values ranging from -0.98% to +0.24% compared to the Control run
(Figure 12f). The most significant changes are the decreases and increases in retention
in the "Gulf" and "Distal" regions, which are -2.17% and +3.54%, respectively. This
demonstrates the long-term trend of southwestward transport of Pearl River-derived
sediments on the shelf (relative to the Control run).

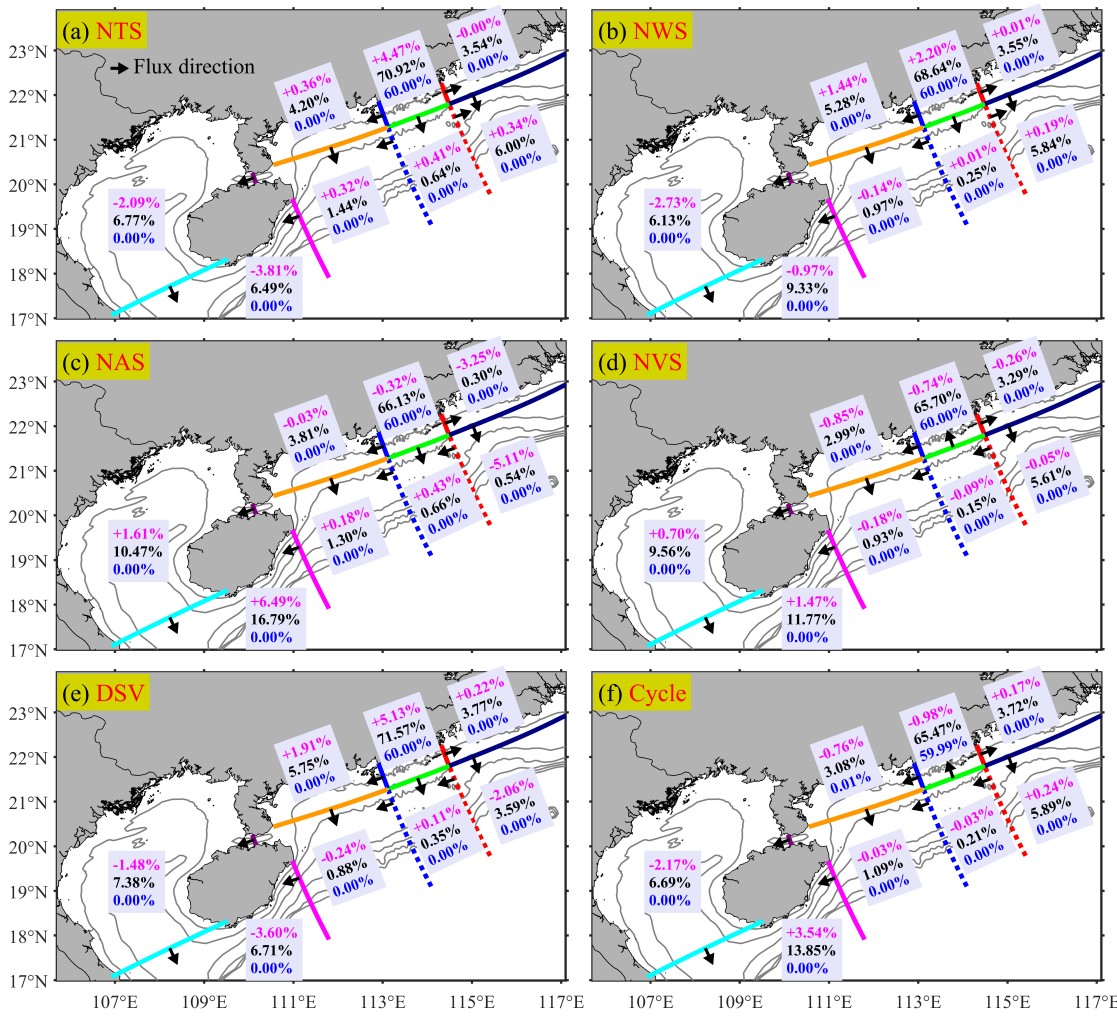


**Figure 12.** Same as Figure 7c, but for the other six cases: (a) NTS, (b) NWS, (c) NAS, (d) NVS, (e) DSV, and (f) Cycle, respectively. All percentages shown in the figure are expressed relative to the annual riverine sediment load (see Figure 3a). Magenta values denote the differences in retention percentage of riverine sediments (Classes 4 and 5; Table 1) between the Control run and each sensitivity case. Black values represent the combined retention of Classes 4 + 5, while blue values indicate Class 5 alone. To obtain the retention percentage for Class 4, simply subtract the Class 5 percentage (blue) from the combined Classes 4 + 5 percentage (black).

## 4. Discussions

### 4.1 Fidelity of our model results

We simulated the suspension, transport, and deposition of the Pearl River-derived sediment over the shelf from April 2017 to March 2018. Comparisons with multiple types of observations demonstrated that the model simulation reasonably well captured the regional patterns and temporal variability of water levels (Figures S2-S4), surface waves (Figures S5-S6), estuarine and shelf currents (Figures S8a-b and S9-S10), salinity and temperature (Figures S7a-b and S8c-d), and SSC (Figures S7c and S8e-f). Even though we have made substantial efforts to collect observational data and conduct calibration and validation, the large scope of the study area and the fact that the study covers an entire year mean that there are some inherent challenges to achieving a complete analysis. The real-world situation may be extremely complex, and these validations may still not be sufficient to address all issues (such as the accurate parameterization of sediment characteristics and their seasonal variations, as well as the proportion of slow-settling fine grains and flocculated flocs in riverine sediments). Therefore, in this section, we intend to discuss the fidelity of our results.

Studies have demonstrated a seasonal dependence of sediment critical shear stress (Xu et al., 2014; Briggs et al., 2015). On the Louisiana shelf, seabed erodibility is controlled by grain size, sediment age, proximity to river sources, bioturbation, and flood deposits, and is higher during the wet season than the dry season (Xu et al., 2014). It is also linked to seasonal hypoxia: sites experiencing hypoxic (hypoxia greater than 75% of the time, hypoxia defined as $< 2$ mg $O_2$ $L^{-1}$) conditions exhibit

greater erodibility, whereas the normoxic (hypoxia less than 25% of the time) site
shows the lowest erodibility (Briggs et al., 2015). These findings imply that sediment
models should incorporate seasonal variations in critical shear stress, parameterized
using field observations or seasonal sampling data. Similarly, in the PRE, field
observations, laboratory experiments, and numerical sensitivity analyses have shown
that the critical shear stress for erosion of sediments in the PRE is higher in winter/dry
season than in summer/wet season (Dong et al., 2020; Cao et al., 2025). The summer
period in the PRE also coincides with seasonal hypoxia (Cui et al., 2018; Cui et al.,
2022), which likely contributes to this seasonal variation in erodibility. Therefore, we
considered the seasonal variation of critical shear stress for erosion in a suite of model
experiments, that is, increasing the critical shear stress for erosion in winter. This
setting was also applied during the 15-month spin-up before conducting the model
experiments (the driving force of the spin-up is realistic, i.e., the settings are
consistent with those of the Control run). After realistic reworking during the spin-up,
the bed sediment grain size distribution (used as initial conditions in the Control run
and all Sensitivity cases except the Cycle case) is quite close to the initial prototype
(Figures 2d-f vs. 2g-i).

When the seasonal variation of critical shear stress for erosion is omitted (NVS

case), the model results suggest a reduced retention of Pearl River-derived sediments
in ①−⑥ regions during winter, alongside an increased retention in the "Gulf" and
"Distal" regions. However, these differences are relatively small in magnitude
compared to the annual load (Figure 12d). Thus, accounting for seasonal variations in
critical shear stress for erosion has a limited influence on the annual-scale retention
patterns. The dispersal distance of fluvial sediments on continental shelves is strongly
influenced by settling velocity (Harris et al., 2008). For example, Apennine-derived
sediments, characterized by lower settling velocities, travel farther before deposition
than Po River sediments, which are predominantly flocculated and settle more rapidly
(Fox et al., 2004; Harris et al., 2008). Likewise, our results suggest that selecting an
appropriate settling velocity parameter exerts a greater control on sediment dispersal
patterns than accounting for seasonal variations in critical shear stress for erosion
(Figure 12d vs. 12e). The results of the DSV case show that a sediment settling
velocity twice that of the Control run leads to the highest retention in the "Proximal"
and "Western" regions across all experiments (Figure 12e), while reducing the
retention in the "Distal" region (Figure 12e). Although the settling velocity we
adopted is based on previous studies (Xia et al., 2004) and model calibrations (Figures
S7c and S8e-f), with due consideration given to the presence of slow-settling single
fine grains and high-settling flocs in riverine sediments, certain discrepancies might
still exist in this setting. These discrepancies are contingent upon the actual magnitude
of the low settling velocity of fine grains. In almost all cases, all flocs are retained in
the "Proximal" region (Figure 12a-e), and only in the Cycle case, flocs accounting for
0.01% of the annual load are retained in the "Western" region west of the "Proximal"
region (Figure 12f, blue values), indicating that high-settling flocs hardly leave the
"Proximal" region. This finding shows close alignment with, yet exhibits minor
distinctions from, the observed patterns in the Mekong Shelf (Xue et al., 2012). Xue
et al. (2012) found that while the preponderance of flocs is deposited on the Mekong
delta front precisely at the river mouth, a quantity equivalent to 1.6% of the annual
riverine sediment load of flocs is deposited on the downdrift delta front further
downstream from the river mouth. This is mainly because the estuarine bay of the
PRE is wider and there are numerous islands near the river mouth. The overwhelming
majority of flocs are either deposited within the estuarine bay or captured by the
surrounding islands. In conclusion, our results are affected by the settling velocity of
fine grains. More field observations and studies on model parameterization regarding
the settling velocity of fine grains are urgently needed.
As previously noted, we classified riverine sediments into two categories based
on established research: 40% slow-settling fine grains and 60% fast-settling flocs.
This 40%/60% distribution is consistent with the setting from earlier studies (Zhang et
al., 2019; Zhang et al., 2021), as summarized in Table 1. While such assumptions are
necessary for modeling purposes, the actual composition of riverine sediments in
natural environments remains uncertain. To evaluate the sensitivity of our results to
this uncertainty, we conducted a conceptual analysis. If all riverine sediments were
hypothetically composed entirely of fast-settling flocs, they would be completely
retained near the source, with no transport to the "Gulf" region. However, this
scenario is inconsistent with the radionuclide measurements obtained from "Gulf"
region surface sediment samples (Lin et al., 2020). On the other hand, if all sediments
were considered slow-settling fine grains, only 16.13% would be retained in
"Proximal" region under normal conditions (or 28.9% in the DSV case), a result that
diverges significantly from established research. Chen et al. (2023) analyzed
high-resolution seismic data and demonstrated that approximately 35% of the Pearl
River-derived sediment has been transported to offshore shelf areas over the past
6,500 years, suggesting that 65% was deposited proximally. Our findings are in close
agreement, indicating that 66.45% of the Pearl River sediments are retained in the
proximal region, while 33.55% are transported elsewhere. This consistency with Chen
et al. (2023) supports the validity of our approach. Taken together, these analyses
confirm that the 40%/60% fraction assumption is a reasonable approximation for
modeling purposes.

Furthermore, our model results demonstrate reasonable reliability in other aspects.

Liu et al. (2009) and Ge et al. (2014) using chirp sonar profiles from the inner shelf of
the South China Sea combined with Zong et al. (2009)'s onshore borehole data, found
that the thickness of Pearl River-derived sediments within the PRE since the Holocene
is over 20 m, while the mud thickness in the shallow waters west of the Chuanshan
Islands (see Figure 1) is approximately 5-10 m. Our calculated annual sediment
thicknesses for these two regions are approximately 2 mm and 0.3 mm (Figure 7d),
respectively. Given our model's annual riverine sediment load of 34.52 megatons,
which has been significantly reduced due to recent human activities (Dai et al., 2008),
compared to the widely accepted Holocene average of around 90 megatons (Liu et al.,
2009), we estimate the total sediment thickness over the past 7500 years to be roughly
39 meters and 6 meters for these depositional zones, consistent with previous studies
(Liu et al., 2009; Zong et al., 2009).
Furthermore, our results reveal that 8.86% of the riverine sediment derived from
the Pearl River is transported to the "Gulf" region (Figure 7c), primarily during the
winter season (Figure 7b). This finding is not only consistent with the earlier
speculation proposed by Ge et al. (2014) but also supplements the conclusions drawn
by Lin et al. (2020). From a hydrodynamic perspective, Shi et al. (2002) found that
the net flux of currents in the Qiongzhou Strait is westward throughout the year. Our
results for both wet summer (Figures 5a-b) and dry winter currents (Figures 6a-b) in
the Qiongzhou Strait are consistent with Shi et al. (2002). This westward flow
contributes to the westward transport of Pearl River sediment to the "Gulf" region.
**4.2 Implications of our model results**
The fate of sediment dispersed from the river into the coastal ocean involves at
least four processes: supply via buoyant plumes; initial deposition; resuspension and
transport by marine processes; and long-term net accumulation (Wright and Nittrouer,
1995). In general, a significant proportion of river sediment tends to deposit in the
estuary and its vicinity (Walsh and Nittrouer, 2009; Hanebuth et al., 2015).
Walsh and Nittrouer (2009) present a hierarchical decision tree designed to
predict the marine dispersal system at a river mouth based on fundamental
oceanographic and morphological characteristics. Within this framework, riverine
sediment deposition is characterized using key factors, including riverine sediment
discharge (greater or less than 2 megatons), shelf width (greater or less than 12 km),
and wave and tidal range conditions (greater or less than 2 m) (Walsh and Nittrouer,

2009).

We aim to analyze our PRE simulation results using the framework established
by Walsh and Nittrouer (2009). Although the Pearl River's riverine sediment
discharge (Figure 2a) exceeds the Walsh and Nittrouer (2009)'s 2 megatons per year
threshold, most of the sediment still remains deposited near the estuary (Figure 7c),
indicating an estuarine accumulation-dominated (EAD) system, unlike the
hierarchical decision tree proposed by Walsh and Nittrouer (2009). Outside the estuary,
the continental shelf, spanning 200-220 km in width (Liu et al., 2014), significantly
exceeds Walsh and Nittrouer (2009)'s 12 km threshold. As a result, most escaped
riverine sediments tend to accumulate on the shelf rather than being captured by
submarine canyons (Figures 7c-d). This wide, shallow shelf promotes sediment
deposition and limits the direct transport of fine sediments into deeper waters (Walsh
and Nittrouer, 2009). Subsequently, given that the annual mean tidal range (Chen et
al., 2016; Gong et al., 2018b) and Hsig (Figure 3d) near the PRE are both below the 2
m threshold established by Walsh and Nittrouer (2009), the majority of escaped
riverine sediments are predominantly deposited in the proximal depo-center. Our
findings demonstrate that most of our outcomes are consistent with the hierarchical
decision tree proposed by Walsh and Nittrouer (2009), except for the application of
the 2 megatons per year threshold for riverine sediment discharge. This phenomenon
can primarily be attributed to the unique geomorphological characteristics of the PRE,
including its broad mouth (Figures 1 and S1), extensive accommodation space
encompassing approximately 2385 km$^2$ of water area (Wu et al., 2018), the presence
and sheltering effect of numerous adjacent islands (Li et al., 2024b), and the division
of fluvial sediment discharge through eight distinct outlets (Hu et al., 2011).
The monsoonal nature of the northern SCS (Figures 4a-b) induces pronounced
seasonal variations in Pearl River-derived sediment transport and deposition (Figures
5 and 6). During the summer wet season, the Pearl River delivers approximately 95.17%
of its annual sediment load to the PRE and the adjacent shelf (Figure 7a) under
relatively calm wind and wave conditions (first column of Figure 4), leading to
predominant proximal deposition (Figure 5f). In contrast, the winter dry season is
characterized by strong northeasterly monsoon winds that generate high-energy waves
(second column of Figure 4), significantly increasing bottom shear stress (Figures
4g-h). This process resuspends previously deposited sediments and facilitates their
redistribution, particularly toward regions such as the "Gulf" region (Figure 6f).
The PRE exhibits distinctive geomorphological features, yet dispersal of its
fine-grained sediment transport on the continental shelf conforms to general patterns
observed offshore of other monsoon-influenced estuarine systems. Similar
multiple-stage sediment delivery and dispersal mechanisms have been documented
offshore of various major estuaries and their adjacent shelves, including the Yellow
River Shelf (Bian et al., 2013; Zeng et al., 2015), Changjiang River Shelf (Zeng et al.,
2015), and Mekong River Shelf (Xue et al., 2012; Eidam et al., 2017), demonstrating
comparable sedimentary processes under monsoon climatic influences. In these
systems, sediment transport is not confined to a single process but rather occurs in
stages, influenced by seasonal variations in hydrodynamic conditions. Like the PRE
Shelf, the Mekong Shelf experiences distinct phases of sediment deposition, with fine
sediments being delivered during periods of high river discharge and then
redistributed by waves and tidal forces, particularly during monsoonal shifts (Xue et
al., 2012; Eidam et al., 2017). These complex patterns highlight the interplay between
riverine inputs, coastal morphology, and oceanographic processes in shaping sediment
dynamics.
These sediment delivery patterns have implications beyond sediment fate,
particularly for carbon cycling. Sediment deposition in coastal and shelf areas plays a
significant role in trapping organic carbon, influencing long-term carbon burial rates
(LaRowe et al., 2020). Sediment dynamics directly influence the fate of organic
carbon (OC) in marine environments, where sediments function as both a sink and a
source of OC, playing a pivotal role in global carbon cycling (Repasch et al., 2021).
The multiple-step transport mechanisms can lead to varying carbon storage locations,
affecting the sequestration potential of these systems. Additionally, resuspension and
redistribution of sediments, especially during high-energy events, may expose
previously buried organic material, leading to carbon remineralization and influencing
coastal nutrient cycles and ecosystem health (Ståhlberg et al., 2006; Moriarty et al.,
2018). Therefore, understanding these patterns is crucial for assessing the broader
impacts on carbon cycling and coastal biogeochemical processes.
**4.3 Limitations and future work**
This study focuses on analyzing simulation results from a typical year,
encompassing both wet and dry seasons from 2017 to 2018, to understand the
seasonal variations and annual patterns of suspension, transport, and deposition of
sediment in the PRE and adjacent shelf. However, it's essential to recognize that the
long-term sediment transport and deposition dynamics in the Pearl River are
influenced by numerous complex factors. These include changes in sea level and
coastal line (Church and White, 2006; Harff et al., 2010; Hong et al., 2020; Lin et al.,
2022; Ma et al., 2023), alterations in wind field and precipitation (Ning and Qian,
2009; Young et al., 2011), natural sedimentation within the Pearl River Delta (Wu et
al., 2010), modifications in sediment load and underwater volume of the estuary
caused by anthropogenic impact (Wu et al., 2014; Wu et al., 2018; Lin et al., 2022),
interannual variations of the shelf circulations (Liu et al., 2020; Deng et al., 2022) and
Kuroshio intrusions (Caruso et al., 2006; Nan et al., 2015; Sun et al., 2020). Therefore,
while this study sheds light on seasonal and annual timescale patterns, it cannot fully
represent the short or long-term transport and deposition trends of the Pearl River
sediment. Yet for many shelf systems, a lot of sediment transport happens during
short-lived events such as hurricanes (Xu et al., 2016; Warner et al., 2017; Georgiou et
al., 2024). Consideration of the episodicity of transport would be helpful for future
studies (Xu et al., 2016; Warner et al., 2017; Georgiou et al., 2024).
Additionally, it's important to note that this article primarily focuses on the fate of
the Pearl River sediment on the inner shelf. However, within the expansion range of
the Pearl River buoyant plume, a number of smaller rivers, including the Jiulong
River, Han River, Moyang River, Jian River, Nanliu River, Changhua River and
Nandu River, also contribute freshwater and sediment to the northern South China Sea
(Milliman and Farnsworth, 2011; Zhang et al., 2012; Liu et al., 2016). Although these

rivers contribute significantly less freshwater and sediment compared to the Pearl River, they still impact seawater salinity, suspended sediment concentration, and seabed geomorphology (Liu et al., 2016; Wang et al., 2023; Zong et al., 2024). Since the 1950s, South China delivers approximately 102 Mt/year of fluvial sediment to the SCS, with the Pearl River alone accounting for 84.3 Mt/year—about 83% of the total sediment load (Milliman and Farnsworth, 2011; Zhang et al., 2012; Liu et al., 2016). The specific contributions of each river are detailed in Table 3. While the data highlight the Pearl River's dominant role in sediment delivery, a comprehensive understanding of sedimentary processes and impacts in the northern South China Sea also requires systematic investigation into the roles of the smaller contributing rivers.

**Table 3.** Annual mean runoff and annual suspended sediment load of major rivers in South China that flow directly into the northern South China Sea since the 1950s (Milliman and Farnsworth, 2011; Zhang et al., 2012; Liu et al., 2016).

| River name | Runoff ($m^3 s^{-1}$) | Suspended sediment load (Mt/year) |
| --- | --- | --- |
| Pearl River | 9075 | 84.3 |
| Jiulong River | 476 | 3.1 |
| Han River | 825 | 10 |
| Moyang River | 269 | 0.8 |
| Jian River | 174 | 1.5 |
| Nanliu River | 162 | 1.1 |
| Changhua River | 120 | 0.08 |

| | | |
|---|---|---|
| Nandu River | 179 | 0.4 |


Then, while the model used in this study has shown good validation results,
conducting more sensitivity experiments on sediment parameters, such as settling
velocity and critical shear stress for erosion, would be beneficial. Settling velocity can
influence the location of sediment depocenters, with higher settling velocities leading
to more proximal entrapment and vice versa (Harris et al., 2008). Similarly, critical
erosion stress can affect the resuspension of deposited sediment, with higher critical
erosion stress resulting in less resuspension and more deposition especially during
neap tides and weak wind wave events (Dong et al., 2020; Choi et al., 2023).
Conducting such sensitivity analyses would enhance our understanding of sediment
dynamics in estuaries and shelves. Besides, the model does not account for cohesive
processes, such as consolidation and flocculation, which can significantly impact
sediment behavior (Sherwood et al., 2018). Our model does not incorporate wave and
current-supported gravity flows, which are important factors influencing sediment
transport in submarine canyon areas (Harris et al., 2005; Ma et al., 2010; Zhang et al.,
2020). Since our study area primarily focuses on the continental shelf and the
simulated results indicate that sediment transport occurs mainly in the shallow inner
shelf, where canyons are relatively rare, this omission has a relatively minor impact
on our results.
Lastly, we employ the COAWST model, which uses an S-coordinate system in
the vertical direction with increased resolution near the surface and bottom layers
(Song and Haidvogel, 1994). This vertical layering allows cell heights to vary,
enabling finer resolution in dynamically important regions and improving
performance in areas with sloping bathymetry compared to traditional
sigma-coordinate systems (Bryan, 1969; Song and Haidvogel, 1994). In addition, our
model includes horizontal grid refinement in the PRE, enhancing its ability to resolve
estuarine features. As a result, the model effectively captures estuarine turbidity
maxima (ETM) and horizontal salinity fronts (Figures S11 and S12, see Supplement).
During summer, multiple ETMs appear near the estuary bottom (Figure S11b), and
while these features persist in winter, their concentrations vary (Figure S12b),
consistent with the findings of Wang et al. (2018), Zhan et al. (2019), Zhang et al.
(2021), Ma et al. (2022), and Ma et al. (2024). Horizontal salinity fronts are also well
represented, showing an upstream shift from the high-discharge summer season to the
low-discharge winter season (Figures S11e–f and S12e–f), in agreement with previous
studies by Zhang et al. (2021) and Ma et al. (2024). Nonetheless, compared with the
S-coordinate system, models that employ vertically adaptive layering (e.g., SCHISM,
the Semi-implicit Cross-scale Hydroscience Integrated System Model, Zhang et al.
(2016)) or Cartesian vertical coordinates (e.g., MITgcm, the MIT General Circulation
Model, Marshall et al. (1997a); Marshall et al. (1997b)) generally perform better in
regions with steep topographic gradients (Bijvelds, 2001). Therefore, future research
could benefit from adopting models with higher horizontal resolution and Cartesian
vertical coordinates to improve the simulation of Pearl River-derived sediment
dynamics across the estuary and the adjacent shelf.

**5. Conclusions**


This study utilizes the COAWST model to quantitatively analyze the seasonal
suspension, transport, and annual fate of Pearl River-derived sediment (classes 4-5 in
Table 1) on the continental shelf over a typical year, capturing key marine variables
such as water level, wave height, flow velocity, salinity, temperature, and SSC.
The monsoonal nature of the northern SCS (Figures 4a-b) induces pronounced
seasonal variations in Pearl River-derived sediment transport and deposition (Figures
5 and 6). During the wet summer, calm conditions foster initial Pearl River-derived
sediment deposition via the buoyant plume (Figures 5 and 7a). Conversely, winter's
stronger winds and waves resuspend and transport Pearl River-derived sediments into
"Gulf" region (Figures 6 and 7b). Our quantitative assessment reveals distinct spatial
patterns in the annual fate of riverine sediments: approximately two-thirds of the Pearl
River-derived sediment is retained within the estuarine vicinity ("Proximal" region),
while about 9% reaches the continental shelf east of the PRE ("Eastern" and
"Southeastern" regions), while similar proportions are transported to and retained in
"Gulf" and "Distal" regions, respectively (Figure 7c). Furthermore, we evaluated the
contributions of different physical processes by comparing the Control run with the
reduced-physics sensitivity experiments. Our analysis reveals distinct roles of tidal
forces, wave action, and background circulation in governing the transport and
deposition of Pearl River-derived sediments (Figures 8, 9, 11a-c, and 12a-c).
Tidal dynamics primarily govern sediment behavior within and offshore of the
PRE. Relative to the Control run, neglecting tides leads to lower bottom shear stress
in the PRE (Figures 8a-b), higher Pearl River-derived sediment deposition in the PRE
(Figure 11a), and reduced sediment retention in "Gulf" and "Distal" regions (Figure
12a). Wave activity primarily controls Pearl River-derived sediment resuspension in
three critical dimensions: (1) the river mouth, (2) the nearshore of "Eastern" and
"Western" regions outside the estuary, and (3) periods characterized by high wave
energy during winter (Figures 9c-d). Excluding waves results in increased modeled
Pearl River-derived sediment deposition in these areas and times (Figure 11b), while
reducing accumulation in the "Gulf" region (Figure 11b). Background circulation is
most influential in summer (Figure 5a vs. Figure 9e), with a strong northeastward
current transporting Pearl River-derived sediments to regions ③−④. Without this
current and under weak southerly winds, only 0.84% of Pearl River-derived sediments
are transported to regions ③−④, while deposition in "Gulf" and "Distal" regions is
greater compared to the Control run (Figure 12c). The sediment model solutions are
also highly sensitive to the parameterization of sediment characteristics and spin-up
durations (riverine or seabed sediments) (Figures 10, 11d-f, and 12d-f). Neglecting the
seasonal increase in critical shear stress for erosion leads to enhanced resuspension
and erosion in regions ①−⑥ during winter (Figure 10b), which in turn shifts more
sediment toward "Gulf" and "Distal" regions compared to the Control run (Figure
12d). Increasing the settling velocity reduces the overall riverine SSC (Figures 10c-d)
and results in a spatial redistribution pattern characterized by greater retention in ①−
③ and ⑤ regions and reduced riverine sediment presence in "Gulf" and "Distal"
regions relative to the Control simulation (Figure 12e). Additionally, the modeled
riverine SSC is influenced by pre-existing Pearl River–derived sediments, as shown in
the Cycle experiment (Figure 10e-f). The experiment highlights the effect of riverine
sediment spin-up, showing that first-year retained Pearl River–derived sediments are
predominantly redistributed from the "Gulf" region toward the more distant "Distal"
region during the second year. In contrast, the model exhibits only minor sensitivity to
the duration of seabed sediment spin-up, as demonstrated in the Cycle2 experiment
(see Supplement), in which riverine sediments present during the spin-up period were
added to seabed sediment classes 1–2 at the start of the Cycle2.

## Acknowledgments

This research was funded by the National Natural Science Foundation of China
(grant numbers 42306015 and 42276169), the China Postdoctoral Science Foundation
(grant number 2023M743988). Wenping Gong is supported by the Southern Marine
Science and Engineering Guangdong Laboratory (Zhuhai) (SML2023SP238). The
authors would like to thank the crew of the R/V Changhe Ocean for their valuable
contribution during the collection of the field data. We express our gratitude to the
Co-editors-in-chief, Dr. Mario Hoppema, and three anonymous reviewers for their
valuable suggestions in enhancing and improving this article.

## Data availability

The HYbrid Coordinate Ocean Model (HYCOM) outputs are from:
http://hycom.org/hycom. The NCEP Climate Forecast System Version 2 (CFSv2)
reanalysis data can be obtained at the following website:

1184 https://rda.ucar.edu/datasets/ds094.1/dataaccess/. The NOAA WAVEWATCH III

1185 global ocean wave model output fields can be downloaded from:

1186 ftp://polar.ncep.noaa.gov/pub/history/waves. Hourly water-level data observed at

1187 Quarry Bay station are provided by the Hong Kong Observatory website:

1188 https://www.hko.gov.hk/sc/tide/marine/realtide.htm?s=QUB&t=TABLE. Hourly

1189 water-level data from Zhapo and Qinglan stations, provided by the Flanders Marine

1190 Institute (VLIZ), are part of the UNESCO/IOC Global Sea Level Observing System

1191 (GLOSS) and accessible at http://www.ioc-sealevelmonitoring.org. The mooring data

1192 for the M1 and M2 stations are sourced from Liu et al. (2023) and Li et al. (2024a).

1193 The figure data and model configuration files used in this paper can be downloaded

1194 from: https://doi.org/10.5281/zenodo.15013448.

**Declaration of Competing interests**

1196  The authors declare that they have no known competing financial interests or

1197 personal relationships that could have appeared to influence the work reported in this

1198 paper.

**CRediT authorship contribution statement**

1200  **GZ:** Conceptualization, Numerical modeling, Validation, Data visualization,

1201 Writing-original draft, and Funding acquisition. **SH:** Writing-review & editing,

1202 Validation. **XY:** Writing-review & editing. **HZ:** Writing-review & editing. **WG:**

1203 Writing-review & editing, and Funding acquisition

**Supplement:**

1205  The Supplement includes validation and analysis of the model's water levels,

Hsig, flow velocities, salinity, temperature, and SSC. It provides additional text and
figures that support the model validation and supplementary analyses, which could
not be fully presented in the main article due to space limitations.

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
