# Peer review of "Modeling dispersal of the Pearl River-derived sediment over the Shelf"

_EGUsphere, 2024_

## Author Response (AR1)

**Response to Reviewer 1**

Dear Reviewer:

We greatly appreciate the time you spent reviewing our manuscript and providing constructive comments and minor revisions. We have made revisions in response to your suggestions.

Comment:

This study investigated the wind and wave effects on the dispersal of the Pearl River-derived sediment using a coupled numerical model. The results demonstrated the crucial role of wind in the westward along-shelf transport of the riverine sediments and the formation of the mud belt along the coasts. The diagnostics of cross-shelf current and suspended sediment concentration revealed the mechanism of winds in the formation of along-shelf currents, vertical mixing, and resuspension of sediment. This work is valuable for understanding the transport processes of the Pearl River sediment and the manuscript is well written. Therefore, I recommend publication of the manuscript after minor revision.

Response:

We very appreciate your comments for our manuscript. These comments and insights are helpful for improving the quality of our paper. We address your comments point-by-point as follows. We thank you for your constructive comments.

Comment:

Were winds and waves included om the ramp simulation from Jan.1, 2016 to Apr. 1,

 The initial bed sediment distribution would be different for the two case simulations. Which case Figure 2 represents for?

Response:

Thank you for your comment. Yes, winds and waves were incorporated into the ramp simulation from January 1, 2016, to April 1, 2017. This period was selected to ensure that the model reached dynamic equilibrium before the main simulation phase. In total, we conducted seven cases: one Control case and six Sensitivity cases. Figure 2 presents the spatial distribution of seabed sediment fractions at the end of the spin-up phase in the Control run on April 1, 2017. This simulation accounts for winds, tides, waves, ambient shelf currents, and seasonal variations in critical erosion stress (i.e., increased critical erosion stress during winter). The same initial bed sediment distribution was applied to all seven cases. We have clarified this in the revised manuscript to ensure transparency and eliminate any ambiguity.

Comment:

The critical shear stress chosen for clay is large. Although clay is cohesive, the large critical shear stress would restrict the transport of clay on the shelf and the riverine clay actually represents riverine sand. Comparing the initial prototype of bottom sediment with that after realistic reworking, the changes in slit component is the most significant because of the low critical shear stress.

Response:

Thank you for your valuable suggestion regarding the critical shear stress for sediment. In our initial setup, the critical shear stress was determined based on

previous literature and model calibration. However, we acknowledge that this approach may have limitations, particularly in capturing seasonal variations in sediment properties. Observations, laboratory experiments, and numerical sensitivity analyses from prior studies have demonstrated that the critical shear stress of sediments in the Pearl River Estuary is higher in winter than in summer.

In response to your insightful comment, we have conducted additional numerical experiments incorporating seasonal variations in critical shear stress. The results reveal that the bed sediment grain size distribution after realistic reworking is now significantly closer to the initial prototype compared to our previous simulations, which did not account for such temporal variability. This improvement underscores the importance of considering seasonal changes in sediment properties for accurately modeling sediment dynamics. Specifically, the revised results highlight that the silt component, characterized by a lower critical shear stress, undergoes the most significant changes, aligning more closely with observed patterns.

To further address this issue, we have revised our study and designed seven numerical cases. Six of these cases incorporate seasonal variations in critical erosion shear stress, while one case excludes this variation to analyze its impact on the ultimate fate of riverine sediments in the Pearl River Estuary. These updates have been integrated into the manuscript, along with a detailed discussion of their implications. Your feedback has greatly enhanced the robustness of our study, and we sincerely appreciate your contribution.

Comment:

Line 374-376. The westward extension of Peral River plume is more significant than the eastward extension. Also see the salinity distribution in Fig. 5a.

Response:

Thank you for your comment. The primary reason for this is that our previous analysis focused on the vertically averaged salinity field. In the revised manuscript, we now present both surface and bottom salinity fields. The surface salinity field more clearly highlights the significant eastward expansion of the plume. We have made the necessary modifications to the manuscript accordingly.

Comment:

Figure 3. The source and location of wind, air temperature and wave measurements need to be presented.

Response:

Thank you for your suggestion. The wind and air temperature data are sourced from NCEP, while the wave measurements are not based on observations but are derived from model results. We have clarified this in the manuscript.

Comment:

Line 614-618 and Figure 8. The eastward transport of sediment is significant when winds and waves were not imposed in the model. It should be aware that the wind-driven northeastward shelf currents can be introduced from the open boundaries that were extracted from HYCOM that contains winds. Because the riverine sediment input is large in wet season and there is no wind/wave induced resuspension in dry season, the yearly averaged sediment flux has a strong northeastward component.

Response:

Thank you for your invaluable feedback, which has significantly strengthened our study. In response to your comments, we have conducted seven additional numerical simulations to systematically evaluate the relative contributions of tides, waves, ambient shelf currents, seasonal variations in critical erosion stress, settling velocity, and the initial bed grain size distribution to the transport and dispersal of Pearl River-derived sediments. These simulations include one Control case and six Sensitivity cases. In our initial simulations, we inadvertently overlooked that the HYCOM-derived flow fields already account for the influence of monsoon-driven currents, South China Sea Warm Current (SCSWC), and Kuroshio intrusions on shelf circulation. This includes the strong northeastward currents in summer and the weaker northeastward or southwestward currents in winter. To address this, we have refined our approach in the NAS case (i.e., no ambient shelf currents) by excluding HYCOM-derived water levels and velocities. This adjustment allows for a clearer assessment of the influence of South China Sea (SCS) circulation while eliminating potential inaccuracies arising from monsoon-driven currents and Kuroshio intrusions. As a result, our model now provides a more accurate representation of the underlying physical processes. We believe these revisions enhance the robustness of our findings, and we sincerely appreciate your insightful suggestions.

**Response to Reviewer 2**

RC2: 'Comment on egusphere-2024-3289', Anonymous Referee #2, 22 Jan 2025

Summary and General Comments:

Dear Reviewer:

We greatly appreciate the time you spent reviewing our manuscript and providing constructive comments. We have made revisions in response to your suggestions. Now we have conducted seven numerical simulations to systematically evaluate the relative contributions of tides, waves, ambient shelf currents, seasonal variations in critical erosion stress, settling velocity, and the initial bed grain size distribution to the transport and dispersal of Pearl River-derived sediments. These simulations include one Control case and six Sensitivity cases.

Comment:

I do not feel that the paper in its current form meets the standard expected for publication in Ocean Science.   My general reasons are (1) some of the analysis seem poorly justified and detract from the main findings; (2) the organization should be improved to better communicate major findings; (3) the skill assessment has geographic and temporal limitations; and (3) the findings are not generalized to other systems. Despite these shortcomings, the paper has potential. The model quantifies suspended sediment dispersal on the shelf off the Pearl River Estuary (PRE) and compares dispersal during summer (wet season) and winter (energetic season). The authors explore the roles of winds and waves by comparing "Case 1" (includes winds and waves) to "Case 2" (neglects winds and waves); but the presentation of this could

Response:

We sincerely thank you for dedicating your valuable time and sharing your professional insights during the review of our manuscript. Your thoughtful comments have been instrumental in improving the quality of our paper and advancing its contribution to the academic community. Below, we provide a detailed, point-by-point response to your suggestions. We are deeply grateful for your constructive and insightful comments.

Comment:

Below are general points that if addressed would strengthen the paper.

Most sections of the Results include both "Case 1" and "Case 2", and consideration of the wind/wave effects are interspersed with the seasonal variations and annual averages. For example, Figure 4g,h (section 4.1) has the impact of winds and waves. Then, Section 4.3 shows the annual averages for both Cases 1 and 2. The analyses would be clearer if the comparisons between Cases 1 and 2 were in one section.

Response:

Thank you for your insightful suggestion. We agree that organizing the comparisons between cases within a single section will enhance clarity and readability. In response, we have restructured the Results section to more clearly distinguish between seasonal variations, annual averages, and the Sensitivity cases.

Comment:

Discussion Section 5.1 does not seem to contribute to the main conclusions, and does

not seem to yield general knowledge for river-influenced shelves. My suggestion is to

remove or shorten this section unless it is better justified.

Response:

Thank you for your valuable suggestion. We have removed Section 5.1, as its content was not closely aligned with the main focus of this study and did not directly support our key findings. We agree that this section lacked significant insights for river-influenced shelves, and its removal enhances the clarity and focus of our discussion.

Comment:

Hydrodynamics: This section estimates idealized geostrophic velocities (Figure 11) and thermal wind shears (Figure 12). These analyses were unclear, and they complicate the paper and do not seem well justified. The numerical model solved the full momentum equations which would include the geostrophic and the thermal wind dynamics. What is gained from these analyses that are useful for the PRE and extendable to other locations?

Response:

Thank you for your valuable feedback. We acknowledge that the analyses of idealized geostrophic velocities and thermal wind shears may have complicated the manuscript and lacked clear justification. In response to your suggestion, we have removed this section from the revised manuscript to improve clarity and focus on the key findings. We appreciate your insightful comments, which have helped us to refine our study.

Comment:

Sediment Transport Analysis: Equation 2 provides the transport equation for suspended sediment. The annual averages of three of these terms are shown in Figure 13, but it is unclear how these were calculated. For example, HADV, units of mg/L/s, does this represent both the u- and v-component? Is it vertically averaged?    Similar for the settling and resuspension terms: are these vertical averages? Near-bed values?

Response:

Thank you for your insightful comments. HADV represents the total horizontal advection of suspended sediment, incorporating both the u- and v-components. All the terms previously presented in Figure 13, including horizontal advection (HADV), settling, and resuspension, are vertically averaged values. However, we now realize that this section is not closely aligned with the main focus of our study. We have therefore removed this section to ensure that the manuscript emphasizes our primary findings more clearly.

Comment:

Lines 714 – 717 are somewhat general statements instead of building on the analyses presented in this section.

Response:

Thank you for your suggestion. We have now removed this content.

Comment:

Discussion Section 5.2 should do a better job of extending this study within the literature. The authors cite papers but often do not provide the context. For example, Line 723 states that most of the Pearl River sediment stays near the estuary. Walsh's

and Hanebuth's papers are cited, but this study is not related to the conceptual frameworks in these papers. Line 741 mentions a "12 km threshold" without any context. Lines 747 – 752 make an interesting point, that Walsh' framework may not be directly suitable for monsoonal systems, particularly when the sediment delivery does not coincide with energetic conditions. This might be expanded and explained.

Response:

Thank you for your constructive feedback. We have made appropriate revisions to Section 5.2 to better contextualize our citations. We believe these modifications enhance the clarity and depth of our discussion.

Comment:

The conceptual model (Figure 14) presents schematics of dispersal dynamics under "normal" conditions (the annual-average?), and of a "no wind no waves" scenario. Figure 14 shows the idealized "no wind no waves" scenario, and the annual average that may not represent "normal conditions" because of the monsoonal nature. A conceptual model that compared the summer (high discharge, low wind/waves) to winter (low discharge, energetic winds/waves) would be interesting.

Response:

Thank you for your insightful comment. We acknowledge your point and agree that our previous conceptual model may not have adequately captured the monsoonal variability. In response, we have removed the original conceptual diagram to better align with the study's focus.

Comment:

The model grid extending from Taiwan to the Beibu Gulf. The authors demonstrate model skill in the "Near" region of the model, and no observations from the "Western" or "Beibu Gulf" regions are shown (see Figure 1 for locations). Observations are limited to two months for water level and waves (Figure S1), and one week for S/T/SSC (Figure S2). However, the paper's most interesting findings rely on the seasonal differences in hydrodynamics and sediment dispersal, and on delivery of sediment to distal regions such as Beibu Gulf. The model results would be more definitive if stations further from PRE were included, and if annual cycles were considered for at least some values (such as water level and wave height).

Response:

Thank you for your valuable comments. We sincerely appreciate your suggestions and have taken steps to enhance the validation of our model results. First, we would like to clarify that the water level validation in the Pearl River Estuary (PRE) region we previously presented already covered a full year (12 months). In response to your suggestion, we have now included additional validation at stations farther from the PRE. Specifically, we have added water level validation (12 months) at two stations: Zhapo in the "Western" region and Qinglan in the "Distal" region. Additionally, we have incorporated a 38-day mooring-based current velocity validation in the "Southwestern" region and a one-week wave validation in the "Beibu Gulf" region. We hope these additional validations strengthen the reliability of our study's findings and provide a more comprehensive assessment of the model's performance. Thank you again for your insightful feedback.

Comment:

Figure S3 shows model skill for several stations that are near the PRE. These observations from August 2017 were used to justify model parameters, but sediment characteristics may vary seasonally. The paper would benefit from consideration of how uncertainties in sediment characteristics (settling velocities, critical shear stresses) propagate into uncertainties in their conclusions.

Response:

Thank you for your valuable suggestion. Now we have conducted seven numerical simulations to systematically evaluate the relative contributions of tides, waves, ambient shelf currents, seasonal variations in critical erosion stress, settling velocity, and the initial bed grain size distribution to the transport and dispersal of Pearl River-derived sediments. These simulations include one Control case and six Sensitivity cases. We have now incorporated the seasonal variation of sediment characteristics in six simulation cases, specifically by increasing the critical shear stress for sediment erosion during winter, based on findings from previous studies. Exp 5 (NVS) was identical to Control case, except that it did not account for the seasonal variation in critical erosion stress, keeping the winter and summer values the same. Exp 6 (DSV) was identical to Control case, except that it sets a double sediment settling velocity of the Control case. Additionally, we have analyzed these results and included a discussion in the main text to assess their impact on our conclusions. We appreciate your insightful feedback, which has helped improve the robustness of our study.

Comment:

In many places the text is overly vague; including Section 5.3. Some examples are given below,

Specific Comments (by line number in the preprint)

The Introduction should better motivate the study and analyses. Many of the references are cited with no summary or context, making it difficult to see how this builds on past studies. The introduction should especially provide stronger motivation for the focus on winds and waves (Lines 78 – 79); and better place the regional setting of the PRE Shelf within the context of past research of the behavior of freshwater and sediment plumes under different wind and wave conditions.

Response:

Thank you for your helpful suggestion. We have made the corresponding revisions. Below is the rewritten version of **The Introduction**:

[revised manuscript text omitted]

Comment:

Sections 3.1 – 3.3: The authors describe the model implementation, but it is impossible to include all of details. Many modeling papers instead provide a complete archive of the model input files (for example see the Data Availability statement in

Moulton et al. JGR 2024 https://doi.org/10.1029/2023JC019685). Some issues that I see with the Methods description include:

Lines 142 – 144 provide model parameters theta_s and theta_b, but this is jargon of limited use even to ROMS modelers.

Response:

Thank you for your valuable suggestion. We understand that certain model parameters, such as theta_s and theta_b, may not be immediately familiar to all readers. Additionally, we recognize the difficulty of including every detail of the model implementation within the manuscript. In response to your comment, we will provide key configuration files (such as "ocean.in" and "sediment.in") as part of the supplementary materials and explicitly reference this in the Data Availability statement. This will enable readers to directly access and review the essential model parameters. We greatly appreciate your feedback and believe these additions will enhance the transparency, reproducibility, and accessibility of our study.

Comment:

Line 171: provide a citation for the wave-current interaction.

Response: Thank you for pointing that out. We have made the necessary revision and added the appropriate citation for the wave-current interaction.

Comment:

Line 176: suggest being more specific, "The freshwater and sediment discharges for the Pearl River …"

Response:

Thank you for your helpful suggestion. We have made the corresponding revisions.

Comment:

Line 180: What is the evidence that one year of spin-up is sufficient?

Response:

Thank you for your comment. In our initial setup, the critical shear stress was determined based on previous literature and model calibration. However, we acknowledge that this approach may have limitations, particularly in capturing seasonal variations in sediment properties. Observations, laboratory experiments, and numerical sensitivity analyses from prior studies have demonstrated that the critical shear stress of sediments in the Pearl River Estuary is higher in winter than in summer. In response to your insightful comment, we have conducted additional numerical experiments incorporating seasonal variations in critical shear stress. The results reveal that the bed sediment grain size distribution after realistic reworking is now significantly closer to the initial prototype compared to our previous simulations, which did not account for such temporal variability. This improvement underscores the importance of considering seasonal changes in sediment properties for accurately modeling sediment dynamics.

Comment:

Lines 188 – 190: Riverine sediment inputs was based on a rating curve and assumptions from earlier papers. The authors do not explain how they divide the total load from Zhang et al. (2012) into the distributary mouths. A 40% / 60% fraction was assumed for silt / clay based on earlier works (Table 1). Assumptions like these are

necessary, but the paper does not address how uncertainties in these types of factors might impact the overall results.

Response:

Thank you for your feedback. In the revised manuscript, we have provided the calculation formula for the rating curve, which is used to estimate the riverine sediment input. Based on this relationship, the total amount of Pearl River sediment input over our 12-month study period was calculated to be 34.52 million tons. This value aligns closely with the annual load reported in 2017 by the Pearl River Water Resources Commission. The riverine sediment input, derived from the river discharge, was allocated across the eight outlets located on the northern boundary (Figure 1b) based on the distribution approach of Hu et al. (2011). Additionally, in the discussion section, we have addressed the potential impact of uncertainties associated with the 40% / 60% fraction. We acknowledge that while such assumptions are necessary, they may introduce uncertainties, and we have discussed how these uncertainties could influence the overall results.

**References**

Hu, J., Li, S., and Geng, B.: Modeling the mass flux budgets of water and suspended sediments for the river network and estuary in the Pearl River Delta, China, Journal of Marine Systems, 88, 252-266, 10.1016/j.jmarsys.2011.05.002, 2011.

Comment:

Lines 190 – 233. The paper could benefit from consideration of the sensitivity of the results to uncertainties in the initial bed grain size distribution, and sediment

properties (settling velocity, critical shear stress). Some specific examples:

The lack of grain size distributions in part of the model domain (i.e. the Beibu Gulf especially) lead to large uncertainties in transport there.

Response:

Thank you for your valuable feedback regarding the sensitivity of the results to uncertainties in the initial bed grain size distribution and sediment properties. In response, we conducted seven different cases to systematically evaluate the influence of these factors, including variations in the initial bed grain size distribution, settling velocity, and critical shear stress. Our findings indicate that the grain size distribution has a relatively minor impact on the transport of Pearl River sediment on the continental shelf. Instead, the sediment transport dynamics are predominantly governed by physical factors such as hydrodynamic conditions, tidal forces, and regional circulation patterns. This suggests that while uncertainties in grain size distribution exist, particularly in areas like the Beibu Gulf, their effect on the overall sediment transport is limited compared to the dominant physical drivers.

Comment:

The text states that the "spin-up greatly reduced the irregularities" (line 231). But that is not evident in the figure.

Response:

Thank you for your comment. We have incorporated seasonal variations in critical erosion stress into our model and re-simulated the results. The revised findings now demonstrate a closer alignment with the initial grain size distribution, effectively

addressing the discrepancy identified in the previous analysis. This refinement offers a more precise representation of sediment dynamics in the study area.

Comment:

Panels 2g-i show that spatial grain size variability develops for example at lon/lat 114E,19N. These features are also seen in the bed shear stress (Figures 4F, 4H). Are they a model artifact or instability? Are they aligned with bathymetry (bathymetric contour lines in Figure 1b might help).

Response:

The features observed at 114E,19N are located near the model boundary where water depths are relatively large and boundary effects may be significant. However, this location is quite far from our main study region of interest. Based on our analysis, these boundary features have minimal impact on the key conclusions of this study. To provide more context, we have now added bathymetric contour lines to the relevant figures (e.g. Figure 1b). The bathymetry shows that 114E,19N is in a deep area near the model boundary. While some model artifacts or instabilities may occur in this boundary region, they do not affect the primary results and interpretations for our main study area. The spatial grain size variability and bed shear stress patterns in our core study region are robust and not influenced by these distant boundary effects.

Comment:

Panel 2i shows that large parts of the grid become much sandier than the initial grain sizes.

Response:

Thank you for highlighting the observation in Panel 2i. We have incorporated seasonal variations in critical erosion stress into our model and re-simulated the results. The revised findings now demonstrate a closer alignment with the initial grain size distribution, effectively addressing the discrepancy identified in the previous analysis. This refinement offers a more precise representation of sediment dynamics in the study area.

Comment:

Figures 9 and 10: The important points could be made with fewer figure panels.

Response: Thank you for your suggestion. We have now removed the original Figures 9 and 10, as we have significantly revised the paper by conducting seven cases. We found that the original Figures 9 and 10 were not closely related to the content of the revised manuscript and appeared redundant.

Comment:

Lines 564 – 566 and elsewhere. The paper mentions thermal winds, upwelling behavior, and a strong jet. The introduction does not provide the context for these processes. The paper does not identify these features within the model results. This adds complexity without reinforcing the main conclusions. I suggest removing these remarks here and elsewhere to focus on the main findings.

Response: Thank you for your comment. We agree with your suggestion and have removed this context, as it was not directly related to our main findings. This revision streamlines the paper and ensures a clearer focus on the key conclusions.

Comment:

Conclusions should be strengthened to better emphasize novel findings of your paper. For example, lines 878 – 883 discuss the overall sediment dispersal patterns modeled, but do not emphasize the disconnect between sediment delivery (summer high discharge) and redistribution (winter high energy). Paragraph lines 884 – 890 also do not provide very much that is novel.

Response:

Thank you for your valuable feedback. In response to your suggestion, we have revised the conclusion to more clearly highlight the novel findings of our paper, particularly the disconnect between sediment delivery during the summer high-discharge period and redistribution during the winter high-energy period. Additionally, we have removed paragraph lines 884–890, as they did not provide significant new insights. We hope these revisions enhance the clarity and focus of our conclusions.

Comment:

Technical Corrections (by line number in the preprint)

Figure 1: Bathymetry unclear. It is hard to see Taiwan Banks and offshore bathymetry that may impact shear stress (Figure 4f) and grain size (Figure 2g-i).

Response:

Thank you for your feedback regarding Figure 1. We have taken your comments into consideration and have revised the figure to improve the clarity of the bathymetry.

Comment:

Lines 132 – 134 (etc.). Cite sources for component numerical models (ROMS, SWAN,

Response:

Thank you for your feedback. We have made the necessary revisions as per your suggestions.

Comment:

In some places the text uses "Case 1" and "Case 2", and in other places it uses "WW" and "NWW". Use either "Case 1 and 2" OR "WW and NWW" in all places.

Response:

Thank you for your feedback. We have made the necessary revisions as per your suggestions.

Comment:

Perhaps add "Boundary Forcing" to Table 2 for ROMS (i.e. for Case 2: do you still use the same open boundaries for water levels, ubar/vbar, T/S, etc.?).

Response:

Thank you for your invaluable feedback, which has significantly strengthened our study. In response to your comments, we have conducted seven additional numerical simulations to systematically evaluate the relative contributions of tides, waves, ambient shelf currents, seasonal variations in critical erosion stress, settling velocity, and the initial bed grain size distribution to the transport and dispersal of Pearl River-derived sediments. These simulations include one Control case and six Sensitivity cases. In our initial simulations, we inadvertently overlooked that the HYCOM-derived flow fields already account for the influence of monsoon-driven

currents, South China Sea Warm Current (SCSWC), and Kuroshio intrusions on shelf circulation. This includes the strong northeastward currents in summer and the weaker northeastward or southwestward currents in winter. To address this, we have refined our approach in the NAS case (i.e., no ambient shelf currents) by excluding HYCOM-derived water levels and velocities. This adjustment allows for a clearer assessment of the influence of South China Sea (SCS) circulation while eliminating potential inaccuracies arising from monsoon-driven currents and Kuroshio intrusions. As a result, our model now provides a more accurate representation of the underlying physical processes. We believe these revisions enhance the robustness of our findings, and we sincerely appreciate your insightful suggestions.

Comment:

Line 372: It is vague to say "widespread expansion of the river plume". Figure 5A does not look like the plume spreads far into the sea; it seems to hug the coast.

Response:

Thank you for your comment. The primary reason for this is that our previous analysis focused on the vertically averaged salinity field. In the revised manuscript, we now present both surface and bottom salinity fields. The surface salinity field more clearly highlights the significant eastward expansion of the plume. We have made the necessary modifications to the manuscript accordingly.

Comment:

Line 389: vague to say "pathway is wide but magnitude is weak". You could be less vague by giving quantitative scales here (and elsewhere).

Response:

Thank you for your feedback. We have made the necessary revisions as per your suggestions.

Below is the rewritten version:

"*The westward transport pathway follows the region where the water depth is shallower than 30 m, with a riverine sediment flux of 10–20 $g^{-1}$ $m$ $s^{-1}$. In contrast, the eastward transport pathway occurs in the 30–60 m depth range, but the riverine sediment flux is below 10 $g^{-1}$ $m$ $s^{-1}$.*"

Comment:

Line 384: Confusing and vague to say "predominantly transported westward and eastward". Suggest something like "there are both westward and eastward fluxes of riverine sediment (Figure 5c)."

Response:

Thank you for your feedback. We have made the necessary revisions as per your suggestions.

Comment:

Line 386: refers to the diversion around Taiwan Bank. However, this feature is not obvious in Figure 5c. Adding bathymetric contours that show Taiwan Bank may help. The time- and depth-averaged currents (figure 5a) show cross-shore variability; but the reported upwelling feature is not obvious.

Response:

Thank you for your suggestion. We have now added bathymetric contours to Figure 5

and have redrawn the figure to illustrate both surface and bottom currents. The bottom currents clearly show strong cross-shore upwelling.

Comment:

Line 449 – 450: is it wind and wave "mixing" that impact the plume behavior; or wind and wave forcing which would include net momentum terms as well as turbulent mixing? Also, you are vague here saying "wider" instead of length scale for the width.

Response:

Thank you for your suggestion. We have now removed this section. Regarding the role of wind, we have chosen not to consider it in this study. The results from the NWS-Control case only represent the estimation bias associated with neglecting wave effects. We have ensured that wind stress is consistent across all our simulations. In this study, wave forcing includes both the net momentum terms generated by the waves and turbulent mixing.

Comment:

Lines 521 – 528 and associated text. Phrases like 'Near region" and "Far off region" are awkward. Suggest "proximal" and "distal".

Response:

Thank you for your suggestion. We have now updated the text by replacing "Near region" and "Far off region" with "proximal" and "distal" to ensure consistency and clarity.

Comment:

Lines 555, 558, 561: the authors use the word "transported" but, then provide the

percentages that are "retained". Suggest using the words "deposited" or "retained" if

you are talking about the sediment deposition.

Response:

Thank you for your suggestion. We have now adopted the term "retained" in our text.

Comment:

Supplement Line 63: the 43 stations are not identified; text should refer to Figure 1.

Response:

Thank you for your feedback. We have made the necessary revisions as suggested,

and the text now refers to Figure S1b for the identification of the 43 stations.

---

## Referee Report (RR1)

**1. Introduction**

This is my first time reviewing this manuscript, and my assessment has been conducted independently without taking earlier versions and previous referee comments into account.

Using a numerical case study, this paper investigates the dispersal of river-derived sediment over a continental shelf. It does well to provide insights into the spreading of sediment from the river plume over the coastal shelf, taking several processes, such as tides, waves and wind into account. While I think the paper does provide new insights into these main dispersal mechanisms, I think the attempt to form a generalised interpretation is currently not attained, and it might be more appropriate to shift the focus more to the specific case that is investigated. As I do see the value of the developed model and the on/off sensitivity analysis of several model aspects, I think a more careful presentation of the results is required for suitable embedding. Therefore, I recommend publication of the manuscript after major revision.

**2. General Comments**

**Clarification of "Overestimation" and "Underestimation"**

The manuscript frequently refers to the "overestimation" or "underestimation" of sediment-related quantities (e.g., deposition, resuspension, retention, and bed shear stress) in several instances (e.g., lines 24, 27, 631, 667, 830, 1004, 1011, 1085, among others). These terms imply comparison with real-world field data, which, to my knowledge, is not available for validation (as also noted in lines 805–808). Instead, these quantities appear to be evaluated relative to the "Control" model run. To avoid potential misinterpretation, I recommend revising the phrasing to clarify that differences are being assessed within the model framework rather than against observational data.

**Evaluation of Research Objective (2)**

It is unclear whether research objective (2) (line 156) has been fully achieved. My interpretation is that the model is calibrated based on hydrodynamics to establish the "Control" run, but it remains uncertain whether the model continues to behave as expected when certain processes (NWS, NTS, NAS) are omitted. Additionally, it would be beneficial to clarify whether the chosen methodology is capable of addressing objective (2).

There is potential for deeper insights into the contributions and interactions of different processes in sediment transport. Since sediment transport exhibits nonlinear responses to environmental forcing, an analysis of how tides and waves interact to influence overall sediment dynamics could strengthen the study. Currently, the approach primarily isolates individual processes, but a more integrated interpretation of process interactions could add value.

Some of the study's findings may already be anticipated based on fundamental process-based reasoning. For instance, the following statements confirm well-known expectations:

- Lines 525–527: "With sediment load during winter nearly negligible, the suspended concentration of riverine sediment is significantly lower compared to the wet summer."
- Lines 657–658: "For the NTS case versus the Control case, tides significantly affected bottom stress."
- Lines 667–668: "However, NWS underestimated the nearshore bottom stress."
- Lines 713–714: "This enhanced settling velocity resulted in an increased deposition."

A more nuanced discussion of these results—particularly emphasizing interactions between multiple processes rather than confirming expected trends—would enhance the manuscript.

**Numerical Model Performance and Limitations**
The manuscript could benefit from a more detailed discussion of numerical model performance, particularly regarding:
- Grid sensitivity: While the hydrodynamic validation appears strong, has numerical behaviour and model convergence been tested under grid refinement?
- Model limitations: Section 4.3 primarily discusses the exclusion of certain physical processes but does not address intrinsic model limitations. Consider including a discussion on potential numerical constraints.
- Sigma-coordinates (Line 189): The sigma-coordinate system is known to introduce challenges in accurately modelling salt transport in regions with steep gradients, where Cartesian coordinates may perform better (Bijvelds, 2001). Given the importance of salinity in estuarine turbidity maxima (ETM) formation, it would be helpful to clarify whether ETM development is well captured in the model and whether any limitations arise from the chosen coordinate system.

The study presents interesting insights into sediment budgets and overall sediment dynamics. In particular, the introduction of the "Cycle" model run is a compelling aspect. It may be beneficial to highlight its implications more prominently, especially regarding the conclusions drawn in lines 730–731.

**3. Technical comments**
The level of significance reported for measured percentages is inconsistent across the manuscript. Given the model's inherent (in)accuracy, the precision of percentage values should be adjusted accordingly. This applies particularly to the Abstract, Section 3.2, and Section 5 (Conclusions).

Line 192: Song and Haidvogel (1994) introduce a new s-coordinate system, not the general sigma-coordinate system introduced in the manuscript. It is unclear which system is actually used and what the two $\theta$-values are, as these are not explicitly described in Song and Haidvogel (1994). Please clarify and revise accordingly.

Lines 303-305. The statement "*The realistic spin-up greatly reduced the irregularities and prepared a more suitable seabed sediment particle size distribution field for subsequent simulations than the initial prototype (Figures 2d-f vs. 2g-i).*" needs further clarification.

Could you elaborate on why this "realistic reworking" is necessary and why it results in a more suitable initial sediment distribution? The figures suggest the presence of spurious oscillations in the model interior and near the open southern boundary, which raises concerns.

Line 483-487. The precise definition of a "river plume" is unclear. How is it quantitatively defined, and what correlation does it have with regions of high suspended sediment concentration (SSC)? Additionally, a more in-depth analysis of the freshwater plume's role in transporting riverine sediment—relative to the overall sediment transport—would strengthen this section. Could you expand on this aspect?

Line 493: Please explain how the sediment flux is determined. Is it computed purely as advective horizontal transport, or are other processes included? Please specify.

Line 541. Figure 6 shows significant salinity gradients near the estuary, which suggests the potential formation of an estuarine turbidity maximum (ETM). Is such an ETM captured in the model? If so, could you comment on its role in sediment transport within the system?

Line 639: the term "probably being artefacts" is vague. Could you clarify what kind of numerical artifacts these are? While they appear as local deviations in deeper regions, they span several kilometres and multiple grid cells. Given their scale, could they significantly influence hydrodynamics across the domain? Furthermore, how does this uncertainty impact confidence in the NAS model results? A more detailed explanation would be helpful.

Line 694+733. Figures 9 and 10 include arrows that are somewhat unclear. While the colours indicate differences between model runs, it appears that the arrows represent only the non-Control run. Could you clarify their specific purpose and whether they provide a direct comparison with the Control case?

Lines 728-731 describe the deposition seen in Figure 11f but provide no reason for it by the driving hydrodynamics. To me, it is unclear why the sediment distribution would change. What is the process behind this? Does this not provide us with the conclusion that a more complete sediment spin-up, as performed in Cycle, is necessary for robust results?

Line 793. The caption of Figure 12 explains the significance of magenta values, but it is unclear what the other values (black, blue, red) represent. Could you specify their meaning?

**4. Textual remarks**

Line 25: The term "distal retention" may not be appropriate in this context. If sediment is never transported to distal regions in the first place, it does not necessarily imply a lower distal retention. Consider rephrasing for clarity.

Line 219: The abbreviation Hsig appears without an introduction. If it refers to significant wave height, please define it properly when first mentioned.

Line 298: I am not sure what "realistic reworking" means, could you explain or rephrase?

Line 495: The sediment flux is not strictly westward and eastward. Please consider rephrasing for accuracy.

Line 502 and 504. The units used for sediment flux are not correct -> g/m/s

Line 524: The term "salinity front" is somewhat vague. Does this refer to a specific isohaline? Please clarify.

Line 542. Figure 6e: "2D riverine Flux" is unclear phrasing to me. Perhaps "Horizontal sediment flux" would be a more precise alternative, as "riverine" may be ambiguous without additional context..

Line 542. Figure 6f. The term "riverine Deposition reworking" is unclear. Consider rephrasing to include "difference," "change," or another term that better captures the intended meaning.

There are some inconsistencies in terminology throughout the manuscript. For example, "Distal" regions is sometimes written in quotation marks, while in other places it appears as distal regions without emphasis. Similarly, "Gulf" regions and gulf regions are used interchangeably. Please ensure consistency in terminology throughout the text.

**References**
Bijvelds, M. D. J. P. (2001). Numerical modelling of estuarine flow over steep topography. Doctoral dissertation, TU Delft, Delft University of Technology. Ipskamp. Retrieved from http://resolver.tudelft.nl/uuid:39c0c858-579a-47fa-a911-4a4114949a11

---

## Author Response (AR2)

Dear authors,

Thank you for the revised manuscript, which has much improved. As referee #1 comments, your manuscript has changed a lot compared to the previous submission. Even if it has much improved, there are still significant comments by both referees, which you should account for in a new revised submission. I encourage you to resubmit, accounting for the useful comments by both referees.

Thanking you and with best wishes

Mario Hoppema

Dear Editor Dr. Hoppema,

Thank you for your time and effort in handling our manuscript. We sincerely appreciate your positive feedback and the opportunity to revise and resubmit.

In response to both reviewers' useful comments, we have carefully revised the manuscript, corrected previous errors, and made substantial improvements. All points have been addressed in detail in the response letter.

Additionally, Sun Yat-sen University has multiple campuses in different cities. All authors are based at the School of Marine Sciences on the Zhuhai campus, so the affiliation in the manuscript is accurate and should remain unchanged.

Thank you again for your kind support. We look forward to your further feedback.

With best regards,

Wenping Gong

**Response to Reviewer 1**

General comments and summary:

I have read the responses to both reviews of the initial submission, and the revision. The authors have done a conscientious job of revising the manuscript to address concerns. Note that this is a very different paper than the initial submission and has a different focus than the original (e.g. "winds" was removed from the title). The initial submission compared two model runs ("with winds and waves" and "without winds and waves"). The revision compares SEVEN model runs, only one of which was in the first submission. The revision removed two discussion sections.

This submission seems much closer to the standard expected for your journal, but the paper will benefit from some additional revisions, as outlined below.

My review of the first submission raised four major concerns: "(1) some of the analysis seem poorly justified … ; (2) the organization should be improved …; (3) the skill assessment has geographic and temporal limitations; and (4) the findings are not generalized to other systems." The authors have addressed most of these; they removed the analyses that were poorly justified, improved organization, and included a more comprehensive skill assessment in the Supplement.

Dear Reviewer:

We greatly appreciate the time you spent reviewing our manuscript and providing constructive comments and minor revisions. These comments and insights are helpful for improving the quality of our paper. We address your comments point-by-point as follows.

Below are general comments that will improve the paper:

1. The introduction is improved upon the original submission. It is better organized and includes more context. However, most of the citations are based on fairly localized, often small-spatial scale, studies of estuaries. This seems inappropriate, given that the model domain is a ~1700 km scale continental shelf. The introduction should rely on continental shelf papers moreso than estuarine. Some suggestions (Geyer et al. 2004; McKee et al 2004; etc.); continental shelf modeling papers (Warner et al. 2017; Dalyander et al, 2013; Harris et al. 2008; Wang et al. 2020; Zang et al. 2019; Xu et al. 2016).

Response:

   Thank you for the thoughtful comment. We appreciate the suggestion to incorporate more references focused on continental shelf processes to better align with the scale of our model domain. In response, we have revised the Introduction to include all the key references you mentioned (Geyer et al., 2004; McKee et al., 2004; Harris et al., 2008; Dalyander et al., 2013; Xu et al., 2016; Warner et al., 2017; Zang et al., 2019; Wang et al., 2020). These additions strengthen the contextual foundation of our study and ensure that the literature cited reflects the appropriate spatial scale of the continental shelf system we investigate.

**References**

Dalyander, P. S., Butman, B., Sherwood, C. R., Signell, R. P., and Wilkin, J. L.: Characterizing wave- and current- induced bottom shear stress: U.S. middle Atlantic continental shelf, Continental Shelf Research, 52, 73-86, https://doi.org/10.1016/j.csr.2012.10.012, 2013.

Geyer, W. R., Hill, P. S., and Kineke, G. C.: The transport, transformation and dispersal of sediment by buoyant coastal flows, Continental Shelf Research, 24, 927-949, 10.1016/j.csr.2004.02.006, 2004.

Harris, C. K., Sherwood, C. R., Signell, R. P., Bever, A. J., and Warner, J. C.: Sediment dispersal in the northwestern Adriatic Sea, Journal of Geophysical Research, 113, 10.1029/2006jc003868, 2008.

McKee, B. A., Aller, R. C., Allison, M. A., Bianchi, T. S., and Kineke, G. C.: Transport and transformation of dissolved and particulate materials on continental margins influenced by major rivers: benthic boundary layer and seabed processes, Continental Shelf Research, 24, 899-926, https://doi.org/10.1016/j.csr.2004.02.009, 2004.

Wang, C., Liu, Z., Harris, C. K., Wu, X., Wang, H., Bian, C., Bi, N., Duan, H., and Xu, J.: The Impact of Winter Storms on Sediment Transport Through a Narrow Strait, Bohai, China, Journal of Geophysical Research: Oceans, 125, e2020JC016069, https://doi.org/10.1029/2020JC016069, 2020.

Warner, J. C., Schwab, W. C., List, J. H., Safak, I., Liste, M., and Baldwin, W.: Inner-shelf ocean dynamics and seafloor morphologic changes during Hurricane Sandy, Continental Shelf Research, 138, 1-18, 10.1016/j.csr.2017.02.003, 2017.

Xu, K., Mickey, R. C., Chen, Q., Harris, C. K., Hetland, R. D., Hu, K., and Wang, J.: Shelf sediment transport during hurricanes Katrina and Rita, Computers & Geosciences, 90, 24-39, https://doi.org/10.1016/j.cageo.2015.10.009, 2016.

Zang, Z., Xue, Z. G., Xu, K., Bentley, S. J., Chen, Q., D'Sa, E. J., and Ge, Q.: A Two Decadal (1993–2012) Numerical Assessment of Sediment Dynamics in the Northern Gulf of Mexico, Water, 11, 938, 2019.

2. The paper models the dispersal of the Pearl River sediment on the continental shelf. However, in many cases, the paper presents the study as if it were a Pearl River Estuary (PRE) study rather than a continental shelf study. Most of the analyses is done for the region of the shelf outside of the PRE (i.e. Regions 2 – 8 on Figure 1). In these cases,

the paper should refer to the shelf region rather than the estuary.

Response:

   Thank you for your comment. We have carefully revised the manuscript to clarify that our focus is on sediment dispersal across the continental shelf rather than within the Pearl River Estuary (PRE). Where applicable, we now explicitly refer to Regions 2–8 (Figure 1) as the shelf region rather than the PRE. This adjustment ensures consistency with the study's broader spatial scope.

3. The abstract and motivation should be more clear about what will be analyzed. The analyses focuses on the dispersal of Pearl River discharged sediment (classes 4 and 5); and not the seabed sediment (seabed and riverine, classes 1 – 3)?

Response:

   Thank you for pointing this out. We have revised both the abstract and the motivation section to clearly specify that our analysis focuses on the dispersal of Pearl River-derived sediment, namely sediment Classes 4 and 5 in Table 1. These correspond to slow-settling single fine grains (Class 4) and fast-settling flocs (Class 5), in contrast to background seabed sediments (Classes 1 to 3), which are not the focus of this study.

Specific Comments (by line number in the preprint)

4. Lines 297 – 328: This section is confusing and should be revised for clarity. For example, better clarify whether Pearl River sediment (classes 4 and 5) was delivered during the 15-month spinup.

Response:

We thank the reviewer for pointing out the ambiguity. We have substantially rewritten lines 297–328 to improve clarity. In particular, the revised text now explicitly states that Pearl River sediments (Classes 4 and 5) were indeed delivered continuously throughout the 15-month spin-up period. The updated text reads:

[revised manuscript text omitted]

Some specific points:

5a. Lines 413 – 414: Run 5 uses the same critical shear stress for winter and summer,

Response:

Thank you for your comment. To clarify, Exp 5 employed a constant critical shear stress ($\tau_{ce}$) value throughout the entire simulation period, using the summer $\tau_{ce}$ value from Table 1 for both winter and summer seasons (i.e., without seasonal variation). We have revised the manuscript to explicitly state this in the Methods section:

*Exp 5 (NVS hereafter) replicated the setup of Experiment 1, but with one modification: it used a constant critical shear stress for erosion ($\tau_{ce}$) across both seasons, specifically adopting the summer $\tau_{ce}$ value from Table 1 throughout the simulation (i.e., no seasonal adjustment between winter and summer).*

5b. Lines 416 – 423; model run 7 is described as considering the impact of the initial sediment bed. But, it actually seems to evaluate whether the original spinup time was sufficient for supplying riverine sediment (classes 4 -5) to the seabed. It does not seem to consider uncertainties in the distribution of size classes 1- 3.

Response:

Thank you for your comment. The main difference between Exp 7 and Exp 1 is that Exp 7 uses the final state of Exp 1 as its initial condition. This means that when Exp 7 starts, it already includes all the changes in sediment classes 1–5 that occurred during the Exp 1 simulation, and thus classes 1–3 are also affected. We agree with your observation: compared to the Cycle 2 case, Exp 7 (Cycle) indeed focuses more on assessing how the presence of previously deposited riverine sediments influences the

evaluation of riverine sediment transport. To avoid misunderstanding, we have revised

and clarified the text. The updated version is as follows:

*Finally, to assess the model's sensitivity to the spin-up duration of Pearl River-derived*

*sediment, particularly regarding the retention of riverine sediments in both the water*

*column and the seabed, we adopted the sediment distributions (Classes 1 to 5) from the*

*Control run on March 31, 2018, as the alternative initial conditions for the Cycle*

*experiment (designated as Exp 7, Cycle hereafter). This setup carries over the full year's*

*evolution of riverine sediment transport and deposition from the Control run (Exp 1),*

*including changes in all sediment classes, into the start of Exp 7. As a result, Exp 7*

*mainly evaluates how the presence of previously deposited riverine sediments*

*influences subsequent sediment transport estimates.*

5c. Lines 423 – 427 seem out of place. It provides the wind forcing used for all model

runs. Perhaps move this to the beginning of the section.

Response:

   Thank you for the suggestion. We agree that the original placement of Lines 423–

427 was not optimal. In the revised manuscript, we have moved this content to the

beginning of Section 2.6 to improve the logical flow and clarity of the methods section.

6. Section 3.3 is difficult to follow, suggest revising it. It is long (12 pages). Each of the

5 figures contains several panels that each compare a different model to the Control run.

Each paragraph of the text, however, discusses one model run, so the reader needs to

look at several figures to understand each paragraph.

Suggest you redo the figures to be organized like the text. If a paragraph describes one model run (e.g. NTS), it can refer to 1 figure that shows the comparison between the model and control for multiple variables (bed stress, SSC, currents, flux, etc).

Response:

Thank you for your comment. We appreciate your suggestion and, in fact, initially intended to organize the figures as you proposed—i.e., having each figure represent one model run and include multiple variables (e.g., bed stress, SSC, currents, fluxes, etc.). However, we encountered several challenges with this approach:

1. It would increase the number of figures. Currently, Section 3.3 contains 5 figures, but under the proposed structure, at least 6–7 figures would be needed, potentially making the manuscript more cumbersome.

2. This organization would lead to inconsistencies in the number of subpanels across figures, as different experiments impact different variables. For instance, Experiments 2–4 affect hydrodynamic processes such as bed stress and currents, whereas Experiments 5–7 do not. As a result, some figures would contain 5–6 subpanels, while others would have only 3–4, which could make comparisons more confusing.

3. Even if we follow the approach where each figure represents a single model run and includes multiple variables (e.g., bed stress, SSC, currents, fluxes, etc.), it is still unavoidable to compare results across different experiments. As a result, it remains necessary to reference multiple figures within a single paragraph.

Therefore, we adopted the current format, following the presentation style used in Xue et al. (2012). In their paper, Figures 5 and 6 present summer and winter current velocity and the difference in Mekong River-derived SSC between experiments and the Control run, and Figure 8 illustrates annual simulated Mekong River-derived sediment deposition in the Mekong River Shelf. Similarly, in their Section 3.2, each paragraph refers to multiple figures (e.g., Figures 5, 6, and 8). This structure has proven effective, as their work was successfully published in Continental Shelf Research and has been cited 112 times as of May 7, 2025 (Google Scholar), indicating its impact and clarity. That said, we fully agree that Section 3.3 was overly long and difficult to follow. In response, we have revised the text and split the original Section 3.3 into two sections (now Sections 3.3 and 3.4) to make the descriptions more concise and precise, thereby enhancing both readability and overall coherence.


Also, I do not think that the "initial sediment bed" experiments (Cycle and Cycle2) are truly tests of the initial bed so much as a test of the length of the spinup time for riverine sediment dispersal.

Response:

We have revised the Conclusion (Section 5) to synthesize key insights rather than restate individual results. The updated takeaways emphasize the roles of tides, waves, and remote forcing in shaping sediment dispersal, aligning with the reviewer's recommendation. To avoid misinterpretation, we also removed terms such as 'underestimation' and 'overestimation' (Lines 1085–1107) and clarified that all comparisons are made within the model framework. Regarding the experiments, the

Cycle case was designed to examine the influence of riverine sediment spin-up by analyzing the redistribution of retained Pearl River–derived sediments during the second simulation year. In contrast, the model exhibits only minor sensitivity to the duration of seabed sediment spin-up, as demonstrated in the Cycle2 experiment (see Supplement), in which riverine sediments present during the spin-up period were added to seabed sediment classes 1–2 at the start of the Cycle2.

Technical Corrections (by line number in the preprint)

11. Line 236: Confusing and vague wording: "More than one year of hydrodynamic and sediment spin-up is sufficient…".

Response:

Thank you for pointing this out. We agree the wording was unclear and have removed the sentence in Line 236 to prevent any potential misunderstanding.

12. Lines 502, 504, etc: the units of sediment flux seem incorrect. g-1 m s-1?

Response:

We appreciate your careful review. This was indeed a typographical error, and we have now corrected it accordingly.

13. Figures 5 and 6: it is difficult to compare the sediment panels (c-f) because they use very different color scales.

Response:

Thank you for your insightful observation regarding Figures 5–6. You are correct that the differing color scales made direct comparison challenging. While we initially employed variable scales to account for the substantial seasonal variations in sediment concentration ranges, we agree that uniform scaling enhances comparability. Accordingly, we have revised both figures to maintain consistent color scales wherever possible. This adjustment indeed improves clarity of cross-seasonal variation while preserving data representation.

However, we should note that Figures 5F and 6F still require distinct value ranges and color scales due to their differing data characteristics:

- **Figure 5F** displays the thickness of Pearl River sediment deposits on the seabed at the end of summer (positive values only).

- **Figure 6F**, in contrast, illustrates the *change* in sediment thickness between the end of winter and the end of summer (including both positive and negative values).

Since these subplots represent fundamentally different metrics, a shared color scale could not be applied without compromising interpretability.

[Figure]

**Figure 5.** Patterns averaged over the entire wet summer season in the Control case: (a) surface and (b) bottom salinity (color, psu) and flow (arrows, m s$^{-1}$); (c) surface and (d) bottom riverine (classes 4 and 5 in Table 1, as follows) SSC (mg L$^{-1}$); (e) depth-integrated horizontal riverine sediment transport rate (color, g m$^{-1}$ s$^{-1}$) and direction (arrows); and (f) riverine sediment deposition thickness (mm) on the seabed during the wet summer season. Flow vectors in regions with water depths exceeding 100 m are masked for clarity.

[Figure]

**Figure 6.** Same as Figure 5, but for the dry winter season in the Control case. Notably, (f) illustrates the changes in riverine sediment deposition (classes 4 and 5 in Table 1) on the seabed at the end of the dry winter season compared to the end of the wet summer season.

14. Line 613: are the "red and blue values" on Figure 7?

Response:

Thank you for catching this discrepancy. Due to a file import error during our extensive revisions, an incorrect version of Figure 7 was inadvertently included. We have now corrected the figure and verified its accuracy. The updated version clearly displays the red and blue values as intended. For clarity, the corrected Figure 7 is shown below. We apologize for any confusion caused by this oversight.

[Figure]

**Figure 7.** Riverine sediment (classes 4 and 5 in Table 1) retention budget percentages at eight regions (see Figure 1) during (a) the wet summer season, (b) the dry winter season, and (c) the entire year in the Control run case. (d) the annual deposition patterns spanning from April 1st, 2017, to March 31st, 2018 in the Control Run. All percentages displayed in the figure are relative to the annual riverine sediment load (see Figure 3a). The black percentage values represent the combined total of riverine sediment Class 4 and Class 5, while the red and blue values denote sediment Class 4 and Class 5, respectively. Arrows indicate the direction of net riverine sediment flux at each transect during the specified period.

15. Line 698: These represent the concentrations of classes 4 and 5 and do not include classes 1 – 3? Suggest you clarify this in the figure caption and perhaps in the text.

Response:

Thank you for pointing this out. Yes, the concentrations shown represent only

sediment classes 4 and 5, excluding classes 1–3. We have clarified this in both the figure caption and the main text accordingly.

16. Figure 12 is complicated. Can it be simplified by removing some of the numbers?

Response:

Thank you for the helpful suggestion. We have revised Figure 12 to improve clarity by removing the percentage labels for sediment class 4. To obtain the values for class 4, readers can simply subtract the total percentage of class 5 sediment (blue numbers) from the combined total of sediment classes 4 and 5 (black numbers). We believe this adjustment simplifies the figure while still allowing readers to derive the full information if needed.

[Figure]

**Figure 12.** Same as Figure 7c, but for the other six cases: (a) NTS, (b) NWS, (c) NAS, (d) NVS, (e) DSV, and (f) Cycle, respectively. All percentages shown in the figure are expressed relative to the annual riverine sediment load (see Fig. 3a). Magenta values denote the differences in retention percentage of riverine sediments (Classes 4 and 5; Table 1) between the Control run and each sensitivity case. Black values represent the combined retention of Classes 4 + 5, while blue values indicate Class 5 alone. To obtain the retention percentage for Class 4, simply subtract the Class 5 percentage (blue) from the combined Classes 4 + 5 percentage (black).

17. Line 827: it seems inappropriate to refer to the original manuscript (preprint).

Response:

Thank you for pointing this out. We agree that referring to the original manuscript (preprint) is unnecessary in this context. We have removed the reference to the preprint and revised the sentence accordingly. The revised sentence now reads:

"*After realistic reworking during the spin-up, the bed sediment grain size distribution (used as initial conditions in the Control run and all Sensitivity cases except the Cycle case) is quite close to the initial prototype (Figures 2d–f vs. 2g–i).*"

18. Table 1, Line 863, and elsewhere: it seems a bit misleading to cite the Ralston paper for the settling velocities used for the riverine sediment because Ralston used a salinity-dependent settling velocity, whereas this paper partitioned sediment into fast- and slow-settling flocs.

Response:

Thank you for your comment. We agree with your observation, and we have removed the citation of the Ralston paper from both Table 1 and Line 863 to avoid any confusion.

19. Line 918: suggest you say "finding is consistent with the earlier…" instead of "confirm.

Response:

Thank you for your valuable suggestion. We have modified the wording on Line 918 to "*This finding is not only consistent with the earlier speculation proposed by Ge et al. (2014) but also supplements the conclusions drawn by Lin et al. (2020).*" to improve

clarity and precision.

Response:

Thank you for your valuable suggestion. We have modified the wording on Line 934 to "*Within this framework, riverine sediment deposition is characterized using key factors, including riverine sediment discharge (greater or less than 2 megatons), shelf width (greater or less than 12 km), and wave and tidal range conditions (greater or less than 2 m) (Walsh and Nittrouer, 2009).* " to improve clarity and precision.

21. Lines 971 – 975: this is confusing because you cite several studies of estuaries. However, your numerical model focuses on the continental shelf dispersal. Should you instead say "The PRE exhibits distinctive geomorphological features, yet dispersal of its fine-grained sediment on the continental shelf conforms to general patterns observed offshore of other monsoon-influenced estuarine systems. … documented offshore of various major …".

Response:

Thank you for pointing out the confusion. Our wording was indeed imprecise, as the studies we cited address sediment transport from the estuary out onto the adjacent continental shelf. We have therefore revised the sentence to read:

*The PRE exhibits distinctive geomorphological features, yet dispersal of its fine-*

*grained sediment transport on the continental shelf conforms to general patterns*

*observed offshore of other monsoon-influenced estuarine systems. Similar multiple-*

*stage sediment delivery and dispersal mechanisms have been documented offshore of*

*various major estuaries and their adjacent shelves, including the Yellow River Shelf*

*(Bian et al., 2013; Zeng et al., 2015), Changjiang River Shelf (Zeng et al., 2015), and*

*Mekong River Shelf (Xue et al., 2012; Eidam et al., 2017), demonstrating comparable*

*sedimentary processes under monsoon climatic influences.*

22. Line 980 and elsewhere: Instead of saying "Like the PRE", suggest you say "Like the PRE Shelf". This paper focuses on the continental shelf offshore of the PRE, not on the PRE itself. The paper should be more clear about this.

Response:

   Thank you for the suggestion. We have revised the text replace "Like the PRE" with "Like the PRE Shelf" where appropriate, to more accurately reflect the study focus on the continental shelf offshore of the PRE.

23. Lines 1026 – 1035: many of these citations are focused on the estuary itself or on global drivers, rather than continental shelf processes. There are no citations for line 1034 (episodicity of shelf transport for "many systems").

Response:

   Thank you for your comment. We have expanded our reference list to ensure that continental-shelf processes are properly cited. In particular, we now reference

interannual variations of shelf circulation (Liu et al., 2020; Deng et al., 2022) and

Kuroshio intrusions (Caruso et al., 2006; Nan et al., 2015; Sun et al., 2020). We have

also supported the statement on the episodicity of shelf transport by adding citations

that demonstrate how a large fraction of sediment flux on many shelf systems occurs

during short-lived, high-energy events such as storms and hurricanes (Xu et al., 2016;

Warner et al., 2017; Georgiou et al., 2024).

The revised text now reads:

*interannual variations of the shelf circulations (Liu et al., 2020; Deng et al., 2022) and*

*Kuroshio intrusions (Caruso et al., 2006; Nan et al., 2015; Sun et al., 2020). Therefore,*

*while this study sheds light on seasonal and annual timescale patterns, it cannot fully*

*represent the short or long-term transport and deposition trends of the Pearl River*

*sediment. Yet for many shelf systems, a lot of sediment transport happens during short-*

*lived events such as hurricanes (Xu et al., 2016; Warner et al., 2017; Georgiou et al.,*

*2024). Consideration of the episodicity of transport would be helpful for future studies*

*(Xu et al., 2016; Warner et al., 2017; Georgiou et al., 2024).*


*Table 3. Annual mean runoff and annual suspended sediment load of major rivers in South China that flow directly into the northern South China Sea since the 1950s (Milliman and Farnsworth, 2011; Zhang et al., 2012; Liu et al., 2016).*

| River name | Runoff ($m^3 s^{-1}$) | Suspended sediment load (Mt/year) |
|:---:|:---:|:---:|
| Pearl River | 9075 | 84.3 |
| Jiulong River | 476 | 3.1 |
| Han River | 825 | 10 |
| Moyang River | 269 | 0.8 |
| Jian River | 174 | 1.5 |
| Nanliu River | 162 | 1.1 |
| Changhua River | 120 | 0.08 |
| Nandu River | 179 | 0.4 |

25. Line 1084: should this say "in the PRE" or "within and offshore of the PRE"? The model domain extends very far beyond the estuary.

Response:

Thanks for pointing this out. We've revised it to 'within and offshore of the PRE' to accurately reflect the model domain's coverage.

26. Figure S1: use different types of markers (circles, squares, triangles, etc.) for the different types of observation points (tide gages, wave stations, survey stations, etc).

Response:

Thank you for your suggestion. We have revised **Figure S1** to improve clarity in distinguishing observation types by adopting different marker styles for each station category, as recommended. The updated **Figure S1** now uses:

- **Colored squares** (■) for wave stations (*W1*: black, *W2*: blue),

- **Colored triangles** (▲) for tidal gauge stations (*Zhapo*: green, *Qinglan*: cyan, *Quarry Bay*: red),

- **Colored diamonds** *(♦) for mooring stations (M2*: red, *M1*: green),

- **Uniform blue dots** (•) for all 43 survey stations.

This scheme ensures immediate visual differentiation between station types while aligning marker colors with those used in the main figures for consistency.

[Figure]

**Figure S1.** (a) Bathymetry contours and the model grid domain (black to white dashed lines), with circled numbers ①-⑧ indicating the eight regions: Proximal, Southern, Eastern, Southeastern, Western, Southwestern, Gulf, and Distal regions, which are delineated by transects and are described in detail in Section 4.2 of the main text. Thick gray contour lines mark the 20 to 80 m isobaths at 20 m intervals, while thin gray lines indicate the 100 to 1000 m isobaths at 100 m intervals. The abbreviations HNI and PRE refer to Hainan Island and the Pearl River Estuary, respectively. (b) A detailed

bathymetry map of the PRE and nearby waters. In panel (a), observation stations are marked by green and cyan triangles (Zhapo and Qinglan tidal gauge stations), black and blue squares (W1 and W2 wave stations), and a red diamond (M2 station), respectively. In panel (b), stations are represented by: a red triangle (Quarry Bay water level station), a green diamond (M1 station in the PRE), and blue dots (43 cruise survey stations), respectively. The red numbers 1-8 indicate the eight outlets of the PRE, where freshwater and sediment from the Pearl River (specifically the fourth and fifth sediment sizes listed in Table 1 of the main article) are discharged into the estuary.

27. All figures should have the units (for example, figure S7C does not identify the units of SSC mg/L?; Figures S9 and S10 do not provide the units of velocity)

Response:

We sincerely appreciate the reviewer's careful attention to the details in our manuscript. We apologize for the oversight in labeling the units in the supplementary figures. We have now revised the figures as follows:

1.  **Figure S7C:** Added the unit "mg/L" for SSC.

2.  **Figures S9 and S10:** Added the unit "m/s" of velocity.

These corrections have been updated in the revised supplementary materials. Thank you for bringing this to our attention, and we hope the revised version meets the journal's standards.

[Figure]

**Figure S7.** The validations of (a) salinity, (b) temperature, and (c) surface SSC at the 43 stations during the 2017 SYSU cruise survey.

[Figure]

**Figure S9.** (a) Observed and (b) simulated eastward current velocity at M2 station.

[Figure]

**Figure S10.** (a) Observed and (b) simulated northward current velocity at M2 station.

28. The font sizes in many figures is very small and hard to read, especially in the supplement figures.

Response:

We appreciate your feedback. We have carefully revised all figures in the main text and supplementary materials to ensure the font sizes are now clearly legible. Thank you for pointing this out.

**Response to Reviewer 2**

**1. Introduction**

This is my first time reviewing this manuscript, and my assessment has been conducted independently without taking earlier versions and previous referee comments into account.

Using a numerical case study, this paper investigates the dispersal of river-derived sediment over a continental shelf. It does well to provide insights into the spreading of sediment from the river plume over the coastal shelf, taking several processes, such as tides, waves and wind into account. While I think the paper does provide new insights into these main dispersal mechanisms, I think the attempt to form a generalised interpretation is currently not attained, and it might be more appropriate to shift the focus more to the specific case that is investigated. As I do see the value of the developed model and the on/off sensitivity analysis of several model aspects, I think a more careful presentation of the results is required for suitable embedding. Therefore, I recommend publication of the manuscript after major revision.

Dear Reviewer:

We sincerely appreciate the time and effort you devoted to reviewing our manuscript and providing thoughtful and constructive comments. We value your independent assessment and acknowledge the importance of your perspectives in strengthening our study.

We have carefully revised the manuscript in response to your suggestions. Your feedback helped us refine our analysis, clarify our presentation, and better position our work within the specific context of the Pearl River-derived sediment dispersal.

Below, we provide a detailed, point-by-point response to each of your comments, along with the corresponding revisions made to the manuscript.

Thank you again for your valuable insights and support in improving the quality of our work.

**2. General Comments**

**Clarification of "Overestimation" and "Underestimation"**

The manuscript frequently refers to the "overestimation" or "underestimation" of sediment-related quantities (e.g., deposition, resuspension, retention, and bed shear stress) in several instances (e.g., lines 24, 27, 631, 667, 830, 1004, 1011, 1085, among others). These terms imply comparison with real-world field data, which, to my knowledge, is not available for validation (as also noted in lines 805–808). Instead, these quantities appear to be evaluated relative to the "Control" model run. To avoid potential misinterpretation, I recommend revising the phrasing to clarify that differences are being assessed within the model framework rather than against observational data.

Response:

We agree with you that these terms could be misleading without observational validation. We have revised all instances (Lines 24, 27, 631, etc.) to clarify that comparisons are made relative to the Control run (e.g., "higher/lower than the Control case" instead of "over/under-estimation"). This change appears in the revised text with track changes.

**Evaluation of Research Objective (2)**

It is unclear whether research objective (2) (line 156) has been fully achieved. My interpretation is that the model is calibrated based on hydrodynamics to establish the "Control" run, but it remains uncertain whether the model continues to behave as expected when certain processes (NWS, NTS, NAS) are omitted. Additionally, it would be beneficial to clarify whether the chosen methodology is capable of addressing objective (2).

There is potential for deeper insights into the contributions and interactions of different processes in sediment transport. Since sediment transport exhibits nonlinear responses to environmental forcing, an analysis of how tides and waves interact to influence overall sediment dynamics could strengthen the study. Currently, the approach primarily isolates individual processes, but a more integrated interpretation of process interactions could add value.

Some of the study's findings may already be anticipated based on fundamental process-based reasoning. For instance, the following statements confirm well-known expectations:

- Lines 525–527: "With sediment load during winter nearly negligible, the suspended concentration of riverine sediment is significantly lower compared to the wet summer."

- Lines 657–658: "For the NTS case versus the Control case, tides significantly affected bottom stress."

- Lines 667–668: "However, NWS underestimated the nearshore bottom stress."

- Lines 713–714: "This enhanced settling velocity resulted in an increased deposition."

A more nuanced discussion of these results—particularly emphasizing interactions between multiple processes rather than confirming expected trends—would enhance the manuscript.

Response:

We appreciate your insightful suggestion. As you noted, our manuscript has undergone extensive revision, including responses to previous referee comments, so your assessment, while independent of those changes, remains valuable.

Originally, our preprint was titled "Wind and Wave Effects on the Dispersal of Pearl River Derived Sediment over the Shelf" and included detailed momentum-balance and sediment-diagnostic analyses of combined wind, wave, and tidal forcings. However, earlier reviewers found those sections overly technical and recommended focusing instead on seasonal hydrodynamic differences and sediment delivery to distal regions such as the Beibu Gulf. In response, we refocused the paper, now titled "Modeling Dispersal of Pearl River Derived Sediment over the Shelf", to emphasize the Control run's summer and winter sediment distributions across the shelf and the ultimate fate of Pearl River derived sediment after one year. We then designed a series of sensitivity experiments that omitted physical forcings (NWS, NTS, and NAS), adjusted sediment parameterizations, and varied spin-up durations to isolate the individual impact of each process on sediment transport.

You raise an excellent point regarding the nonlinear interactions between tides and waves, which remains a frontier topic in our field. We explored these combined effects in detail in our earlier JGR publication (Zhang et al., 2021), which demonstrated how waves, tides, wind, and freshwater jointly shape both longitudinal and lateral sediment transport and deposition in the Pearl River Estuary. Although a comprehensive analysis of tide and wave interactions lies beyond the streamlined scope of the current study, our present goal was to establish a clear baseline (Objective 1) and then assess how omitting specific forcings or altering sediment characteristics and spin-up durations modifies those baseline results (Objective 2).

To address your comment, we have:

1. Quantified statements that previously confirmed well-known expectations.

2. Revised the Objective (2) to clarify how our methodology evaluates model behavior under different combinations of forcings.

3. Substantially updated the Results and Discussion sections to better explain the impacts of each forcing.

We believe these changes enhance the manuscript's logical flow and align it more closely with its stated objectives, while also laying the groundwork for future investigations into complex process interactions.

**References**

Zhang, G., Chen, Y., Cheng, W., Zhang, H., and Gong, W.: Wave Effects on Sediment Transport and Entrapment in a Channel-Shoal Estuary: The Pearl River Estuary in the Dry Winter Season, Journal of Geophysical Research: Oceans, 126, 10.1029/2020jc016905, 2021.

**Numerical Model Performance and Limitations**

The manuscript could benefit from a more detailed discussion of numerical model performance, particularly regarding:

- Grid sensitivity: While the hydrodynamic validation appears strong, has numerical behaviour and model convergence been tested under grid refinement?

- Model limitations: Section 4.3 primarily discusses the exclusion of certain physical processes but does not address intrinsic model limitations. Consider including a discussion on potential numerical constraints.

- Sigma-coordinates (Line 189): The sigma-coordinate system is known to introduce challenges in accurately modelling salt transport in regions with steep gradients, where Cartesian coordinates may perform better (Bijvelds, 2001). Given the importance of salinity in estuarine turbidity maxima (ETM) formation, it would be helpful to clarify whether ETM development is well captured in the model and whether any limitations arise from the chosen coordinate system.

The study presents interesting insights into sediment budgets and overall sediment dynamics. In particular, the introduction of the "Cycle" model run is a compelling aspect. It may be beneficial to highlight its implications more prominently, especially regarding the conclusions drawn in lines 730–731.

Response:

We sincerely appreciate the reviewer's careful reading and insightful comments. You are right: we inadvertently mischaracterized the coordinate system in the original

manuscript. In fact, we used an S-coordinate system in which cell heights vary vertically to provide increased resolution near the surface and bottom.

We employ the COAWST model, which uses an S-coordinate system in the vertical direction with increased resolution near the surface and bottom layers (Song and Haidvogel, 1994). This vertical layering allows cell heights to vary, enabling finer resolution in dynamically important regions and improving performance in areas with sloping bathymetry compared to traditional sigma-coordinate systems (Bryan, 1969; Song and Haidvogel, 1994). In addition, our model includes horizontal grid refinement in the Pearl River Estuary, enhancing its ability to resolve estuarine features. As a result, the model effectively captures estuarine turbidity maxima and horizontal salinity fronts (Figures S11 and S12; see also in Supplement). During summer, multiple turbidity maxima appear near the estuary bottom (Figure S11b). These features persist in winter but with varying concentrations (Figure S12b), consistent with findings by Wang et al. (2018), Zhan et al. (2019), Zhang et al. (2021), Ma et al. (2022) and Ma et al. (2024). Horizontal salinity fronts shift upstream from high-discharge summer conditions to low-discharge winter conditions (Figures S11e–f and S12e–f), in agreement with previous studies by Zhang et al. (2021) and Ma et al. (2024). Nonetheless, compared with the S-coordinate system, models that employ vertically adaptive layering such as SCHISM (the Semi-implicit Cross-scale Hydroscience Integrated System Model; Zhang et al., 2016) or Cartesian vertical coordinates such as MITgcm (the MIT General Circulation Model; Marshall et al., 1997a, 1997b) generally perform better in regions with steep topographic gradients (Bijvelds, 2001). Therefore, future research could

benefit from adopting models with higher horizontal resolution and Cartesian vertical coordinates to improve the simulation of Pearl River-derived sediment dynamics across the estuary and adjacent shelf.

The text has been revised to accurately describe the vertical grid configuration as follows: "The vertical grid uses a terrain-following S-coordinate system (Song and Haidvogel, 1994) with 20 layers and a stretching transformation for higher resolution near the surface and bottom." We regret any confusion caused by this oversight and thank the reviewer for bringing it to our attention.

The "Cycle" model run, which examines the effect of spin-up duration, demonstrates that Pearl River–derived sediment entering and accumulating in various regions of the model domain during the first year continues to migrate southwestward into the second year. This migration is driven by the annually averaged net alongshore current, which remains predominantly directed toward the southwest. The current becomes stronger during the winter monsoon under the influence of prevailing northeasterly winds, whereas the opposing summer southerly winds are comparatively weaker.

[Figure]

**Figure S11.** Summer-averaged (a–b) total suspended sediment concentration (classes 1–5 in Table 1) and the black lines mark the 100 mg L$^{-1}$ contours, (c–d) Pearl River–derived suspended sediment concentration (classes 4–5 in Table 1), and (e–f) horizontal salinity gradient magnitude (SGM) and the black lines mark the 1.5 psu km$^{-1}$ contours. Columns 1 and 2 represent surface and bottom layers, respectively.

Dry winter season (2017.10.1-2018.3.31)

[Figure]

**Figure S12.** Same as Figure S11, but for winter-averaged ones.

**3. Technical comments**

The level of significance reported for measured percentages is inconsistent across the manuscript. Given the model's inherent (in)accuracy, the precision of percentage values should be adjusted accordingly. This applies particularly to the Abstract, Section 3.2, and Section 5 (Conclusions).

Response:

Thank you for the valuable suggestion. In response, we have revised the numerical expressions in the Abstract to better reflect the model's inherent uncertainty. Specifically, we now use approximate language such as "approximately two-thirds" to convey key findings without implying excessive precision. In the main text, we have also added clarifying statements to emphasize that the reported percentages are model-derived estimates based on specific simulation conditions. These changes are intended to prevent any potential misinterpretation of the results as overly precise or directly comparable to observational data.

Line 192: Song and Haidvogel (1994) introduce a new s-coordinate system, not the general sigma-coordinate system introduced in the manuscript. It is unclear which system is actually used and what the two θ-values are, as these are not explicitly described in Song and Haidvogel (1994). Please clarify and revise accordingly.

Response:

We sincerely appreciate the reviewer's careful reading and insightful comment. You are right; we inadvertently mischaracterized the coordinate system in the original

manuscript. In fact, we used an S-coordinate system. The text has been revised to accurately describe the vertical grid configuration as follows: "*The vertical grid used a terrain-following Sigma S-coordinate system (Song and Haidvogel, 1994) with 20 layers and a stretching transformation for higher resolution near the surface and bottom.*" We regret any confusion caused by this oversight and thank the reviewer for bringing it to our attention.

Lines 303-305. The statement "The realistic spin-up greatly reduced the irregularities and prepared a more suitable seabed sediment particle size distribution field for subsequent simulations than the initial prototype (Figures 2d-f vs. 2g-i)." needs further clarification. Could you elaborate on why this "realistic reworking" is necessary and why it results in a more suitable initial sediment distribution? The figures suggest the presence of spurious oscillations in the model interior and near the open southern boundary, which raises concerns.

Response:

Thank you for the helpful comment. We agree that further clarification of the spin-up procedure is warranted. The "realistic spin-up" refers to a 15-month model integration using the coupled hydrodynamics (ROMS), wave (SWAN), and sediment transport (CSTM) components. This process allows the initial, idealized sediment distribution to evolve under realistic dynamic forcing, including tides, waves, and currents, thereby minimizing artificial gradients or unrealistic spatial patterns that may arise due to limitations in the number, representativeness, and timing of field sediment

sampling relative to the model start date. As a result, the sediment field after spin-up (Figures 2g to 2i) exhibits spatial patterns that are more physically plausible and better aligned with the hydrodynamic context of our study region. In response to the reviewer's concern about spurious features, the observed patterns are indeed located near the open southern boundary, where water depths are relatively large. These areas lie well outside our primary region of interest—the Pearl River Estuary and its adjacent continental shelf. As shown in Figure 1 and Figure S1a, the deep bathymetry in this boundary zone can occasionally amplify numerical noise. However, these features are geographically distant and exert minimal influence on sediment dynamics within our focus area. The modeled grain size distributions and bottom shear stress fields across the core domain remain robust and unaffected by these peripheral boundary effects.

To improve clarity, we have revised the manuscript as follows:

*The initial prototype field underwent a 15-month spin-up period (from January 1, 2016, to March 31, 2017), during which the bottom sediment composition evolved under realistic hydrodynamic forcings from the ROMS, SWAN, and CSTM models. This method has been utilized in numerous previous studies, including those by Bever et al. (2009), van der Wegen et al. (2010), and Zhang et al. (2021). This process allows the initially idealized sediment distribution to evolve under realistic dynamic forcings, including tides, waves, and currents, thereby minimizing unreasonable spatial patterns introduced by the Kriging interpolation method. Such unreasonable spatial patterns may arise due to limitations in the number, representativeness, and timing of field sediment samples relative to the model start date. As a result, the sediment field after*

*the spin-up period (Figures 2g–i) exhibits spatial patterns that are more physically*

*plausible and better aligned with the hydrodynamic conditions of the study region.*

Line 483-487. The precise definition of a "river plume" is unclear. How is it quantitatively defined, and what correlation does it have with regions of high suspended sediment concentration (SSC)? Additionally, a more in-depth analysis of the freshwater plume's role in transporting riverine sediment—relative to the overall sediment transport—would strengthen this section. Could you expand on this aspect?

Response:

Thank you for this insightful comment. In our study, we define the river plume based on surface salinity, using a threshold of 33.5 psu to delineate the plume boundary. This definition is strictly salinity-based and independent of suspended sediment concentration (SSC). However, there is indeed a seasonal correlation between the river plume and high surface SSC. During the summer, when river discharge is high and water column stratification is strong, surface SSC is primarily influenced by advection from the buoyant river plume. In these conditions, high SSC regions closely align with the freshwater plume, as sediment is efficiently transported by the low-salinity, high-momentum freshwater outflow. In contrast, during winter, when river discharge is low and vertical mixing is more intense, the correlation between the plume and SSC is much weaker. In this season, SSC is largely governed by resuspension processes driven by strong winds and waves, rather than by freshwater transport. We have added clarification on this point in the revised manuscript and expanded the discussion of the

river plume's seasonal role in sediment transport to better reflect its relative contribution across different regimes.

Line 493: Please explain how the sediment flux is determined. Is it computed purely as advective horizontal transport, or are other processes included? Please specify.

Response:

Thank you for the insightful comment. We have revised the main text to clarify this point. The sediment flux in our study refers specifically to the depth-integrated horizontal advective transport of riverine sediment. It is calculated as the product of the horizontal velocity field and the suspended sediment concentration, integrated over the water column. This calculation does not include vertical fluxes or other processes such as settling or resuspension, which are accounted for separately within the model framework. The clarification has been added to the manuscript for transparency.

Line 541. Figure 6 shows significant salinity gradients near the estuary, which suggests the potential formation of an estuarine turbidity maximum (ETM). Is such an ETM captured in the model? If so, could you comment on its role in sediment transport within the system?

Response:

Thank you for this insightful comment. As you noted, our model has been well validated and demonstrates good skill in reproducing suspended sediment concentrations. Although the primary focus of this study is on the dispersal of Pearl

River-derived sediment over the continental shelf, the model does indeed capture the formation of estuarine turbidity maxima (ETMs) within the estuary. Recent studies by Ma et al. 2024 have thoroughly investigated the seasonal evolution and spatial connectivity of multiple ETMs in the PRE. While our original intention was not to revisit those previously addressed topics, your comment highlights an important aspect of the sediment dynamics. In response, we have now included the modeled ETM results in the Supplement and have cited the relevant literature by Ma et al. (2023, 2024) to provide proper context. Additionally, we have revised the manuscript and Supplement to reflect these updates.


[Figure]

**Figure S11.** Summer-averaged (a–b) total suspended sediment concentration (classes 1–5 in Table 1) and the black lines mark the 100 mg L$^{-1}$ contours, (c–d) Pearl River–derived suspended sediment concentration (classes 4–5 in Table 1), and (e–f) horizontal salinity gradient magnitude (SGM) and the black lines mark the 1.5 psu km$^{-1}$ contours. Columns 1 and 2 represent surface and bottom layers, respectively.

Dry winter season (2017.10.1-2018.3.31)

[Figure]

**Figure S12.** Same as Figure S11, but for winter-averaged ones.

Line 639: the term "probably being artefacts" is vague. Could you clarify what kind of

numerical artifacts these are? While they appear as local deviations in deeper regions,

they span several kilometres and multiple grid cells. Given their scale, could they significantly influence hydrodynamics across the domain? Furthermore, how does this uncertainty impact confidence in the NAS model results? A more detailed explanation would be helpful.

Response:

Thank you for your thoughtful comment. We agree that our previous wording may have been unclear. We have revised the text for clarity. The regions showing abnormal values are not located on the continental shelf, but rather in the deeper areas near the southern boundary of the model domain, where water depths exceed 200 m. As such, the changes in bottom stress observed in these localized zones, although spanning several kilometers and grid cells, occur far from the areas influenced by Pearl River-derived sediment. Therefore, they do not significantly impact the nearshore or shelf-scale sediment dynamics that are the focus of this study.

Additionally, we acknowledge that the term "overestimations" may misleadingly imply comparison with observational data, which is not available for bottom stress validation in this context. To avoid misinterpretation, we have revised the phrasing to clarify that the differences are relative to the Control model run rather than against field observations. The revised text now reads:

*Some pronounced deviations are noted in localized deeper areas near the southern boundary of the domain (Figures 8e-f). These deviations, likely arising from boundary condition effects, are situated far from the Pearl River-derived sediment distribution areas (Figures 5-6). Consequently, they do not influence the dynamics of the Pearl*

*River-derived sediment transport over the continental shelf (Figures 8e-f).*

Line 694+733. Figures 9 and 10 include arrows that are somewhat unclear. While the colours indicate differences between model runs, it appears that the arrows represent only the non-Control run. Could you clarify their specific purpose and whether they provide a direct comparison with the Control case?

Response:

Thank you for your comment. Specifically, Figures 9 and 10 present the seasonal surface currents and the differences in suspended sediment concentration (SSC) between the control and sensitivity runs. The colors indicate riverine SSC differences between model runs, while the arrows represent surface currents from the non-Control runs. We need to show both the circulation patterns in each experiment and the differences in riverine SSC relative to the Control run. Because the experiments in Figure 10 use the same hydrodynamics as the Control run, comparing Figures 9 and 10 reveals the circulation differences between NTS, NWS and NAS and the Control case. This presentation style has been used in previous studies. For example, Xue et al. (2012) illustrated current velocity and SSC differences in their Figures 5 and 6 by comparing experimental cases with the Control run on the Mekong River Shelf. We fully acknowledge your concern and have therefore enlarged the arrows in Figures 9 and 10 to more clearly depict the flow fields in each case. This adjustment improves the clarity of the figures and enhances the overall coherence of the descriptions.

Lines 728-731 describe the deposition seen in Figure 11f but provide no reason for it by the driving hydrodynamics. To me, it is unclear why the sediment distribution would change. What is the process behind this? Does this not provide us with the conclusion that a more complete sediment spin-up, as performed in Cycle, is necessary for robust results?

Response:

Thank you for the thoughtful comment. We agree that the driving hydrodynamic processes behind the sediment redistribution shown in Figure 11f were not clearly explained in the original text. In the Cycle experiment, the initial conditions for the five sediment classes were established using the Class 1–5 sediment suspensions and depositions from the end of the Control run on March 31, 2018, thereby initiating a second control simulation. The new riverine sediment input and its transport processes during the Cycle experiment are nearly identical to those in the Control run. Therefore, compared to the Control run, the Cycle experiment specifically focuses on evaluating the impact of the presence of pre-existing Pearl River-derived sediments on estimates of riverine SSC and the annual seabed riverine sediment budget in the second year.

To clarify, the observed changes are primarily driven by the annually averaged net alongshore current, which remains predominantly southwestward. This current intensifies during the winter monsoon, dominated by northeasterly winds, while the

summer southerly winds are comparatively weaker. Consequently, sediments deposited during the first year are resuspended and transported farther southwestward during the second year. Thus, the sediment distribution changes observed in the Cycle experiment are attributable to this annually persistent, seasonally modulated hydrodynamic forcing. Furthermore, we fully agree with the reviewer's observation that these results emphasize the importance of incorporating pre-existing sediments through a more comprehensive spin-up process, as demonstrated in the Cycle experiment. Excluding previously deposited sediments may constrain the model's capacity to represent long-term sediment transport and accumulation. We will revise the manuscript to clearly describe these processes and explicitly link the hydrodynamic drivers to the sediment redistribution shown in Figure 11f.

Line 793. The caption of Figure 12 explains the significance of magenta values, but it is unclear what the other values (black, blue, red) represent. Could you specify their meaning?

Response:

Thank you for pointing this out. We acknowledge that the previous caption was not clearly written, and we have now updated it for better clarity. Additionally, we have revised Figure 12 by removing the separate percentage labels for sediment Class 4. Readers can now derive the Class 4 retention percentage by subtracting the Class 5 percentage (blue) from the combined Classes 4 + 5 percentage (black). We believe this adjustment streamlines the figure while still allowing readers to access all necessary

information. The revised Figure 12 and its caption now read:

[Figure]

**Figure 12.** Same as Figure 7c, but for the other six cases: (a) NTS, (b) NWS, (c) NAS,

(d) NVS, (e) DSV, and (f) Cycle, respectively. All percentages shown in the figure are

expressed relative to the annual riverine sediment load (see Fig. 3a). Magenta values

denote the differences in retention percentage of riverine sediments (Classes 4 and 5;

Table 1) between the Control run and each sensitivity case. Black values represent the

combined retention of Classes 4 + 5, while blue values indicate Class 5 alone. To obtain

the retention percentage for Class 4, simply subtract the Class 5 percentage (blue) from

the combined Classes 4 + 5 percentage (black).

**4. Textual remarks**

Line 25: The term "distal retention" may not be appropriate in this context. If sediment is never transported to distal regions in the first place, it does not necessarily imply a lower distal retention. Consider rephrasing for clarity.

Response:

Thank you for pointing this out. We agree that the term "distal retention" may have caused confusion in this context. We have revised the sentence to remove this ambiguous expression and improve clarity.

Line 219: The abbreviation Hsig appears without an introduction. If it refers to significant wave height, please define it properly when first mentioned.

Response:

Thank you for pointing this out. We have revised the text to properly define the abbreviation. The updated sentence now reads: "*This exchange included significant wave height (Hsig), surface peak wave period, mean wave direction and length, wave energy dissipation, and the percentage of breaking waves from SWAN to ROMS, as well as water level and current from ROMS to SWAN.*"

Line 298: I am not sure what "realistic reworking" means, could you explain or rephrase?

Response:

Thank you for the comment. By "realistic reworking," we refer to the natural redistribution of initially idealized sediment distribution through physical processes

such as tides, waves, and currents during the spin-up period. This reworking allows the sediment field to adjust to the prevailing hydrodynamic conditions, resulting in spatial patterns that are more physically plausible and representative of natural sediment dynamics. To improve clarity, we have revised the manuscript as follows:

*The initial prototype field underwent a 15-month spin-up period (from January 1, 2016, to March 31, 2017), during which the bottom sediment composition evolved under realistic hydrodynamic forcings from the ROMS, SWAN, and CSTM models. This method has been utilized in numerous previous studies, including those by Bever et al. (2009), van der Wegen et al. (2010), and Zhang et al. (2021). This process allows the initially idealized sediment distribution to evolve under realistic dynamic forcings, including tides, waves, and currents, thereby minimizing unreasonable spatial patterns introduced by the Kriging interpolation method. Such unreasonable spatial patterns may arise due to limitations in the number, representativeness, and timing of field sediment samples relative to the model start date. As a result, the sediment field after the spin-up period (Figures 2g–i) exhibits spatial patterns that are more physically plausible and better aligned with the hydrodynamic conditions of the study region.*

Line 495: The sediment flux is not strictly westward and eastward. Please consider rephrasing for accuracy.

Response:

Thank you for the suggestion. We agree that the original wording was not sufficiently accurate. We have revised the sentence to: "*The riverine sediment exhibits both southwestward and northeastward fluxes (Figure 5e).*"

Line 502 and 504. The units used for sediment flux are not correct -> g/m/s

Response:

We appreciate your careful review. This was indeed a typographical error, and we have now corrected it accordingly.

Line 524: The term "salinity front" is somewhat vague. Does this refer to a specific isohaline? Please clarify.

Response:

Thank you for the comment. We agree that the term "salinity front" could be more clearly defined. We have revised the sentence for clarity as follows:

*The expansion of the Pearl River buoyant plume is constrained to the southwestward direction by strong northeasterly winds (Figure 6a), resulting in a narrow cross-shore width of the buoyant plume and the formation of a strong horizontal salinity gradient (i.e., a salinity front, particularly within the 30–33.5 psu range shown in Figure 6a) outside the estuary (Figure 6a). Flow velocity increases near this salinity front, facilitating the westward extension of the buoyant plume through the Qiongzhou Strait into the "Gulf" region.*

Line 542. Figure 6e: "2D riverine Flux" is unclear phrasing to me. Perhaps "Horizontal sediment flux" would be a more precise alternative, as "riverine" may be ambiguous without additional context..

Line 542. Figure 6f. The term "riverine Deposition reworking" is unclear. Consider rephrasing to include "difference," "change," or another term that better captures the intended meaning.

Response:

Thank you for the helpful suggestions. In response, we have revised the subplot titles in Figures 5e and 6e from "2D riverine Flux" to "Horizontal riverine sediment flux" for improved clarity and precision. The figure captions have also been updated to specify that this refers to the depth-integrated horizontal flux of riverine sediment (corresponding to classes 4 and 5 in Table 1). Additionally, the title in Figure 6f has been changed from "riverine Deposition reworking" to "Riverine sediment deposition change" to more accurately reflect the intended meaning.

[Figure]

**Figure 5.** Patterns averaged over the entire wet summer season in the Control case: (a) surface and (b) bottom salinity (color, psu) and flow (arrows, m s$^{-1}$); (c) surface and (d) bottom riverine (classes 4 and 5 in Table 1, as follows) SSC (mg L$^{-1}$); (e) depth-integrated horizontal riverine sediment transport rate (color, g m$^{-1}$ s$^{-1}$) and direction (arrows); and (f) riverine sediment deposition thickness (mm) on the seabed during the wet summer season. Flow vectors in regions with water depths exceeding 100 m are masked for clarity.

[Figure]

**Figure 6.** Same as Figure 5, but for the dry winter season in the Control case. Notably, (f) illustrates the changes in riverine sediment deposition (classes 4 and 5 in Table 1) on the seabed at the end of the dry winter season compared to the end of the wet summer season.

There are some inconsistencies in terminology throughout the manuscript. For example, "Distal" regions is sometimes written in quotation marks, while in other places it appears as distal regions without emphasis. Similarly, "Gulf" regions and gulf regions are used interchangeably. Please ensure consistency in terminology throughout the text.

Response:

Thank you for pointing out the inconsistency in terminology. We have carefully reviewed the manuscript and standardized the usage by consistently placing terms such as "Distal" regions and "Gulf" regions in quotation marks throughout the text for clarity and emphasis.

---

## Author Response (AR3)

Dear Dr. Zhang and co-authors,

We have now received two reviews for your revised manuscript. Referee 3 is satisfied but has some final comments. Referee 2 states that the manuscript is closer to publication but thinks it still needs some work. The referee has listed very helpful comments which can further improve the manuscript.

I agree with the referees that publication has come closer. However, the manuscript is not there yet. Please consider the comments by the two referees and submit a revised manuscript accounting for these comment and suggestions.

I will make my decision after this round of reviews and revisions.

With best wishes

Mario Hoppema

- editor

**Dear Dr. Hoppema,**

Thank you for your continued support and the opportunity for us to revise and resubmit our manuscript.

We have carefully addressed all comments from both reviewers, corrected previous issues, and made substantial improvements. A detailed point-by-point response is provided in the accompanying letter.

We appreciate your time and consideration, and look forward to your further evaluation.

**With best regards,**

Wenping Gong

**Response to Referee 2 (Report #2)**

I have read the responses to reviews of the resubmission, and the revision. The authors have done a conscientious job of revising the manuscript to address concerns and the paper is improved. This submission seems closer to the standard expected for your journal, but in my opinion is not there. While the paper is improved, it suffers from being almost entirely focused on the numerical model with less attention paid to the process-based implications of the results. The organization of the paper should be improved to increase clarity and reduce redundancy.

Response:

Thank you for your continued review and for acknowledging the improvements in our revised manuscript. We appreciate your comment that the paper is now closer to the journal's standard.

In this revision, we have further improved the clarity and structure of the manuscript by reducing redundancy and enhancing the organization. While our study is based on a numerical model, we agree that emphasizing the process-based implications is important. We have adjusted the text accordingly to better reflect what the results reveal about sediment transport processes on the northern South China Sea shelf.

We hope these revisions address your concerns and improve the overall impact of the paper.

General Comments:

1. The paper would be more impactful if it emphasized more the lessons learned about

dispersal processes, rather than emphasizing the work of modeling dispersal.

Response:

Thank you for the insightful suggestion. While previous studies on the long-term distribution of Pearl River-derived sediments have primarily relied on seismic profiles, sediment cores, and radiometric dating methods, numerical investigations of their transport processes remain limited, particularly in terms of capturing the seasonal variability and spatial connectivity of cross-shelf and along-shelf sediment dispersal, despite the widespread use of numerical modeling in sediment dynamics research.

Our study aims to address this gap by employing a well-calibrated and validated numerical model, supported by observational data, to investigate sediment dispersal processes. We agree that geological methods provide valuable insights into long-term sediment deposition patterns, but we also believe that numerical modeling serves as a powerful and complementary tool for examining the dynamic mechanisms that govern sediment transport across a range of spatial and temporal scales.

1a. This is reflected in the current title of the paper which focuses on the act of implementing a model; not the process-based results derived from the model, or the timescales covered. Is that appropriate for this journal?

Response:

Thank you for the valuable comment. We have revised the manuscript title to better reflect the focus on process-based results and timescales. The new title is: "Physical Drivers and Parameter Sensitivities of Pearl River-derived Sediment Dispersal on the

Northern South China Sea Shelf: A modeling study".

1b. In comparing the cases studies to the control run: the paper presents these in terms of the act of modeling dispersal instead of what is learned about processes via the comparison. For example, in the abstract (lines 27 – 29) "the absence of tidal forcing reduces bottom shear stress…"; (lines 32 - 33) when "ambient currents" are "omitted". These describe what happens in the model when forcings are removed from the model. The paper would have more impact if it emphasized what these results show about the importance of tides and large-scale currents in dispersing sediment. These are two examples from the abstract, but the paper takes this approach for all the comparisons of case studies to the control run.

Response:

Thank you for the constructive feedback. We have revised the abstract and relevant sections throughout the manuscript to place greater emphasis on the physical insights gained from the case comparisons, highlighting the roles of tides, waves, and large-scale circulation in sediment dispersal, rather than focusing on model setup or scenario differences.

1c. For the "cycle experiment", the results show that riverine sediment continues to be dispersed at timescales that exceed 1 year. This is not surprising for such a large-domain model. The analysis does not provide insight into the actual residence times expected.

Response:

Thank you for the comment. While it is indeed expected that sediment dispersal can extend beyond one year in a large-domain model, we believe it is still necessary to explicitly simulate and analyze this process. Our "cycle experiment" reveals not only that Pearl River-derived sediment continues to disperse after one year, but also that the transport direction remains predominantly southwestward. In particular, sediment deposited in the Beibu Gulf is shown to continue moving southward, which offers additional understanding of the longer-term redistribution of riverine material.

We agree that quantifying sediment residence times is a distinct and important topic. However, it is beyond the scope of the present study, which focuses on identifying the physical drivers, sediment properties, and sediment conditions that influence the along- and cross-shelf transport of Pearl River-derived sediments. We consider residence time analysis a promising direction for future work and potentially a separate publication.

1d. Some model experiments focus on forcing (tides, waves, large-scale currents), and others on sediment properties (critical shear stress, settling velocity, seabed grain sizes). The analysis is presented in terms of model inputs (if tides are omitted, then sediment is retained in PRE, etc.); rather than what can be learned from the experiments. The paper would be more impactful if it phrased results in terms of the relative impacts of different forcings (for example: which has more effect, the tides or the waves?); and relative impacts of uncertainty in sediment properties.

Response:

Thank you for the constructive suggestion. We fully agree that emphasizing the relative impacts of different physical drivers and sediment property uncertainties strengthens the interpretation of the results. In fact, our analysis already highlights these aspects.

We have shown that tidal forcing is the dominant driver of sediment dynamics within the estuary, while wave forcing plays a more significant role at the river mouth and along adjacent coastlines. In contrast, large-scale background circulation primarily influences the eastward dispersal of sediments offshore during the summer. These region-specific impacts of physical drivers are explicitly discussed in the revised manuscript to further clarify their relative importance.

Regarding sediment property uncertainties, we have already addressed this point in Section 4.1. Our results indicate that variations in settling velocity have a stronger influence on sediment distribution than the seasonal variation in critical shear stress for erosion. Additionally, changes in seabed sediment grain size distributions have a relatively minor effect on the overall dispersal pattern of Pearl River-derived sediments. We also find that the timescale of analysis, whether seasonal or extending beyond one year, has a more substantial impact on the shelf-wide distribution of sediment than the uncertainty in seabed properties. These conclusions have been further emphasized in the revised discussion.

1e. Finally, the conclusions (Section 5) are very focused on modeling rather than

lessons learned regarding the system as a whole. This makes the paper less impactful than it could be.

Response:

Thank you for the helpful suggestion. We have revised the Conclusions (Section 5) to place greater emphasis on the key physical insights gained from the study, rather than focusing solely on the modeling approach.

2. Comparison of 1-year model to Holocene records (Pages 51, etc). It is interesting to compare patterns calculated by the model for 1-year to the longer-term geologic record; but the paper would be more impactful if the differences in the timescales were acknowledged. Also, the paper needs to be clear about whether (or how) the geologic data were used to calibrate sensitive model parameters (settling velocity).

Response:

Based on our calculated Pearl River-derived sediment thickness, and considering that the Holocene sediment load is 2.61 times that of the modeled year, we estimated the long-term accumulation by multiplying the number of Holocene years by 2.61 and the modeled annual sediment thickness. This approach is straightforward. We also mention that the calibration of SSC, including associated sediment properties such as settling velocity, was conducted without any reference to geological data.

2a. Text starting line 929 should begin a new paragraph. This compares the 1-year model results to Holocene sediment thicknesses (6,500 years). Were the sediment

settling velocities "calibrated" to match this result? If so this should not be cited as confirmation that the partitioning is valid.

Response:

Thank you for the comment. We would like to clarify that the sediment settling velocities, along with other sediment characteristics associated with SSC, were calibrated independently of any geological data. At no stage did we adjust or tune the model parameters to match the Holocene sediment thicknesses. Therefore, this comparison was not used as a basis for model calibration, and we believe it provides an independent line of support for the modeled sediment partitioning.

2b. It is not clear how the values in lines 947 – 950 were scaled to bridge the large differences in timescales.

Response:

Thank you for the comment. As noted, the Holocene sediment load is estimated to be 2.61 times that of the modeled year. To account for the timescale difference, we multiplied the modeled annual sediment thickness by both the number of Holocene years and the 2.61 scaling factor. This provided an estimate of the long-term accumulation, allowing for a direct comparison of spatial patterns.

3. The parametrization of the sediment model remains confusing, even after revision. Part of the confusion comes from the fact that the paper is not well organized and not concise. Topics are repeated in different parts of the paper and sometimes seem to

Response:

Thank you for your valuable feedback. We acknowledge that the organization and clarity of the sediment model parametrization could be improved. In response, we have carefully revised the manuscript to enhance its structure, reduce redundancy, and ensure consistency throughout. We have also clarified key points to avoid any contradictions. We hope these changes have improved the readability and understanding of the sediment model parameters.

3a. For example: the setting of riverine sediment properties includes the choices of 60%/40% split and settling velocities (0.005 and 0.6 mm/s). These are said to follow Bever and MacWilliams (line 350); Zhang et al 2019 (line 359, 388). Then, line 366 lists 6 source papers plus "model calibration" but does not refer to the Supplement. Line 897 attributes the choice of settling velocities to Xia et al. and calibrations and cites the Supplement. The Supplement shows modeled and observed SSC but does not mention calibration of ws.

Looking through the source papers, the 60/40 split is from Zhang et al. 2019. The choice of settling velocities seem to be calibrated but is not clearly described. The authors need to be concise and clear about selection of these important values.

Response:

Thank you for the detailed comment. Xia et al. (2004) conducted in situ observations of settling velocities for suspended sediments in the Pearl River Estuary and reported

a range of values for different grain sizes. These ranges were used to guide extensive sensitivity experiments and model calibration. Through this process, we selected representative fixed settling velocities that allowed the model to reproduce observed SSC values reasonably well. This calibration approach is a standard and widely accepted practice in sediment transport modeling.

As we have acknowledged in Section 4.1, the uncertainty in settling velocities, especially for slow-settling fine particles, can lead to variability in the model results. Nevertheless, we have made great effort to constrain the values within reasonable bounds based on both observations and model performance.

The 60%/40% class split is necessary, as the model requires an assumed distribution. This ratio was adopted from Zhang et al. (2019), where the median grain size of suspended sediments in the estuary was around 8 μm, making the 60/40 partition a reasonable approximation. As noted in Section 4.1, we have also discussed the implications of this choice and confirmed that it is within a plausible and representative range for the system.

3b. Another example is treatment of riverine sediment for the spin-up and cycling cases. The authors revised this but not achieved clarity and conciseness. At Line 351, the paper states that after the spin-up class 4 and 5 sediment were "added as class 1 and 2" (do the authors mean "added to class 1 and 2"?). Then in line 464 the "Cycle experiment" is described in a confusing way. After several readings: I think that the difference between the Control Run and the Cycle experiment is whether the Class

4&5 sediment from the "spin-up" period was added to Class 1&2 in the initial bed before the model run.

Response:

Thank you for pointing out the typo. Our original intention was indeed "added to class 1 and 2," and we have corrected this in the manuscript.

Regarding the description of the Cycle experiment, as stated in the revised manuscript:

"Finally, to assess the model's sensitivity to the spin-up duration of Pearl River-derived sediment, particularly regarding the retention of riverine sediments in both the water column and the seabed, we adopted the sediment distributions (Classes 1 to 5) from the final state of the Control run on March 31, 2018, as the alternative initial conditions for the Cycle experiment (designated as Exp 7, Cycle hereafter). This setup carries over the full year's evolution of riverine sediment transport and deposition from the Control run (Exp 1), including changes in all sediment classes, into the start of Exp 7. As a result, Exp 7 mainly evaluates how the presence of previously deposited riverine sediments influences subsequent sediment transport estimates."

Thus, the primary difference between the Control Run and the Cycle experiment is that the Cycle experiment includes an additional year of accumulated Pearl River sediment.

4. Section 4.2 is improved but the writing is still a bit rough. The paper does not

provide the context of Walsh and Nittrouer's classification scheme. Lines 976 and 986 seem contradictory (line 976 says your results are "unlike" Walsh & Nittrouer's framework, whereas Line 9787 says your results are "consistent").

Response:

Thank you for the insightful comment. To clarify, our findings demonstrate that most results are consistent with the hierarchical decision tree proposed by Walsh and Nittrouer (2009), except for the application of the 2 megatons per year threshold for riverine sediment discharge. While the Pearl River's sediment discharge exceeds this threshold, most sediment remains deposited near the estuary, indicating an estuarine accumulation-dominated (EAD) system. This behavior differs from the predictions of Walsh and Nittrouer's hierarchical decision tree.

We acknowledge that the original wording was somewhat rough and potentially confusing. We have revised this section to better contextualize Walsh and Nittrouer's classification scheme and clearly explain the apparent inconsistency.

5. Page 58 and 59 use a lot of text to argue for more sensitivity tests to better constrain sediment dispersal. This can probably be shortened. But more importantly, the argument should be that more DATA is needed. Without additional data streams, there seems little argument for doing more model runs. The paper might be more impactful if the authors provided insight into what type of data would be most useful.

Response:

Thank you for the valuable comment. We agree that additional observational data are

essential for improving model constraints and advancing understanding of sediment dispersal. In response, we have revised the relevant section (Pages 58 and 59) to shorten the discussion on sensitivity tests and to emphasize the need for more data. We now highlight the importance of direct measurements, such as settling velocities and erosion thresholds under varying hydrodynamic conditions, as particularly useful for better parameter calibration and model validation.

Specific Comments (by line number in the preprint)

6. Line 32, Lines 452, 453, 471: use of the term "Ambient circulation" is not clear. You need to define what you mean by this (large, non-local scale forcing?).

Response:

Thank you for pointing this out. We have revised the text to clarify the meaning of "ambient circulation." It now refers specifically to the remotely forced ambient shelf current and residual (non-tidal) water levels, which are driven by large-scale, non-local forcing.

7. Line 44 (abstract): use of the word "dispersal mechanisms" seems a mis-statement. "Dispersal mechanisms" are things like bedload, suspended load, gravity flows. This paper looks at the sensitivity of suspended transport and dispersal to forcings and parameters.

Response:

Thank you for the clarification. We agree with your observation and have revised the

text accordingly. The term "dispersal mechanisms" has been replaced to better reflect our focus on the sensitivity of suspended sediment transport and dispersal to physical forcings and sediment-related parameters.

The updated sentence now reads: "*Overall, this study reveals the transport pathway and fate of the Pearl River-derived sediment and provides a model-based assessment of its seasonal behavior and the sensitivity of suspended sediment dispersal to physical drivers and sediment parameters or conditions on the northern SCS shelf.*"

8. Introduction seems a bit long and could be shortened. Some of the text seems too specific and is redundant with later methods (for example lines 159 – 163).

Response:

Thank you for your valuable suggestion. We have shortened and streamlined the Introduction by removing overly specific details and reducing redundancy with later sections, enhancing clarity and focus.

9. Paragraph starting at line 328: Revise for clarity. Suggest (line 333) "initially estimated sediment distribution to evolve…"; (line 335) "introduced by Kriging, sparse or problematic data"; (line 338) "after the spin-up period is thought to exhibit spatial patterns that are better aligned…".

Response:

Thank you for the helpful suggestions. We have revised the paragraph accordingly to improve clarity, incorporating the recommended wording where appropriate.

10. Lines 406 – 419: this paragraph provides information about the climatology of summer conditions. It is unclear what these values represent. Are these values climatological averages over several years? (if so, data sources should be cited and the time-period noted). Or are they the values from the 1-year period modeled?

Response:

Thank you for the comment. The values presented are not climatological averages. As noted in the first paragraph of Section 2.5, all values correspond to the period from April 1, 2017, to March 31, 2018, which matches the 1-year duration of the model simulation. The data sources used for these values are also clearly stated in the manuscript.

11. Section 3.2, Page 33: I found this description confusing.

Response:

Thank you for the helpful comment. We agree that the original description could have been clearer and have made appropriate revisions to improve readability and clarity. The revised text is as follows:

"*We present the sediment fluxes and retention amounts in different regions. Figure 7a-c illustrates the proportion of riverine sediment retention budget within each region, expressed as a percentage of the total annual river sediment load input (Figures 3a), for the wet summer season (Figure 7a), the dry winter season (Figure 7b), and the entire year (Figure 7c), based on the Control run, respectively.*

*Meanwhile, Figure 7d illustrates the annual deposition over the shelf.*

*The retention of Pearl River sediment on the continental shelf exhibits significant seasonal variations (Figures 7a-c). During the wet summer (characterized by high discharge and relatively calm wind/waves), the PRE and continental shelf receive 95.17% of the annual sediment input (Figures 3a and 7a). Of this, about two-thirds is retained in the "Proximal" region (Figure 7a). Influenced by the prevailing southerly winds and northeastward shelf currents, 13.01% of the annual sediment load is retained in the "Eastern" and "Southeastern" regions (Figure 7a). Meanwhile, the shelf west of the PRE (⑤-⑧ regions) retains 15.87% of the annual load, with the "Western" region alone accounting for 8.48% (Figure 7a). In contrast, only 0.92% and 2.3% enter the more remote "Gulf" and "Distal" regions, respectively (Figure 7a). The "Southern" region retains a mere 1.22% of the sediment (Figure 7a).*

*In the dry winter (characterized by low discharge and energetic winds/waves), the PRE and the continental shelf receive only 4.83% of the annual sediment load (Figures 3a and 7b). The sediment distribution during this season primarily reflects reworking of previously retained sediments from summer (Figure 7b). Retention in the "Proximal" region increases slightly (+1.38%)in retention, while retention decreases in the ②-⑥ regions. Much of this remobilized sediment is transported farther offshore and retained in the "Gulf" and "Distal" regions (Figure 7b).*

*The annual sediment budget reveals that 66.45% of the Pearl River sediment is retained in the "Proximal" region (Figure 7c). Additionally, 9.2% is retained in the "Eastern" and "Southeastern" regions (Figure 7c), primarily during summer (Figures*

*7a vs. 7c), while 24.12% is retained on the shelf west of the PRE (⑤-⑧ regions), with most of that occurring in the "Gulf" and "Distal" regions during winter (Figures 7b vs. 7c).*

*The annual deposition thickness of the Pearl River-derived sediments (Figure 7d) reveals significant deposition within the "Proximal" region, with many areas exceeding 10 mm despite wintertime resuspension and redistribution. In the "Eastern" region, deposition reaches a magnitude of 0.1 mm, while the inner shelf west of the PRE ("Western" and "Gulf" regions) exhibits significantly greater accumulation. For instance, the deposition west of the Chuanshan Islands reached a magnitude of 0.5 mm. In the "Gulf" region, deposition is primarily concentrated in the northeastern part, extending southwestward along the 30-60 m isobaths. Sediments transported southwestward along the east coast of Hainan Island and into the "Distal" regions remain largely suspended in the water column due to the greater water depth, with limited deposition on the seabed.*

[Figure]

*Figure 7. Riverine sediment (classes 4 and 5 in Table 1) retention budget percentages at eight regions (see Figure 1) during (a) the wet summer season, (b) the dry winter season, and (c) the entire year in the Control run case. (d) the annual deposition patterns spanning from April 1st, 2017, to March 31st, 2018 in the Control Run. All percentages displayed in the figure are relative to the annual riverine sediment load (see Figure 3a). The black percentage values represent the combined total of riverine sediment Class 4 and Class 5, while the red and blue values denote sediment Class 4 and Class 5, respectively. Arrows indicate the direction of net riverine sediment flux at each transect during the specified period."*

12. Figures 7 and 12: These figures are still confusing and there is probably a more effective way to present this data. The numbers are hard to see and hard to compare. The percentages are relative to the "total riverine load" which was 40% for class 4 and 60% for class 5. The blue numbers therefore show that *all* of class 5 is retained

in the PRE; except Figure 12(f) where 59.99% remains.

Response:

Thank you for your comments. The results presented in Figures 7 and 12 reflect our calculations accurately, showing that nearly all of Class 5 sediment is retained in the PRE, except for Figure 12(f) where about 59.99% remains. Regarding the clarity of the figures and numbers, we believe that the PDF version we uploaded is sufficiently clear for interpretation. However, we would appreciate any specific suggestions or examples from you on how we might improve the presentation for better readability and comparison.

13. Table 3: this is interesting but seems tangential to the paper and adds ½ a page to an already long paper.

Response: Thank you for the suggestion. We have removed Table 3 as recommended.

Text suggestions:

14. The writing could be improved. I will not try to provide an exhaustive list, but here are some specific examples.

Response:

Thank you for the valuable feedback. We have carefully revised the manuscript to improve the writing throughout.

15. Line 104: "SCS shaped by the East Asian Monsoon". That is confusing, the sea is

not "shaped" by the monsoon.

Response:

Thank you for the comment. We have revised the sentence to read "*The northern SCS, under the influence of the East Asian Monsoon,*" to avoid the confusion caused by "shaped by the East Asian Monsoon."

16. Line 143: "Holocene sedimentary processes": do you mean "Holocene sedimentary deposits"?

Response:

Thank you for the comment. We have revised "Holocene sedimentary processes" to "Holocene sedimentary deposits" as suggested.

17. Line 156: suggest the word "factors" or "drivers" instead of "processes".

Response:

Thank you for the helpful suggestion. We have replaced the word "processes" with "drivers" in the manuscript as recommended.

18. Line 241: suggest "Water level and current velocity".

Response:

Thank you for the suggestion. We have revised the phrase to "Water level and current velocity" as recommended.

19. Line 296, 300 and elsewhere: suggest "1,981" instead of "1981".

Response:

Thank you for the suggestion. We have revised the number format to "1,981" instead of "1981" as recommended.

20. Line 743: "Vectors show the seasonal mean surface current fields in each experiment". This is unclear because the figure panels are difference plots; not plots for a single experiment.

Response:

Thank you for your comment. We confirm that the figures are indeed plotted this way, following the approach used in previous studies. Specifically, Figures 9 and 10 present the seasonal surface currents and the differences in suspended sediment concentration (SSC) between the Control and sensitivity runs. The colors indicate riverine SSC differences between model runs, while the arrows represent surface currents from the non-Control runs.

This presentation allows us to show both the circulation patterns in each experiment and the differences in riverine SSC relative to the Control run. Because the experiments in Figure 10 share the same hydrodynamics as the Control run, comparing Figures 9 and 10 reveals the circulation differences between NTS, NWS, NAS, and the Control case.

This style of presenting both velocity vectors and SSC differences by comparing experimental cases with a Control run has been commonly used. For example, Xue et

al. (2012) illustrated current velocity and SSC differences in their Figures 5 and 6 by comparing experimental cases with the Control run on the Mekong River Shelf (Continental Shelf Research, 37, 66–78).

**References**

Zuo Xue, Ruoying He, J. Paul Liu, and John C. Warner (2012). Modeling transport and deposition of the Mekong River sediment. Continental Shelf Research, 37, 66–78. https://doi.org/10.1016/j.csr.2012.02.010.

21. Line 901: suggest "all flocs in the model are retained…". Can the authors use this model result to conclude something about the processes that are important?

Response:

Thank you for your valuable suggestion. We have revised the text accordingly. Our model results indicate that high-settling flocs are largely retained within the "Proximal" region, suggesting that these sediment particles have limited transport beyond this area. This highlights the importance of settling velocity in controlling sediment retention and dispersal processes in the region.

22. Avoid terms in the conclusion that are very specific to this paper; e.g. "regions 3-4" (Line 1150).

Response:

Thank you for your valuable suggestion. We have revised the manuscript to ensure that specific region names such as "regions 3-4" are no longer used in the conclusion

and abstract sections.

23. Data availability: The authors' data products should be listed at the beginning instead of at the end of this section. Lines 1193 – 1194 provide the link to the authors' own data products. Lines 1181 – 1192 provide links to data that were not generated by the authors.

Response:

Thank you for your valuable suggestion. We have rearranged the Data Availability section accordingly, placing the links to our own data products at the beginning, followed by links to data not generated by us.

**Response to Referee 3 (Report #1)**

I have read the revision and the responses to both reviews of the second submission. I think the authors have done a satisfying job in responding to all comments and have provided a new manuscript that presents its results much more successfully. The submission communicates the dominant processes that lead to the spreading of river-derived sediment over the continental shelf well.

I advise accepting this paper for publication, after some technical adjustments:

Response:

Thank you very much for your positive evaluation and constructive suggestions. We appreciate your time and effort in reviewing our manuscript and will carefully address the technical adjustments as advised.

1. Line 99: Is it possible to make the URL a reference?

Response:

Thank you for the suggestion. We have revised the sentence to cite the source as a reference, as shown below:

"*The present average annual (2001-2022) freshwater and riverine sediment loads are 2.74 × 10$^{11}$ m$^3$ and 2.84 × 10$^7$ tons, respectively (Ministry of Water Resources of the PRC, 2022).*

***References***

*Ministry of Water Resources of the PRC. 2022. Bulletin of River Sediment in China. http://www.mwr.gov.cn/sj/#tjgb.*"

2. Lines 160-163 and Line 171. In my opinion, manuscript-specific jargon should not appear in the introduction. This especially refers to the references to sediment classes and Table 1 in lines 156 – 175. Please consider rephrasing to retain generality in the introduction and specify further details in the methodology section 2.

Response:

Thank you for the insightful comment. We agree that manuscript-specific jargon, such as sediment classes and references to Table 1, is better suited for the methodology section. Accordingly, we have revised the introduction to maintain generality and moved the detailed descriptions of sediment classification and related references to Section 2.

3. Line 278, please consider more commonly used variables such as c or C_s for concentration and Q for discharge, similar to Zhang et al., (2012).

Response:

Thank you for the helpful suggestion. We have revised the notation to adopt more commonly used variables, specifically $C\_s$ for concentration and $Q$ for discharge, in line with the conventions used by Zhang et al. (2012).

4. Line 299, Figure 2: Please rephrase "After realistic reworking" to "After spin-up"

Response:

Thank you for the suggestion. We have revised the label in Figure 2 from "After realistic reworking" to "After spin-up" as recommended. The updated Figure 2 and its caption have been modified accordingly.

[Figure]

**Figure 2.** Row 1 presents the spatial distribution patterns of seabed sediment fractions derived from 1,981 sampling sites, while Row 2 demonstrates the initial spatial distribution prototype of seabed sediment fractions developed based on the observational data presented in Row 1. Row 3 shows the spatial distribution patterns of seabed sediment fractions following the completion of spin-up phase in the Control run case on April 1st, 2017, with Columns 1, 2, and 3 representing the fractions of clay, silt, and sand, respectively.

5. Line 651, Figure 7: Suggestion, is it worth it to add red and blue arrows as well to indicate the flux direction of each sediment class?

Response:

Thank you for the suggestion. We appreciate the idea of adding arrows to indicate flux directions for each sediment class. However, we believe this may not be necessary, as Class 5 sediments are largely trapped within the Pearl River Estuary (PRE), and the sediments transported over the shelf primarily belong to Class 4. Therefore, adding additional arrows for other classes may introduce visual redundancy without significantly enhancing interpretability.

6. Line 703-706. Please rephrase this sentence. The words "while" and "but" together seem to be used incorrectly?

Response:

Thank you for your comment. We have revised the sentence accordingly. The updated sentence now reads: "*the NTS case demonstrates that tides significantly enhance bottom stress (Figures 8a–b), while having minimal impact on the mean circulation.*"

7. Line 876: Please consider removing the term "realistic reworking".

Response:

Thank you for the suggestion. We have removed the term "realistic reworking" as recommended.

8. Line 1067 and Line 1072, Table 3: While I appreciate the efforts of adding this information, I do not think it is a suitable addition to this manuscript. Highlighting

that the PRE is dominant, with 83% of the sediment load, is sufficient to attain the goal of the paper. Advise is to remove Table 3.

Response:

Thank you for the suggestion. We have removed Table 3 as recommended.

9. Lines 1167-170. To me, this was a confusing addition for me at the end of the paper. Please avoid introducing a new term and explaining its meaning (Cycle2) in the last sentence of the manuscript.

Response:

Thank you for the comment. We agree that introducing a new term at the end could be confusing, and we have removed the related content accordingly.